# Archilles' Heel in Semi-open LLMs: Hiding Bottom against Recovery Attacks

## Abstract

To address privacy concerns with large language models, industrial users request local fine-tuning and deployment, but unencrypted models risk theft. Hardware-based security provides protection but is constrained by secure memory, leading to semi-open configurations. Semi-open models balance security and customization by keeping key layers closed-source within a secure environment while allowing others to be fine-tuned, but closed-source layers are susceptible to recovery attacks. In this paper, we explore the design of semi-open models with fewer closed-source layers, aiming to increase customizability while ensuring resilience to recovery attacks. We analyze the contribution of closed-source layer to the overall resilience and theoretically prove that in a deep transformer-based model, there exists a transition layer such that even small recovery errors in layers before this layer can lead to recovery failure. Building on this, we propose **SCARA**[1], a novel approach that keeps only a few bottom layers as closed-source. SCARA employs a fine-tuning-free metric to estimate the maximum number of layers that can be publicly accessible for customization. We apply it to five models (1.3B to 70B parameters) to construct semi-open models, validating their customizability on six downstream tasks and assessing their resilience against various recovery attacks on sixteen benchmarks. We compare SCARA to baselines and observe that it generally improves downstream customization performance and offers similar resilience with over **10** times fewer closed-source parameters. We empirically investigate the transition phenomenon and analyze the effectiveness and limitations of our scheme.

## 1 Introduction

Vendors of Large Language Models (LLMs) have recently introduced models with significant training costs and exceptional performance (Minaee et al., 2024; Zhao et al., 2023). Industrial users such as healthcare organizations, financial institutions, and government agencies, often seek locally deployed, fine-tuned models to prevent privacy leaks (NEVO et al., 2024; Guerra-Manzanares et al., 2023). However, deploying models without encryption exposes the architecture and parameters, making them vulnerable to model theft attacks that retrieve parameters from CPU, RAM, and vRAM, thereby risking vendors'

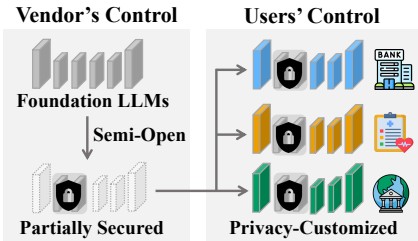

**Vendor's Control**     **Users' Control**

Foundation LLMs

Semi-Open

Partially Secured     Privacy-Customized

Figure 1: Semi-open Deployment.

intellectual property (Rakin et al., 2022; Hu et al., 2020). To mitigate this, vendors employ hardware-based security techniques such as Trusted Execution Environments (TEEs) and encrypted inference to safeguard model weights and prevent unauthorized access during inference (Nayan et al., 2024). Despite their advantages, TEEs have limited secure memory (e.g., 4GB–8GB in Intel TDX), insufficient for large language models (Dubey et al., 2024). This leaves part of the model exposed, creating a semi-open configuration that balances security and customization, as shown in Figure 1.

This semi-open approach introduces a tradeoff between model security and customizability. Closed-sourcing or securing more parameters limits the fine-tuning on private data, reducing the ability of model to adapt to specific needs. Conversely, open-sourcing or exposing more parameters increases

---

vulnerability to model recovery attacks (Zanella-Beguelin et al., 2021), where attackers query the secured module and train a new module to imitate its functionality (Solaiman, 2023). While the community has extensively studied recovery attacks against closed-source models (Chen et al., 2023; Jiang et al., 2023b), defending against recovery attacks in the semi-open setting is an uncharted area. Recovery attackers targeting fully closed-source models seek to fine-tune a new model that precisely replicates the closed-source model (Tamber et al., 2024; Dubiński et al., 2024). In contrast, attackers in semi-open settings are not required to exactly replicate the closed-source module. Instead, they can fine-tune the concealed module alongside the exposed module to reconstruct the overall functionality.

Beyond the closed-source amount, the specific sections concealed are vital for defending against recovery attacks. Shen et al. (2023) suggested concealing several top layers and keeping the bottom public. However, the benefits of hiding bottom layers (near the input) versus top layers (near the output) are still unclear. Therefore, we examine the impact of each layer on resilience and theoretically identify a transition layer. Any recovery error in bottom layers before this transition layer leads to a high probability of recovery failure. In contrast, errors in later layers have limited impact. This finding suggests that keeping the bottom layers closed-sourced offers better protection against recovery attacks than the top layers, even when the same number of layers is hidden.

In this paper, we introduce SCARA, a selective closed-sourcing method for designing semi-open models that balances customizability with resilience against recovery attacks. Building on our theoretical findings, SCARA selectively hides a few bottom layers. It determines the minimal number of closed-source layers using a recovery difficulty score, a metric that estimates recovery performance without requiring fine-tuning. This score is based on the initial average recovery loss during the attack. SCARA identifies the optimal closed-source strategy by selecting the layers with metric corresponding to the worst recovery performance. Consequently, models designed by SCARA retain most layers as publicly accessible, achieving customizability comparable to fully open-source models while remaining resilient to recovery attacks. Our main contributions are as follows:

- We theoretically demonstrate the existence of a transition layer in LLMs. We prove that small recovery errors in bottom decoder layers before this layer can lead to recovery failure with high probability, whereas errors in later layers have a limited impact. (see Section 4.1)

- We introduce SCARA, a selective closed-source approach that conceals a few bottom decoder layers to enhance customizability while maintaining resilience. Specifically, we propose a metric that does not require fine-tuning, but correlates with the recovered performance under attacks, enabling us to approximately find the minimal number of hidden layers. (see Section 4.2)

- We compare our approach with two baselines across five models (1.3B to 70B parameters), assessing customizability on six tasks and resilience against three recovery strategies across sixteen benchmarks in six domains. Experiments show that our method significantly improves downstream performance while maintaining comparable resilience against recovery attacks, with over 10 times fewer closed-source parameters than the baselines. For example, the semi-open Llama2-70B produced by our method hides only 2.5% of the parameters but achieves a 30% higher downstream performance score than the baselines in the Financial domain. We also observe a performance improvement of over 40% on Mistral-7B. Additionally, our method maintains similar resilience against recovery attacks compared to both baselines. (see Section 5.2)

- We empirically investigate the presence of transition layers and the correlation between our metric and the recovered performance of each closesd-source combination. We conclude by analyzing the hyper-parameter sensitivity and discuss the limitations of our approach. (see Section 5.4)

## 2 RELATED WORKS

**Model Customization.** Vendors have introduced three main strategies for model customization, each with distinct trade-offs. First, fine-tuning APIs allow customization of fully closed-source models (e.g., La Plateforme, Azure AI Services) while restricting access (Finlayson et al., 2024). Second, embedding models offer richer customization by enabling users to select and modify subsequent structures (Sarıtaş et al., 2024; Lee et al., 2024), but lack of joint pre-training may degrade performance (Nussbaum et al., 2024) and increase vulnerability to recovery attacks (Caspari et al., 2024; Tamber et al., 2024). Third, open-source models offer full customization flexibility yet pose challenges to model control and usage supervision (Bommasani et al., 2022; Roumeliotis et al., 2023).

**Model Recovery Attacks.** Prior attacks (Tramèr et al., 2016; Krishna et al., 2020; Dziedzic et al., 2023a) attempt to recover the functionality of fully closed-source models through API queries. Carlini et al. (2024) advanced these by entirely extracting the embedding projection layer and hidden dimension size. Recently, various defenses against fully closed-source model recovery attacks have been proposed (Jiang et al., 2024), including malicious queries detection (Shang et al., 2024), watermarking (Zhang et al., 2021), fingerprinting (Guan et al., 2022), etc. These methods do not directly apply to the semi-open settings with only partial model information.

**Semi-Open Model.** Previous studies (Lin et al., 2024; Chen et al., 2024; Qiao & Zhou, 2023) explore opening bottom layers of models for user customization, while keeping later layers closed-sourced to maintain vendor control. For example, Shen et al. (2023) introduced SAP, which open-sources the first six transformer layers but limits customization options. Meanwhile, Dubiński et al. (2024); Dziedzic et al. (2023b) proposed a semi-open approach where encoder models are offered as APIs, allowing users to customize task-specific subsequent modules. However, (Liu et al., 2022) and Sha et al. (2023) showed that these encoder models are still vulnerable to recovery attacks.

## 3 PRELIMINARIES

### 3.1 SECURITY THREAT: SEMI-OPEN MODEL RECOVERY

**Semi-open LLMs.** Let $\mathbf{X} \in \mathbb{R}^{n \times d}$ denote the input data matrix, where each row corresponds to a $d$-dimensional feature vector representing a single token. Let $f : \mathbb{R}^{n \times d} \to \mathcal{Y}$ denote a victim model, capable of processing the feature matrix $\mathbf{X}$ and producing an element in the set $\mathcal{Y}$ as output. Modern LLMs typically adopt a multi-layer architecture to capture complex patterns in the input data. Specifically, $f$ is a composition of multiple decoder layers, i.e., $f(\mathbf{X}; \boldsymbol{\theta}) = \varphi_L \circ ... \circ \varphi_1(\mathbf{X})$. All decoder layers $\varphi_1, ..., \varphi_L$ share the same architecture but each layer is equipped with distinct parameters. The parameters of all layers are denoted by the vector $\boldsymbol{\theta}$. We consider a semi-open setting, in line with Zanella-Béguelin et al. (2021) and Xu et al. (2021), where certain layers of the LLM are closed-sourced while others remain public. Let the closed-sourced set $I \subseteq \{1, ..., L\} \triangleq [L]$ denote the index set of hidden layers, while its complement $I^c$ contains the public layer indices.

**Semi-open Model Recovery Attack.** The semi-open model recovery attack aims to replicate a target language model (LLM) (Carlini et al., 2024). Under the threat model (Shen et al., 2023), the adversary can query the semi-open model, access its output logits, and retrieve output representations from the closed-source module. With knowledge of the closed-source architecture but not its parameters, the adversary fine-tunes a replacement model us-

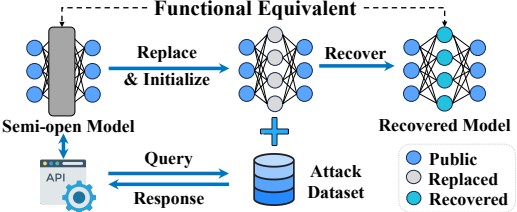

Figure 2: Workflow of semi-open model recovery attack

ing these logits or representations as training labels. As shown in Figure 2, the attack begins by constructing a dataset $\mathcal{D}$ through queries to the victim model and employs three attack strategies: (1) **FT-all**, which fine-tunes both the replacement and open-source modules using logits; (2) **FT-closed**, which fine-tunes only the replacement model using logits while keeping the open-source module fixed; and (3) **SEM** (Tamber et al., 2024), which fine-tunes the replacement model using representations without involving the open-source module. Let $\boldsymbol{\theta}_{\text{FT}}(I, \mathcal{D})$ represent the recovered parameters under the attack dataset $\mathcal{D}$ and the closed-source set $I$.

### 3.2 PROBLEM FORMULATION

In this paper, we consider the performance of a large language model within a defined distribution, denoted as $\mathbb{P}_{\mathbf{X} \times Y}$, representing the relationship between the input matrix $\mathbf{X}$ and corresponding label $Y$. We assume that the victim LLM $f(\mathbf{X}; \boldsymbol{\theta})$ performs well within this distribution. Additionally, we presume the attack set $\mathcal{D}$ consists of independent and identically distributed (i.i.d.) samples drawn from $\mathbb{P}_{\mathbf{X} \times Y}$. To assess the alignment between the outputs of LLM and the ground-truth labels, we use a scoring function, denoted as $s : \mathcal{Y} \times \mathcal{Y} \to \mathbb{R}^+$. For any closed-source index set $I \subseteq [L]$, we introduce the concept of a "**Recovery Ratio**" $R(I)$. This ratio measures the extent to which the

recovered model $\boldsymbol{\theta}_{\mathrm{FT}}(I, \mathcal{D})$ can replicate the behavior of the victim model $f(\mathbf{X}; \boldsymbol{\theta})$, expressed as

$$R(I) = \frac{\mathbb{E}[s(f(\mathbf{X}; \boldsymbol{\theta}_{\mathrm{FT}}(I, \mathcal{D})), Y)]}{\mathbb{E}[s(f(\mathbf{X}; \boldsymbol{\theta}), Y)]}. \tag{1}$$

Here, $\mathbb{E}$ in the numerator reflects the expectation computed over random samples $(\mathbf{X}, Y)$ drawn from $\mathbb{P}_{\mathbf{X} \times Y}$, the random attack set $\mathcal{D}$, and the random initialization of parameters within the closed-source layers during fine-tuning. Conversely, the term $\mathbb{E}$ in the denominator solely considers the expectation over random samples. With this definition, the term $R([L])$ denotes the recovery ratio of the recovered model under a fully-closed approach, where $[L] = \{1, ..., L\}$. Hence, the following question arises.

*Given $\varepsilon > 0$, what is the smallest closed-source set $I$ for which $R(I) \leq (1 + \varepsilon)R([L])$?*

This question essentially asks whether it is feasible to identify a minimal closed-source index set $I$, such that, under this closed-source strategy, the resulting recovered model exhibits similarity to the model recovered under fully-closed approach. In other words, the recovery score does not surpass that of fully-closed approach by more than a factor of $(1 + \varepsilon)$.

## 4 METHODOLOGY

In this section, we investigate how each layer affects customizability and resilience against recovery attacks. We begin with an experiment involving two semi-open Llama2-70B models, each with either the first two (Semi-Open-1) or the last two (Semi-Open-2) decoder layers closed-sourced. We compare their customization performance and recovered performance under the recovery attack. Figure 3 (a) and (b) show that although two semi-open models perform similarly on six downstream tasks, closed-sourcing the first two layers offers significantly greater resilience than the last two. Moreover, we compare the Semi-Open-1 model to the fully-closed model and observe that this semi-open model can achieve better customizability and comparable resilience at the same time. Therefore, we conjecture that, with a sufficient number of closed-sourced layers before a certain transition layer, a semi-open model can simultaneously achieve great customizability on downstream tasks and strong resilience against recovery attacks. In this section, we first present a theoretical result showing the existence of transition layers and then introduce our selective closed-sourcing approach.

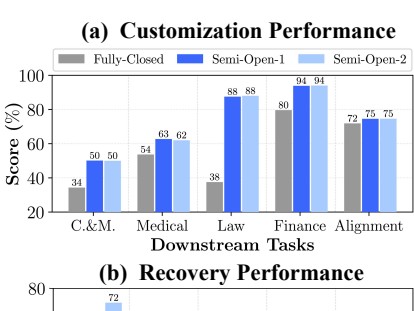

(a) **Customization Performance**

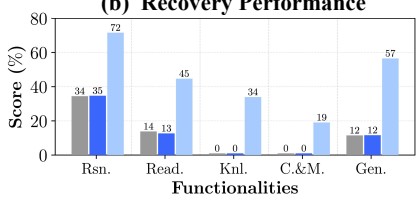

(b) **Recovery Performance**

Figure 3: Customizability and resilience comparison in Llama2-70B. Higher scores indicate better customizability in Fig. (a) and weaker resilience in Fig. (b). Details can be found in Appendix C.1

### 4.1 RESILIENCE TRANSITION LAYER IN INFINITELY DEEP TRANSFORMERS

**Model Overview.** Let us revisit our large language model composed of $L$ layers, denoted as $f(\mathbf{X}; \boldsymbol{\theta}) = \varphi_L \circ ... \circ \varphi_1(\mathbf{X})$. Recall that each row of the feature matrix $\mathbf{X} \in \mathbb{R}^{n \times d}$ represents a $d$-dimensional vector for an input token. We treat each layer $\varphi_i$ as a transformer layer, where each layer processes an $n \times d$ dimensional matrix as input and outputs another $n \times d$ matrix. Thus, the model $f$ outputs a matrix of $n$ rows and $d$ columns, indicating that the large language model outputs a feature vector for each token. Moreover, we assume that each layer contains a normalized residual self-attention function, defined as

$$\varphi_i(\mathbf{X}; K_i, Q_i) = \mathbf{X} + \mathrm{softmax}\left(\frac{\mathbf{X}Q_i(\mathbf{X}K_i)^\top}{\sqrt{d_Q}\|\mathbf{X}\|^2}\right)\mathbf{X}, \tag{2}$$

where $Q_i \in \mathbb{R}^{d \times d_Q}$ and $K_i \in \mathbb{R}^{d \times d_Q}$ are projection parameter matrices for the $Q$ and $K$ matrices in the transformer, respectively. Additionally, $\sqrt{d_Q}$ and the matrix norm $\|\mathbf{X}\|$ denote normalization factors provided by the normalization layer. We consider the strategy of concealing the $\alpha L$-th layer with $\alpha \in [0, 1]$ and $\alpha L \in \mathbb{N}$ while keeping other layers public. After the semi-open model recovery, we assume the parameters of the recovered model in the public layers are identical to the victim

model, while those in the proprietary layer deviate. Let $\hat{K}_{\alpha L}$ and $\hat{Q}_{\alpha L}$ denote the recovered weight matrix of the proprietary layer, i.e., $\boldsymbol{\theta}_{\text{FT}}(\{\alpha L\}) = \{(K_1, Q_1), ..., (\hat{K}_{\alpha L}, \hat{Q}_{\alpha L}), ..., (K_L, Q_L)\}$. Let $\hat{\varphi}_{\alpha L}$ denote the function of the recovered proprietary layer, i.e., the $\alpha L-$th layer, in the recovered model. In this subsection, we consider the normalized output of an infinitely deep model whose $\alpha L$-th layer is closed-sourced and subjected to the attack. The output of the recovered model is

$$\hat{f}_{\infty}(\mathbf{X}) = \lim_{L \to \infty} \frac{\varphi_L \circ ... \varphi_{\alpha L+1} \circ \hat{\varphi}_{\alpha L} \circ \varphi_{\alpha L-1} \circ ... \circ \varphi_1(\mathbf{X})}{\|\varphi_L \circ ... \varphi_{\alpha L+1} \circ \hat{\varphi}_{\alpha L} \circ \varphi_{\alpha L-1} \circ ... \circ \varphi_1(\mathbf{X})\|_F},$$

where $\| \cdot \|_F$ denotes the Frobenius norm of a given matrix. We consider this infinitely deep network as our ideal model because, in real-world settings, most large-scale models are sufficiently deep. Next, we present the following theorem to illustrate the existence of a critical value $\alpha^*$ such that if $\alpha < \alpha^*$, the recovered LLM outputs identical feature vectors for all tokens. Conversely, if $\alpha > \alpha^*$, the output feature vectors may vary across tokens.

**Theorem 1.** *Assume that $\mathbb{P}_{\mathbf{X} \times Y}$ is defined on a countable domain $\mathcal{X} \times \mathcal{Y}$ with $\mathbf{0}_{n \times d} \notin \mathcal{X}$. Assume that parameter matrices $\{K_i, Q_i\}_{i \geq 1}$ in the victim model $f$ have uniform bounded norms, i.e., $\|K_i\| \leq D$ and $\|Q_i\| \leq D$ for some $D > 0$. There exists an $\alpha^* \in (0, 1)$ depending on $D$ such that the following two statements are true.*

*(1) Let $\alpha < \alpha^\star$ and $\{K_i, Q_i\}_{i \geq 1}$ be any parameter matrix sequence in the victim model. Let $\hat{K}_{\alpha L}$ and $\hat{Q}_{\alpha L}$ be the recovered parameter matrices drawn from a continuous distribution supported on $\mathbb{R}^{n \times d}$. With probability one, for any input $\mathbf{X} \in \mathcal{X}$, the row vectors in the matrix $\hat{f}_{\infty}(\mathbf{X})$ are identical.*

*(2) Let $\alpha > \alpha^\star$. There exists a victim model with parameter matrix sequence $\{K_i, Q_i\}_{i \geq 1}$ such that for any recovered parameter matrices $\hat{K}_{\alpha L}$ and $\hat{Q}_{\alpha L}$, the row vectors in the matrix $\hat{f}_{\infty}(\mathbf{X})$ are not entirely the same for some input feature matrix $\mathbf{X} \in \mathcal{X}$.*

**Remark 1:** The proof is provided in Appendix A. This theorem demonstrates that if the recovered parameters of the bottom layers (i.e., $\alpha < \alpha^*$) are obtained through a randomized algorithm, such as stochastic gradient descent, with a continuous distribution supported on $\mathbb{R}^{n \times d}$, the recovery will certainly fail, as it will produce the same feature vector for every token. In contrast, keeping the later layers closed-sourced (i.e., $\alpha > \alpha^*$) does not maintain this property, indicating that it is more effective to closed-source the bottom layers before the transition layer, rather than the later ones.

**Remark 2:** The theorem relies on the assumption that the distribution is defined over a countable domain, $\mathcal{X} \times \mathcal{Y}$, typically satisfied by inputs such as sentences or images. We show in the proof that for each input matrix $\mathbf{X} \in \mathcal{X}$, there are two zero-measure sets $\mathcal{K}(\mathbf{X})$ and $\mathcal{Q}(\mathbf{X})$ such that the recovered matrices must avoid to satisfy the theorem. Hence, the countable unions $\mathcal{K} = \bigcup_{\mathbf{X} \in \mathcal{X}} \mathcal{K}(\mathbf{X})$ and $\mathcal{Q} = \bigcup_{\mathbf{X} \in \mathcal{X}} \mathcal{Q}(\mathbf{X})$ are also zero-measure sets, ensuring that when recovered matrices do not belong to these sets, the conditions in the theorem are met for any input matrix $\mathbf{X}$ in the input space.

Theorem 1 shows that hiding bottom layers improves resilience, suggesting closed-sourcing from the first layer may be effective. Next, we present an approach to identify the minimal set of hidden layers.

## 4.2 SCARA: Selective Closed-sourcing Approach Against Recovery Attack

We propose a method to approximately find the smallest bottom layer index set $I$ that satisfies $R(I) \leq (1 + \varepsilon)R([L])$. A simple approach is to start with $I_l = \{1, \ldots, l\}$ for each $l$ beginning from 1, then evaluate the recovery ratio $R(I_l)$ after the attack, and identify the smallest $l$ that meets the inequality. This extensive fine-tuning process is time-consuming, prompting the critical question: *Can we create a fine-tuning-free metric that predicts LLM performance under semi-open model recovery attacks?* Hence, our goal is to establish a metric directly correlated with the recovery ratio.

In the recovery ratio $R(I)$, each $I$ has the same denominator, so our focus is on a metric related to the numerator, specifically $\mathbb{E}[s(f(\mathbf{X}; \boldsymbol{\theta}_{\text{FT}}(I, \mathcal{D})), Y)]$, which measures the average performance score of the recovered model. This average performance score generally inversely correlates with the average testing loss $\mathbb{E}[\ell(f(\mathbf{X}; \boldsymbol{\theta}_{\text{FT}}(I, \mathcal{D})), Y)]$, where $\ell$ denotes the cross-entropy loss employed by LLM. Therefore, our goal becomes finding the $l$ such that

$$\mathbb{E}[\ell(f(\mathbf{X}; \boldsymbol{\theta}_{\text{FT}}(\{1, ..., l\}, \mathcal{D})), Y)] \geq (1 - \varepsilon)\mathbb{E}[\ell(f(\mathbf{X}; \boldsymbol{\theta}_{\text{FT}}([L], \mathcal{D})), Y)].$$

However, calculating both sides of this inequality requires knowing the recovered parameters from the fine-tuning process. To bypass this, we aim for an approximate solution. The recovered parameters

are generated through gradient descent, starting from the initial parameters $\boldsymbol{\theta}_0(I)$, with the hidden layers being randomly initialized. Using the Taylor Expansion, we find

$$\mathbb{E}\left[\ell\left(f(\mathbf{X};\boldsymbol{\theta}_{\text{FT}}(I,\mathcal{D})),Y\right)\right] = \mathbb{E}\left[\ell\left(f(\mathbf{X};\boldsymbol{\theta}_0(I)),Y\right)\right] + \mathcal{O}(\|\boldsymbol{\theta}_{\text{FT}}(I,\mathcal{D}) - \boldsymbol{\theta}_0(I)\|_2).$$

Previous research (Choi et al., 2024; Bailly et al., 2022) suggests the difference $\|\boldsymbol{\theta}_{\text{FT}}(I,\mathcal{D}) - \boldsymbol{\theta}_0(I)\|_2$ is minor for large networks compared to the dataset size $|\mathcal{D}|$. In models like a single-layer ReLU network (Anthony et al., 1999; Zou et al., 2020), the difference $\|\boldsymbol{\theta}_{\text{FT}}(I,\mathcal{D}) - \boldsymbol{\theta}_0(I)\|_2$ is of order $\mathcal{O}\left(\frac{|\mathcal{D}|}{\sqrt{N}}\right)$ (Jacot et al., 2018; Wei et al., 2019), where $N$, the number of model parameters, which is much larger than the dataset size in LLMs (Dubey et al., 2024; Liu et al., 2024). Hence, the first term that does not require fine-tuning dominates, suggesting it as a viable metric for predicting the recovery ratio. Thus, we define the first term as "**Recovery Difficulty**" (RD($I$)) with the expression:

$$\text{RD}(I) = \mathbb{E}_{\mathbf{X},Y,\boldsymbol{\theta}_{\mathbf{o}}(I)}\left[\ell\left(f(\mathbf{X};\boldsymbol{\theta}_0(I),Y)\right)\right].$$

This score, which can be estimated using a sample average, represents the recovered performance of the model when specific layers $I$ are closed-sourced. A higher **RD**($I$) suggests worse recovery performance, indicating a lower recovery ratio $R(I)$. Therefore, our SCARA operates in the following way. SCARA begins by sampling evaluation data targeting general capabilities from the underlying distribution, and then computes RD($I_l$) for each set of closed-sourced layers $I_l = \{1, ..., l\}$ for $l = 1, ..., L$. SCARA stops at the smallest $l^*$ that satisfies RD($I_{l^*}$) $\geq (1 - \varepsilon)$RD($[L]$).

## 5 EXPERIMENTS

### 5.1 EXPERIMENTAL SETTINGS

In this subsection, we introduce the experimental setups. Details can be found in Appendix B.

**Models.** We consider **five** open-source, decoder-only structured LLMs with various architectures. Specifically, we select Llama2-70B-chat, Llama2-7B-chat (Touvron et al., 2023), Mistral-7B-v0.1 (Jiang et al., 2023a), Phi-2 (Abdin et al., 2024), and Phi-1.5 (Li et al., 2023). We designate these pre-trained models as the base models for customization and victims in semi-open model recovery attacks.

**Attack Methods.** We recover models produced by different closed-source approaches using three attack methods: FT-all, FT-closed and SEM. Following (He et al., 2021), a diverse attack set is required for full recovery. Therefore, we merge data evenly form two general datasets, MMLU benchmark (Hendrycks et al., 2021) and Alpaca 52k (Wang et al., 2022), resulting in a 51k combined set. Moreover, we also construct four larger general datasets (100k–500k) to strengthen the attack.

**Baselines.** We compare SCARA with the other two baselines: SAP-DP and the fully-closed (Eiras et al., 2024) approach. The SAP (Shen et al., 2023) framework keeps the first six decoder layers open and the rest closed-source. SAP-DP extends SAP by adding Laplace noise to the model outputs, a common strategy for model protection (Lee et al., 2018). The fully-closed approach represents the extreme, where all layers are closed-sourced.

**Implementation Details of SCARA.** We apply the SCARA algorithm to identify the smallest closed-source set $I$ such that $R(I) \leq (1 + \varepsilon)R([L])$. To calculate recovery difficulty (RD), we use cross-entropy loss and approximate the expectation over samples distributed on the general domain and randomly initialized closed-source parameters. This is done using a 1,500-sample evaluation set randomly sampled from the MMLU benchmark and Alpaca 52k, with closed-source parameters initialized via Xavier initialization and averaged over three random seeds (20, 42, 1234). For models up to 7B parameters, we use four RTX 4090 GPUs, while for Llama2-70B, we use four A100 GPUs. We find that $\varepsilon = 0.05$ yields optimal performance. For $\varepsilon$ sensitivity, see Section 5.3.

**Evaluation Benchmarks** We assess customizability on six downstream tasks: Code (Zheng et al., 2024b), Math (Yue et al., 2023), Medical (Zhang et al., 2023), Finance (Wang et al., 2023b), Law (Guha et al., 2024), and Alignment (Meng et al., 2024). To fully evaluate recovered functionalities, we focus on six capabilities domains following Llama2 report (Touvron et al., 2023). Specifically, we assess the recovered model across **sixteen** benchmarks grouped into (1) *Commonsense Reasoning* (Rsn.); (2) *Reading Comprehension* (Read.); (3) *World Knowledge* (Knl.); (4) *Code*; (5) *Math*; and (6) *General Ability* (Gen.).

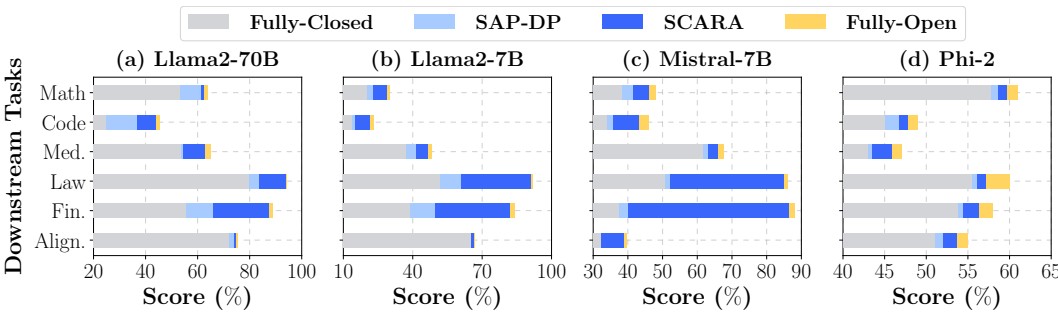

Figure 4: Customization performance of models closed-sourced by SCARA on six downstream tasks.

Table 1: Recovery ratios on 6 functionalities under FT-all (SCARA |SAP-DP| Fully-closed). "H.E." in Code domain presents the benchmark "HumanEval". More details are available in Appendix C.2.

| | Benchmark | Llama2-70B | Llama2-7B | Mistral-7B | Phi-2 | Phi-1.5 |
|---|---|---|---|---|---|---|
| **Rsn.** | PIQA | 62.6\|59.8\|63.0 | 64.7\|64.7\|64.6 | 63.0\|61.2\|60.2 | 68.3\|65.6\|65.7 | 70.6\|69.5\|66.7 |
| | Winogrande | 68.5\|67.7\|68.3 | 76.8\|74.8\|76.6 | 67.2\|69.0\|68.3 | 68.3\|64.9\|64.8 | 70.3\|67.8\|67.6 |
| | ARC-easy | 31.9\|32.8\|31.3 | 36.3\|35.5\|34.9 | 32.3\|34.7\|32.0 | 43.2\|35.3\|33.9 | 40.5\|37.8\|36.1 |
| | ARC-challenge | 38.5\|38.1\|44.2 | 47.8\|46.6\|50.9 | 39.7\|42.6\|44.5 | 36.8\|36.6\|35.3 | 46.1\|44.4\|47.5 |
| | Hellaswag | 31.4\|31.4\|32.4 | 33.9\|34.0\|35.0 | 32.2\|32.0\|31.3 | 37.4\|37.3\|34.3 | 42.0\|41.0\|40.0 |
| **Read.** | LAMBADA | 0.01\|0.00\|0.00 | 0.02\|0.00\|0.01 | 0.16\|0.00\|0.01 | 1.34\|0.04\|0.00 | 1.37\|0.00\|0.00 |
| | BoolQ | 47.2\|47.1\|53.9 | 59.5\|56.0\|65.0 | 48.3\|46.8\|56.7 | 56.7\|50.3\|55.8 | 61.7\|54.9\|60.8 |
| | SQuADv2-EM | 0.00\|0.00\|0.00 | 0.00\|0.00\|0.00 | 0.00\|0.00\|0.00 | 0.00\|0.00\|0.00 | 0.00\|0.00\|0.00 |
| | SQuADv2-F1 | 1.50\|1.68\|0.34 | 0.68\|0.88\|0.82 | 1.69\|0.36\|0.93 | 3.65\|0.39\|0.90 | 1.28\|1.07\|2.64 |
| | OBQA | 54.5\|54.5\|57.1 | 57.4\|52.5\|59.2 | 57.7\|56.8\|56.3 | 0.00\|0.00\|0.02 | 0.04\|0.00\|0.00 |
| **Knl.** | NaturalQuestions | 0.00\|0.02\|0.00 | 0.01\|0.01\|0.08 | 0.00\|0.00\|0.02 | 0.01\|0.00\|0.06 | 0.21\|0.00\|0.00 |
| | TriviaQA | 0.00\|0.02\|0.00 | 0.00\|0.00\|0.03 | 0.00\|0.00\|0.01 | 0.01\|0.00\|0.01 | 0.01\|0.00\|0.00 |
| **Code** | MBPP&H.E. | 0.00\|0.00\|0.00 | 0.00\|0.00\|0.00 | 0.00\|0.00\|0.00 | 0.00\|0.00\|0.00 | 0.00\|0.00\|0.00 |
| **Math** | GSM8K | 0.02\|0.00\|0.06 | 0.00\|0.00\|0.00 | 0.00\|0.00\|0.00 | 0.00\|0.00\|0.00 | 0.00\|0.00\|0.00 |
| **Gen.** | MMLU | 36.8\|38.3\|36.5 | 52.9\|50.0\|53.3 | 40.4\|36.9\|37.2 | 42.6\|40.3\|40.5 | 56.7\|54.1\|54.1 |
| | BBH | 0.00\|0.00\|0.00 | 0.00\|0.00\|0.00 | 0.00\|0.00\|0.00 | 0.01\|0.00\|0.00 | 0.00\|0.00\|0.00 |
| **Average Recovery Ratio(↓)** | | 21.9\|21.8\|22.8 | 25.3\|24.4\|25.9 | 22.5\|22.4\|22.8 | 23.9\|22.3\|22.4 | 26.2\|25.3\|25.4 |
| **Closed-source Ratio(↓)** | | **2.50**\|92.5\|100. | **3.16**\|81.3\|100. | **3.16**\|81.6\|100. | **6.25**\|81.3\|100. | **8.33**\|75.0\|100. |

**Metrics.** We measure model customizability through its improvements on benchmarks. For resilience, we calculate the "Average Recovery Ratio" (ARR) by averaging the recovery ratios across benchmarks. A lower ARR indicates higher resilience offered by the closed-sourced set. Additionally, we define $\Delta\mathbf{ARR}(I) = \mathrm{ARR}(I) - \mathrm{ARR}([L])$ to compare the resilience between closed-sourcing set $I$ and the fully-closed approach. A smaller $\Delta\mathbf{ARR}$ suggests similar resilience to the fully-closed model.

## 5.2 MAIN RESULTS

In this subsection, we compare SCARA with three baselines, demonstrating its superior customizability on downstream domains while preserving similar resilience against model recovery attacks.

**Customizability: SCARA vs. Baselines.** We compare the customization performance of SCARA with closed-source baselines. Results are shown in Figure 4 and detailed in Appendix B.6.

On 70B and 7B models, SCARA consistently surpasses SAP-DP and fully-closed approaches across six domains and aligns closely with the performance of the fully-open approach, where all parameters are accessible. For instance, in the Law domain, SCARA improves scores by 10% over SAP-DP and fully-closed approaches on Llama2-70B, with this improvement rising to 35% on 7B models. Similar patterns of enhanced customizability are also evident in Phi-2 model, though the improvement on the Law domain narrows to only 1%. Furthermore, SCARA maintains performance comparable to the

Table 2: Recovery ratios on Llama2-70B.

| Strat. | Method | Rsn. | Read. | Knl. | C.&M. | Gen. | ARR |
|--------|--------|------|-------|------|-------|------|-----|
| **FT-c.** | SCARA | 47.1 | 21.6 | 0.00 | 0.03 | 18.7 | 22.6 |
| | SAP-DP | 46.2 | 19.5 | 0.00 | 0.00 | 19.0 | 21.8 |
| | F-Closed | 47.8 | 21.2 | 0.00 | 0.08 | 18.5 | 22.8 |
| **SEM** | SCARA | 48.2 | 21.9 | 0.00 | 0.00 | 18.5 | 22.4 |
| | SAP-DP | 47.1 | 21.1 | 0.00 | 0.00 | 18.3 | 22.3 |
| | F-Closed | 47.8 | 21.2 | 0.00 | 0.08 | 18.5 | 22.8 |

Table 3: ARR of SCARA vs. attacks.

| Model | FT-all | FT-closed | SEM |
|-------|--------|-----------|-----|
| Llama2-70B | 21.9 | 22.6 | 22.4 |
| Llama2-7B | 25.3 | 24.8 | 25.0 |
| Mistral-7B | 22.5 | 22.5 | 22.0 |
| Phi-2 | 23.9 | 23.7 | 22.1 |
| Phi-1.5 | 26.2 | 26.9 | 24.7 |

fully-open approach, demonstrating that hiding a small portion of parameters has a small impact on downstream customizability. More results are reported in Appendix C.3.

**Resilience: SCARA vs. Baselines.** We compare the resilience of SCARA with other closed-source baselines under three recovery strategies: FT-all, FT-closed, and SEM attack. As shown in Table 1, under FT-all attack, SCARA, SAP-DP, and fully-closed approach show similar resilience across various architectures and domains, with ARR differences within 1.4%. For instance, on Llama2-70B, SCARA keeps only 1.25% of parameters hidden, yet achieves an ARR of 21.9%, comparable to SAP-DP (21.8%) and the fully-closed approach (22.8%), which keep 92.5% and 100% of parameters closed-sourced, respectively. This pattern also extends across five architectures, indicating that our approach effectively preserves resilience with limited parameter closed-sourced. As shown in Table 2, SCARA achieves comparable resilience under FT-closed and SEM attacks. Under FT-closed attack, the recovery ratio differences among the three approaches remain under 2.1% across six domains. Similarly, under SEM attack, recovery ratios of SCARA align closely with the other approaches. These results show that SCARA offers resilience against model recovery on par with fully-closed approach, despite requiring fewer private parameters. More details are in Appendix C.4.

**Resilience: SCARA vs. Recovery Strategies.** Table 3 shows that SCARA effectively defends three recovery attack strategies on all models. We observe that SEM, a typical and effective attack for recovering embedding models, does not show a significant boost in recovery performance. This can be because SEM attackers focus on recovering only the proprietary embedding module, while the semi-open model recovery attackers aim at recovering the full functionality of the entire model, including both proprietary and public modules. The targets of these two attackers are different since even small errors in bottom layers can lead to significant output deviations. To see this, we add small perturbations on parameters in the first layer of Llama2-7B model and evaluate the hidden representation deviation at the output of each decoder layer. Figure 5 shows that the norm of deviation increases as the layer index increases, indicating that small errors are amplified by subsequent layers, leading to large deviations in the final output. Therefore, SCARA closed-sources the first several layers, effectively leveraging this amplification, making the functionality recovery more difficult and ensuring strong resilience against recovery attack. We report details in Appendix B.7 and C.5.

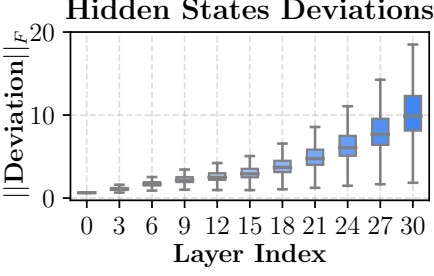

Figure 5: Amplification of error in Llama2-7B.

Table 4: SCARA vs. dataset scales.

| Scale | Rsn. | Read. | Knl. | C.&M. | Gen. | ARR |
|-------|------|-------|------|-------|------|-----|
| 51k | 51.7 | 21.6 | 0.01 | 0.00 | 28.3 | 25.3 |
| 100k | 51.3 | 21.5 | 0.13 | 0.00 | 29.6 | 25.3 |
| 200k | 51.4 | 21.7 | 0.11 | 0.00 | 29.7 | 25.2 |
| 300k | 51.6 | 21.7 | 0.11 | 0.00 | 30.5 | 25.5 |
| 500k | 51.8 | 22.0 | 0.09 | 0.00 | 30.8 | 25.8 |

**Resilience: SCARA vs. Recovery Dataset Scales.** We further evaluate the resilience of SCARA against FT-all by increasing the recovery dataset scale on Llama2-7B to determine if larger datasets would compromise its effectiveness. More details on the attack dataset are in Appendix B.2. Table 4 shows the recovery ratio achieved by SCARA under each attack dataset. We observe that increasing the scale of the attack dataset leads to only a mild increase in recovery ratios, indicating a limited impact on SCARA. For instance, recovering with the 500k samples results in only a 0.5% ARR improvement over 51k samples. This suggests that the resilience provided by SCARA cannot be easily compromised by simply increasing the dataset scale. Details are reported in Appendix C.6.

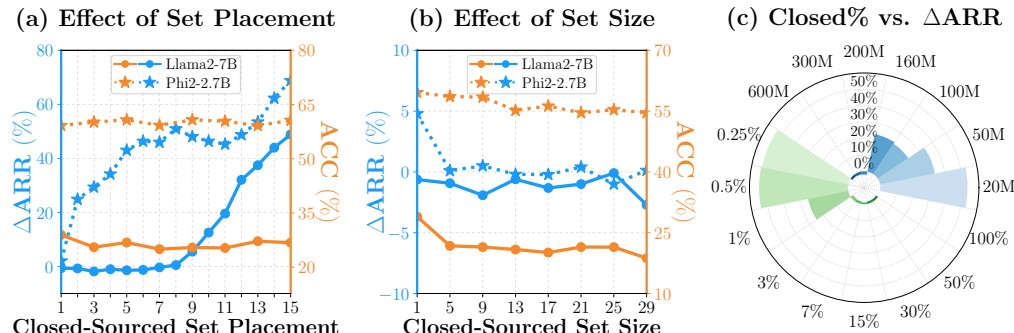

Figure 6: (a) shows the trends in customizability and resilience changes in Llama2-7B and Phi-2 with different placements of same-sized closed-source sets. (b) presents the patterns of customizability and resilience in Llama2-7B and Phi-2 as the closed-source set size varies, starting from the first decoder layer. (c) depicts $\Delta$ARR for different closed-sourced parameter quantities and proportions in Llama2-7B. Smaller $\Delta$ARR indicates similar resilience to the fully-closed model, while higher ACC reflects better customizability.

## 5.3 ANALYSIS OF THE CUSTOMIZABILITY-RESILIENCE TRADE-OFF IN SCARA

**Closed-source Module Placement vs. Trade-off.** Theorem 1 demonstrates that the bottom layers before a transition layer provide stronger resilience against model recovery attacks. However, it remains unclear how hiding these layers might impact model customizability. To investigate this, we designed semi-open models with closed-source layer sets of equal size using Llama2-7B and Phi-2. These models were customized for the math domain and evaluated under FT-all recovery attacks. Figure 6(a) shows that while the placement of the closed-source set has minimal impact on customizability, it significantly affects resilience, consistent with Theorem 1. For Llama2-7B, the resilience transition occurs at the eighth layer set, where $\Delta$ARR remains close to zero for earlier sets, indicating that hiding layers before this point ensures strong resilience. Importantly, customization accuracy remains stable regardless of placement, further supporting the effectiveness of hiding layers before the transition. In contrast, Phi-2 exhibits an earlier transition at the first layer set, where only the first layer achieves a balance between customization and resilience, with subsequent sets resulting in diminished resilience. These results suggest that placing the closed-source set before the transition layer optimizes the trade-off between customization and resilience against recovery attacks. Further analysis on Mistral-7B and Phi-1.5 is provided in Appendix B.8.

**Closed-source Module Size vs. Trade-off.** We investigate how the size of the closed-source module impacts the trade-off between customizability and resilience to recovery attacks. Semi-open models based on Llama2-7B and Phi-2 are created by incrementally increasing the number of hidden layers starting from the first. These models are customized on the math domain and evaluated for resilience under the FT-all attack, with results shown in Figure 6(b). For Llama2-7B, the results reveal a clear transition in customizability, while resilience remains largely unaffected by module size. Customization accuracy drops from 29% to 21% as the closed-source module grows from one to five layers, while $\Delta$ARR stays near zero, indicating strong resilience regardless of closed-sourced size. Further, as shown in Figure 6(c), resilience emerges when at least 3% of parameters—equivalent to a single decoder layer—are closed-sourced. This suggests that hiding the first layer alone provides the best trade-off between customization and resilience.

In contrast, Phi-2 shows a different pattern: as the closed-source module size increases, customization accuracy declines, but resilience improves significantly. This is evident from a marked decrease in $\Delta$ARR as the module size grows from one to five layers, suggesting enhanced resilience to recovery attacks. These findings indicate that larger models like Llama2-7B achieve an optimal balance with fewer closed-source layers, while smaller models like Phi-2 require more layers to maintain resilience against recovery attacks. Further analyses are provided in Appendix B.9.

## 5.4 DISCUSSIONS

**Effectiveness of RD on large models.** We assess the efficacy of the recovery difficulty (RD) in estimating the performance of the recovered model. Specifically, we calculate the Pearson and Spearman correlation coefficients between RD and ARR across different capability groups. As shown in Figure 7(a), we observe a negative correlation between the recovery difficulty and average recovery

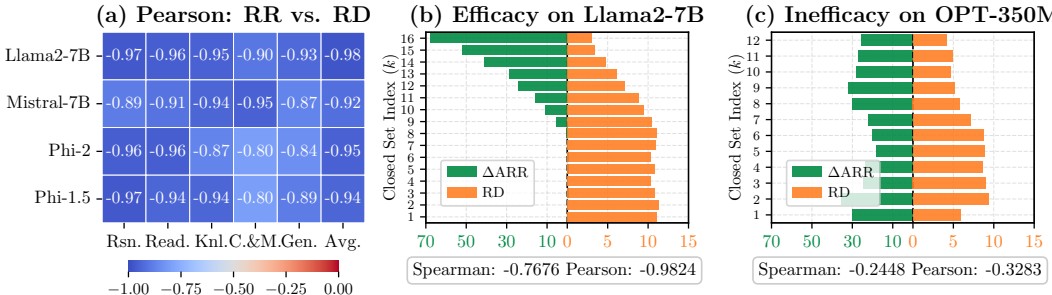

Figure 7: (a) presents the Pearson coefficient between recovery ratio (RR) and recovery difficulty (RD) across four models and six domains. (b) and (c) depict the link between ΔARR and RD for Llama2-7B and OPT-350M.

ratio. For example, in Llama2-7B, the Pearson coefficient is consistently below -0.80, reaching as low as -0.98. We observe similar phenomena in other models with varying architectures and sizes, confirming RD as a reliable predictor of recovered model performance and the effectiveness of SCARA. Further analysis and results of Spearman coefficients can be found in Appendix B.10.

**Ineffectiveness of RD on Smaller Model.** Theorem 1 and Figure 6(a) demonstrate the existence of transition layers in deep transformers, yet their presence in shallow transformers remains unclear. Therefore, we hide and attack same-sized layer sets in OPT-350M (Zhang et al., 2022) which contains only 350M parameters. We set the layer set size to two and subsequently calculate ΔARRs for each set. As shown in Figure 7 (b) and (c), we observe the absence of transition layer in OPT, along with notable inconsistencies between RD and ΔARR values. Specifically in OPT-350M, the best resilience is achieved by closed-sourcing middle layers instead of the initial ones, suggesting that bottom layers may not offer better resilience. Therefore, SCARA fails to identify the smallest closed-sourced set in this case, suggesting its unsuitability for smaller models. Details are in Appendix C.9.

**Sensitivity of SCARA to $\varepsilon$.** We assess the sensitivity of SCARA to $\varepsilon$ by incrementally adjusting $\varepsilon$ from 0.05 to 1 in steps of 0.05, and calculate the ΔARR of five recovered models. As shown in Figure 8, we observe that SCARA exhibits low sensitivity to changes in $\varepsilon$. For instance, the ΔARRs stabilize across all models as $\varepsilon$ increases. This stability arises due to larger $\varepsilon$ values requiring smaller closed-sourced sets to satisfy $R(I) \leq (1+\varepsilon)R([L])$, thereby reducing the need for extensive layer closed-sourced. Details are in Appendix B.4.

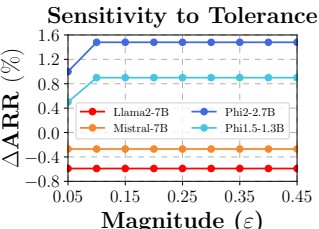

Figure 8: Sensitivity to $\varepsilon$.

**Limited Defense against Adversarial Attack.** We compare SCARA and SAP-DP in defending against three black-box adversarial attacks on Llama2-7B. Specifically, we apply the membership inference (Fu et al., 2023) (MIA), attribute inference (Staab et al., 2023) (AIA), and prompt injection (Liu et al., 2023) (PIA) attacks to the semi-open models produced by SAP-DP and SCARA. As shown in Table 5, we observe that SAP-DP outperforms SCARA across all three attacks, but still performs worse than the gold standard. This is because SCARA does not introduce additional output perturbation and thus provide limited defense against black-box adversarial attacks. Details can be found in Appendix B.11.

Table 5: Performance of SCARA defending adversarial attacks. ↓ indicates the smaller the better.

| Approach | MIA↓ | AIA↓ | PIA↓ |
|---|---|---|---|
| Gold Std. | 58.0 | 43.9 | 0.00 |
| SCARA | 72.3 | 85.0 | 26.5 |
| SAP-DP | 72.2 | 83.9 | 24.9 |

## 6 CONCLUSION

In this paper, we explored finding minimal closed-sourced sets to enhance LLM customizability while preserving their resilience against semi-open model recovery attacks. We theoretically prove that minor errors in bottom decoder layers prior to a transition layer greatly reduce recovery attack success. We introduced SCARA, which selectively closed-sources a small set of layers, achieving superior customizability and comparable resilience to SAP-DP and fully-closed. We empirically investigated the existence of customization and resilience transitions, showed the impact of closed-source size on model resilience, analyzed the effectiveness of our approach, and finally discussed its limitations.

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

## A  PROOF OF THEOREM 1

In this section, we prove Theorem 1. We first revisit the our model, present several important lemmas and finally present the proof.

### A.1  MODEL OVERVIEW

The recovered model $f(\mathbf{X}; \boldsymbol{\theta})$ is structured as a sequence of $L$ transformer layers,

$$f(\mathbf{X}) = \varphi_L \circ \varphi_{L-1} \circ ... \circ \varphi_{\alpha L+1} \circ \hat{\varphi}_{\alpha L} \circ_{\alpha L-1} \circ ... \circ \varphi_1(\mathbf{X}), \tag{3}$$

where $\mathbf{X} \in \mathbb{R}^{n \times d}$ represents the input, interpreted as an assembly of $n$ tokens, each possessing $d$ hidden dimensions. Each transformer layer, indexed by $1 \le i \le L$, is represented by $\varphi_i$, which maps $\mathbb{R}^{n \times d}$ to $\mathbb{R}^{n \times d}$ and can be defined as follows,

$$\varphi_i(\mathbf{X}; K_i, Q_i) = \left[\mathbf{I}_n + \text{softmax}\left(\frac{\mathbf{X}Q_i(\mathbf{X}K_i)^\top}{\sqrt{d_Q}\|\mathbf{X}\|^2}\right)\right]\mathbf{X}, \tag{4}$$

where $Q_i \in \mathbb{R}^{d \times d_Q}$, $K_i \in \mathbb{R}^{d \times d_Q}$ represent projection parameter matrices. Here, the $\alpha L$-th layer is the recovered layer and the others are the public layers. For simplicity, we use the function $\hat{\varphi}_{\alpha L}$ to denote mapping of the recovered layer, i.e., $\hat{\varphi}_{\alpha L}(\mathbf{X}) = \varphi_{\alpha L}(\mathbf{X}; \hat{K}_{\alpha L}, \hat{Q}_{\alpha L})$.

### A.2  BOUNDS ON DIFFERENT ORTHOGONAL COMPONENTS

**Lemma 1.** *For any $1 \le l \le L$, $1 \le p \le d$, any $\mathbf{X} \in \mathbb{R}^{n \times d}$, we have*

$$\max_{\boldsymbol{v}:\|\boldsymbol{v}\|_2=1, \boldsymbol{v}\perp\mathbb{I}_n} \left|\boldsymbol{v}^\top \varphi_l(\mathbf{X}; K_l, Q_l)[p]\right| \le (1 + \beta_D) \max_{\boldsymbol{v}:\|\boldsymbol{v}\|_2=1, \boldsymbol{v}\perp\mathbb{I}_n} \left|\boldsymbol{v}^\top \mathbf{X}[p]\right|, \tag{5}$$

*where $\mathbb{I}_n$ is a column vector with dimensions $n \times 1$ and each element is 1, $\mathbf{X}[p]$ is the p-th column of the input $\mathbf{X}$, $\varphi_l(\mathbf{X}; K_l, Q_l)[p]$ is the p-th column of the l-th self-attention output, the coefficient $\beta_D$ satisfies $0 < \beta_D < 1$ and it is related to the upper bound of the L2-norm of matrices $K_l, Q_l$.*

*Proof.* Let $\boldsymbol{u} = \left\{\boldsymbol{u}_{l,1} = \frac{\mathbb{I}_n}{\sqrt{n}}, \boldsymbol{u}_{l,2}, \ldots, \boldsymbol{u}_{l,n}\right\}$ denote the eigenvectors of $\text{softmax}\left(\frac{\mathbf{X}Q_l(\mathbf{X}K_l)^\top}{\sqrt{d_Q}\|\mathbf{X}\|^2}\right)$.

Assume $\sigma_{l,1}, \sigma_{l,2}, \ldots, \sigma_{l,n}$ denote the eigenvalues of $\text{softmax}\left(\frac{\mathbf{X}Q_i(\mathbf{X}K_i)^\top}{\sqrt{d_Q}\|\mathbf{X}\|^2}\right)$ and $-1 < \sigma_{l,n} < \beta_D$ for any $l, n$. Thus we have

$$\boldsymbol{v}^\top \varphi_l(\mathbf{X}; K_l, Q_l)[p] = \boldsymbol{v}^\top \left[\mathbf{I}_n + \text{softmax}\left(\frac{\mathbf{X}Q_l(\mathbf{X}K_l)^\top}{\sqrt{d_Q}\|\mathbf{X}\|^2}\right)\right]\mathbf{X}[p] \tag{6a}$$

$$= \boldsymbol{v}^\top \left[\mathbf{I}_n + \text{softmax}\left(\frac{\mathbf{X}Q_l(\mathbf{X}K_l)^\top}{\sqrt{d_Q}\|\mathbf{X}\|^2}\right)\right] \sum_{k=1}^{n} \alpha_{pk}\boldsymbol{u}_{l,k} \tag{6b}$$

$$= \boldsymbol{v}^\top \sum_{k=1}^{n} \alpha_{pk}(1 + \sigma_{l,k})\boldsymbol{u}_{l,k} \tag{6c}$$

$$\le \max_{\boldsymbol{v}:\|\boldsymbol{v}\|_2=1, \boldsymbol{v}\perp\mathbb{I}_n} \left|\sum_{k=2}^{n} \alpha_{pk}(1 + \sigma_{l,k})\boldsymbol{v}^\top u_{l,k}\right| \tag{6d}$$

$$= \left\|\sum_{k=2}^{n} \alpha_{pk}(1 + \sigma_{l,k})\boldsymbol{u}_{l,k}\right\|_2 \tag{6e}$$

$$= \left[\sum_{k=2}^{n} \alpha_{pk}^2(1 + \sigma_{l,k})^2\right]^{1/2} \tag{6f}$$

$$\le (1 + \beta_D) \max_{\boldsymbol{v}:\|\boldsymbol{v}\|_2=1, \boldsymbol{v}\perp\mathbb{I}_n} \left|\boldsymbol{v}^\top \mathbf{X}[p]\right|, \tag{6g}$$

where

$$\beta_D = \max_{\|K_l\|_2 \le D, \|Q_l\|_2 \le D} \max_{\boldsymbol{v}:\|\boldsymbol{v}\|_2=1, \boldsymbol{v} \perp \mathbb{I}_n} \left\| \mathrm{softmax}\left( \frac{\mathbf{X}Q_l(\mathbf{X}K_l)^\top}{\sqrt{d_Q}\|\mathbf{X}\|^2} \right) \boldsymbol{v} \right\|_2 < 1.$$

The equation equation 6c is due to $\boldsymbol{u}_{l,k}$ are the eigenvectors of $\mathrm{softmax}\left( \frac{\mathbf{X}Q_l(\mathbf{X}K_l)^\top}{\sqrt{d_Q}\|\mathbf{X}\|^2} \right)$. The inequality equation 6e is because when $\boldsymbol{v} = \frac{\sum_{k=2}^n \alpha_{pk}(1+\sigma_{l,k})\boldsymbol{u}_{l,k}}{\left\| \sum_{k=2}^n \alpha_{pk}(1+\sigma_{l,k})u_{l,k} \right\|_2}$, we have the maximum value.

$\square$

**Lemma 2.** *For any $K_l, Q_l \in \mathbb{R}^{d \times s}$ and any $\mathbf{X} \in \mathbb{R}^{n \times d}$, the following equation always holds:*

$$\left| \mathbb{I}_n^\top \varphi_i(\mathbf{X}; K_i, Q_i)[p] \right| = 2\left| \mathbb{I}_n^\top \mathbf{X}[p] \right|, \tag{7}$$

*where $\mathbf{X}[p]$ is the $p$-th column of the input $\mathbf{X}$, $\varphi_i(\mathbf{X}; K_i, Q_i)[p]$ is the $p$-th column of the $l$-th self-attention output.*

*Proof.* Assume that a set of orthogonal basis for $\mathbb{R}^n$ is $\{\boldsymbol{u_1}, \boldsymbol{u_2}, \ldots, \boldsymbol{u_n}\}$, where $\boldsymbol{u_1} = \frac{\mathbb{I}_n}{\sqrt{n}}$. Then we can rewrite $\mathbf{X}[p]$ as $\mathbf{X}[p] = \sum_{j=1}^n \alpha_{pj} \boldsymbol{u_j}$, where $\alpha_{pj}(1 \le p \le d)$ are the corresponding coefficients for the $p$-th column of $\mathbf{X}$ under the orthogonal basis. Next, we calculate $\left| \mathbb{I}_n^\top f(\mathbf{X})[p] \right|$ and $\left| \mathbb{I}_n^\top \mathbf{X}[p] \right|$, respectively. Note that $\mathbb{I}_n^\top \boldsymbol{u_j} = 0$ for all $j \ne 1$. Therefore, we can obtain that,

$$\mathbb{I}_n^\top \mathbf{X}[p] = \sqrt{n}\alpha_{p1}. \tag{8}$$

Then we can get

$$\left| \mathbb{I}_n^\top \mathbf{X}[p] \right| = |\sqrt{n}\alpha_{p1}|. \tag{9}$$

Let $\sigma_{i1}, \sigma_{i2}, \ldots, \sigma_{in}$ denote the eigenvalues of $\mathrm{softmax}\left( \frac{\mathbf{X}Q_i(\mathbf{X}K_i)^\top}{\sqrt{d_Q}\|\mathbf{X}\|^2} \right)$. Applying the Perron–Frobenius theorem for Markov matrices Lemmens & Nussbaum (2012), we deduce that for the matrix $\mathrm{softmax}\left( \frac{\mathbf{X}Q_l(\mathbf{X}K_i)^\top}{\sqrt{d_Q}\|\mathbf{X}\|^2} \right)$, there exists only one eigenvalue equal to 1, while all other eigenvalues in absolute value are strictly less than 1. Without loss of generality, we assume $\sigma_{i1} = 1$, implying $|\sigma_{ij}| < 1$ for $j \ne 1$. Recalling the definition of $\varphi_i(\mathbf{X}; K_i, Q_i)$ and considering the linear operation, we can rewrite it as follows:

$$\varphi_i(\mathbf{X}; K_i, Q_i)[p] = \sum_{j=1}^n \alpha_{pj}(1 + \sigma_{ij})\boldsymbol{u_j}. \tag{10}$$

Then we calculate the term $\left| \mathbb{I}_n^\top \varphi_i(\mathbf{X}; K_i, Q_i)[p] \right|$ as follows,

$$\left| \mathbb{I}_n^\top \varphi_i(\mathbf{X}; K_i, Q_i)[p] \right| = \left| \mathbb{I}_n^\top (\sum_{j=1}^n \alpha_{pj}(1 + \sigma_{ij})\boldsymbol{u_j} \right| \tag{11a}$$

$$= \left| \sqrt{n}(\alpha_{p1}(1 + \sigma_{i1})) \right| \tag{11b}$$

$$= 2|\sqrt{n}\alpha_{p1}|, \tag{11c}$$

where equation 11a is induced by substituting the equation equation 10 into $\left| \mathbb{I}_n^\top \varphi_i(\mathbf{X}; K_i, Q_i)[p] \right|$, equation 11b is due to $\mathbb{I}_n^\top \boldsymbol{u_j} = 0$ for all $j \ne 1$, equation 11c follows the fact that $\sigma_{i1} = 1$.

$\square$

## A.3 PROOF OF THEOREM 1

We first prove the following result. For simplicity of notations, we use $f(\mathbf{X})[p]$ to denote the $p$-th $(1 \le p \le d)$ column of the the recovered model $f(\mathbf{X})$, where the parameters in the $\alpha L$-th layer is replaced with the matrices $\hat{K}_{\alpha L}$ and $\hat{Q}_{\alpha L}$. We use the function $\hat{\varphi}_{\alpha L}(\mathbf{X}) = \varphi_{\alpha L}(\mathbf{X}; \hat{K}_{\alpha L}, \hat{Q}_{\alpha L})$ to

denote the mapping of the $(\alpha L)$-th layer. Then we are going to show that there exists $\alpha^\star = \log_2 \frac{2}{1+\beta_D}$ and $0 < \beta_D < 1$ makes the following equations hold.

(1) Assume $\alpha < \alpha^\star$. For any $\mathbf{X}$, $\|K_i\|_2 \le D$, $\|Q_i\|_2 \le D$, there exists a zero measure set $\mathcal{K}(\mathbf{X})$ and $\mathcal{Q}(\mathbf{X})$ such that

$$\lim_{L \to \infty} \left\| \frac{f(\mathbf{X})[p]}{\|f(\mathbf{X})[p]\|_2} - \frac{\mathbb{I}_n}{\sqrt{n}} \right\|_2 = 0. \tag{12}$$

(2) For any $\alpha > \alpha^\star$, there exists a sequence of matrix $\{K_i, Q_i\}_{i \ge 1}$ such that for any recovered matrix $K_{\alpha L}$ and $Q_{\alpha L}$, we have $\|K_i\|_2 \le D$, $\|Q_i\|_2 \le D$, we have,

$$\lim_{L \to \infty} \left\| \frac{f(\mathbf{X})[p]}{\|f(\mathbf{X})[p]\|_2} - \frac{\mathbb{I}_n}{\sqrt{n}} \right\|_2 = \sqrt{2}. \tag{13}$$

*Proof.* Based on Lemma equation 1, we obtain that

$$\max_{\boldsymbol{v}: \|\boldsymbol{v}\|_2 = 1, \boldsymbol{v} \perp \mathbb{I}_n} \left| \boldsymbol{v}^\top f(\mathbf{X})[p] \right| \le (1+\beta)^L \max_{\boldsymbol{v}: \|\boldsymbol{v}\|_2 = 1, \boldsymbol{v} \perp \mathbb{I}_n} \left| \boldsymbol{v}^\top \mathbf{X}[p] \right|. \tag{14}$$

Based on Lemma equation 2, we know that

$$\left| \mathbb{I}_n^\top f(\mathbf{X})[p] \right| = 2^{(1-\alpha)L-1} \left| \mathbb{I}_n^\top \hat{\varphi}_{\alpha L} \circ \varphi_{\alpha L-1} \circ \cdots \circ \varphi_1(\mathbf{X})[p] \right|. \tag{15}$$

We firstly prove the equation equation 12. When

$$\left| \mathbb{I}_n^\top f(\mathbf{X})[p] \right| = 2^{(1-\alpha)L-1} \left| \mathbb{I}_n^\top \hat{\varphi}_{\alpha L} \circ \varphi_{\alpha L-1} \circ \cdots \circ \varphi_1(\mathbf{X})[p] \right| \ne 0, \tag{16}$$

then we have

$$\left\| \frac{f(\mathbf{X})[p]}{\|f(\mathbf{X})[p]\|_2} - \frac{\mathbb{I}_n}{\sqrt{n}} \right\|_2 = \left[ 2 - \frac{2 \mathbb{I}_n^\top f(\mathbf{X})[p]}{\sqrt{n} \sqrt{\frac{(\mathbb{I}_n^\top f(\mathbf{X})[p])^2}{n} + (\boldsymbol{v}^\top f(\mathbf{X})[p])^2}}} \right]^{1/2} \tag{17a}$$

$$= \sqrt{2} \left[ 1 - \frac{1}{\sqrt{1 + \frac{n(\boldsymbol{v}^\top f(\mathbf{X})[p])^2}{(\mathbb{I}_n^\top f(\mathbf{X})[p])^2}}}} \right]^{1/2} \tag{17b}$$

$$\le \sqrt{2} \left[ 1 - \frac{1}{\sqrt{1 + \frac{n(1+\beta)^{2L} |\boldsymbol{v}^\top \mathbf{X}[p]|^2}{2^{2[(1-\alpha)L-1]} |\mathbb{I}_n^\top \hat{\varphi}_{\alpha L} \circ \varphi_{\alpha L-1} \circ \cdots \circ \varphi_1(\mathbf{X})[p]|^2}}}} \right]^{1/2} \tag{17c}$$

$$\le 2\sqrt{2n} \left( \frac{1+\beta}{2^{1-\alpha}} \right)^L \frac{|\boldsymbol{v}^\top \mathbf{X}[p]|}{|\mathbb{I}_n^\top \hat{\varphi}_{\alpha L} \circ \varphi_{\alpha L-1} \circ \cdots \circ \varphi_1(\mathbf{X})[p]|}, \tag{17d}$$

where the inequality equation 17c is based on the inequality equation 14 and equation 15. The inequality equation 17d is based on Lemma equation 3. Therefore, if $\alpha < \log_2 \frac{2}{1+\beta_D}$ and $\left| \mathbb{I}_n^\top f(\mathbf{X})[p] \right| \ne 0$, then we have $\lim_{L \to \infty} \left( \frac{1+\beta_D}{2^{1-\alpha}} \right)^L = 0$. Now we can consider when $\left| \mathbb{I}_n^\top f(\mathbf{X})[p] \right| = 0$. In fact, it is easy to show that this can only happens when $\hat{K}_{\alpha L}$ and $\hat{Q}_{\alpha L}$ belong to certain sets making $\left| \mathbb{I}_n^\top f(\mathbf{X})[p] \right| = 0$, which corresponds to zero measure set $\mathcal{K}(\mathbf{X})$ and $\mathcal{Q}(\mathbf{X})$ depending on the input $\mathbf{X}$. Since the input space is countable, therefore, the union $\cup_{\mathbf{X} \in \mathcal{X}} \mathcal{K}(\mathbf{X})$ and $\cup_{\mathbf{X} \in \mathcal{X}} \mathcal{Q}(\mathbf{X})$ are also zero-measure sets.

To prove equation equation 13, let $K^\star$, $Q^\star$ with $\|K^\star\|_2 \le D$, $\|Q^\star\|_2 \le D$ satisfy the following condition,

$$\max_{\boldsymbol{v}: \|\boldsymbol{v}\|_2 = 1, \boldsymbol{v} \perp \mathbb{I}_n} \left\| \text{softmax} \left( \frac{\mathbf{X} Q_l (\mathbf{X} K_l)^\top}{\sqrt{d_Q} \|\mathbf{X}\|^2} \right) \boldsymbol{v} \right\|_2 = \beta_D. \tag{18}$$

Let $\boldsymbol{v}^\star$ be the solver of the above optimization problem equation 18 and consider the $K_l = K^\star$, $Q_l = Q^\star$ and $\mathbf{X}^\star = [\boldsymbol{v}^\star, \boldsymbol{v}^\star, \cdots, \boldsymbol{v}^\star]$. Clearly, $\boldsymbol{v}^\star \perp \mathbb{I}_n$. Assume there exists $\boldsymbol{u} : \|\boldsymbol{u}^\star\|_2 = 1$ satisfying $\boldsymbol{u}^\star \perp \mathbb{I}_n$, $\boldsymbol{u}^\star \perp \boldsymbol{v}^\star$, therefore we can rewrite $f(\mathbf{X}^\star)[p]$ as follows,

$$f(\mathbf{X}^\star)[p] = \frac{\mathbb{I}_n^\top}{\sqrt{n}} f(\mathbf{X}^\star) \frac{\mathbb{I}_n}{\sqrt{n}} + \boldsymbol{v}^{\star\top} f(\mathbf{X}^\star) \boldsymbol{v}^\star + \boldsymbol{u}^{\star\top} f(\mathbf{X}^\star) \boldsymbol{u}^\star. \tag{19}$$

For any $1 \leq l \leq L$, based on Lemma equation 1, we know that

$$\left| \boldsymbol{v}^{*\top} f\left(\mathbf{X}^{\star}\right)[p] \right| = (1 + \beta_D)^L \left| \boldsymbol{v}^{*\top} \mathbf{X}^{\star}[p] \right|. \tag{20}$$

Since

$$\left| \mathbb{I}_n^{\top} f\left(\mathbf{X}^{\star}\right)[p] \right| = 2^L \left| \mathbb{I}_n^{\top} \mathbf{X}^{\star}[p] \right| = \left| \mathbb{I}_n^{\top} \boldsymbol{v}^{\star} \right| = 0 \tag{21}$$

and

$$\left| \boldsymbol{v}^{*\top} f\left(\mathbf{X}^{\star}\right)[p] \right| = (1 + \beta_D)^L \left| \boldsymbol{v}^{*\top} \mathbf{X}^{\star}[p] \right| \neq 0, \tag{22}$$

then we have

$$\left\| \frac{f(\mathbf{X}^{\star})[p]}{\|f(\mathbf{X}^{\star})[p]\|_2} - \frac{\mathbb{I}_n}{\sqrt{n}} \right\|_2 = \left[ 2 - \frac{2\mathbb{I}_n^{\top} f(\mathbf{X}^{\star})[p]}{\sqrt{n}\,\|f(\mathbf{X}^{\star})[p]\|_2} \right]^{1/2} \tag{23a}$$

$$= \left[ 2 - \frac{2\mathbb{I}_n^{\top}}{\sqrt{n}} \frac{f(\mathbf{X}^{\star})[p]}{\sqrt{\frac{1}{n}(\mathbb{I}_n^{\top} f(\mathbf{X}^{\star})[p])^2 + (\boldsymbol{v}^{\star\top} f(\mathbf{X}^{\star})[p])^2 + (\boldsymbol{u}^{\star\top} f(\mathbf{X}^{\star})[p])^2}} \right]^{1/2} \tag{23b}$$

$$\geq \left[ 2 - \frac{2\mathbb{I}_n^{\top}}{\sqrt{n}} \frac{f(\mathbf{X}^{\star})[p]}{\sqrt{\frac{1}{n}(\mathbb{I}_n^{\top} f(\mathbf{X}^{\star})[p])^2 + (\boldsymbol{v}^{\star\top} f(\mathbf{X}^{\star})[p])^2}} \right]^{1/2} \tag{23c}$$

$$= \left[ 2 - 2 \frac{\frac{\mathbb{I}_n^{\top} f(\mathbf{X}^{\star})[p]}{\sqrt{n}|\boldsymbol{v}^{\star\top} f(\mathbf{X}^{\star})[p]|}}{\sqrt{1 + \frac{|\mathbb{I}_n^{\top} f(\mathbf{X}^{\star})[p]|^2}{n|\boldsymbol{v}^{\star\top} f(\mathbf{X}^{\star})[p])|^2}}} \right]^{1/2} \tag{23d}$$

$$= \left[ 2 - 2 \frac{\frac{2^{(1-\alpha)L-1}|\mathbb{I}_n^{\top} \hat{\varphi}_{\alpha L} \circ \varphi_{\alpha L-1} \circ \cdots \circ \varphi_1(\mathbf{X}^{\star})[p]|}{\sqrt{n}(1+\beta_D)^L|\boldsymbol{v}^{\star\top} \mathbf{X}^{\star}[p]|}}{\sqrt{1 + \frac{2^{2[(1-\alpha)L-1]}}{n(1+\beta_D)^{2L}} \frac{|\mathbb{I}_n^{\top} \hat{\varphi}_{\alpha L} \circ \varphi_{\alpha L-1} \circ \cdots \circ \varphi_1(\mathbf{X}^{\star})[p]|^2}{|\boldsymbol{v}^{\star\top} \mathbf{X}^{\star}[p]|^2}}} \right]^{1/2}, \tag{23e}$$

where equation equation 23b is based on equation 19, equation equation 23e is based on equation 22 and equation 15. When $\alpha > \log_2 \frac{2}{1+\beta_D}$, we have $\lim_{L\to\infty} \left( \frac{2^{1-\alpha}}{1+\beta_D} \right)^L = 0$. Thus we have $\lim_{L\to\infty} \left\| \frac{f(\mathbf{X}^{\star})[p]}{\|f(\mathbf{X}^{\star}[p]\|_2} - \frac{\mathbb{I}_n}{\sqrt{n}} \right\|_2 = \sqrt{2}$. This indicates that the $p$-th column of the output matrix $f(\mathbf{X}^{\star})$ is not parallel to $\mathbf{I}_n$ for any $p$. This further indicates that the output matrix does not have the identical vector in each row. $\qquad\square$

## A.4   Technical Lemma

**Lemma 3.** *For any $x \in (0, 1)$, it always holds $\left[ 1 - \frac{1}{\sqrt{1+x^2}} \right]^{1/2} \leq x$.*

*Proof.* To establish the inequality $\left[ 1 - \frac{1}{\sqrt{1+x^2}} \right]^{1/2} \leq x$, we begin by proving,

$$1 - \frac{1}{\sqrt{1 + x^2}} \leq x^2. \tag{24}$$

To demonstrate equation 24, we equivalently show

$$1 - x^2 \leq \frac{1}{\sqrt{1 + x^2}}. \tag{25}$$

Subsequently, it suffices to verify

$$(1 - x^2)(\sqrt{1 + x^2}) \leq 1. \tag{26}$$

This is equivalent to proving

$$(1 - x^2)^2 (1 + x^2) \leq 1. \tag{27}$$

Thus, our focus shifts to demonstrating

$$(1 - x^2)(1 - x^4) \leq 1. \tag{28}$$

Clearly, equation 28 holds true for any $x \in (0, 1)$. $\qquad\square$

# B Experiment Details

## B.1 Model Details.

The foundation models we use in our experiments are selected from open-source repositories, and Table 6 shows the basic information of the models and their sources. Specifically, we employ Llama2-70B-chat[2], Llama2-7B-chat[3], and Mistral-7B-v0.1[4]. For smaller models, we select Phi-2[5] and Phi-1.5[6]. We also consider OPT model[7], which has only 350 million parameters and 24 decoder layers.

Table 6: Model Info

| Model | Size | Decoder Layers |
|---|---|---|
| Llama2-70B-chat (Touvron et al., 2023) | 70B | 80 |
| Llama2-7B-chat (Touvron et al., 2023) | 7B | 32 |
| Mistral-7B-v0.1 (Jiang et al., 2023a) | 7B | 32 |
| Phi-2 (Abdin et al., 2024) | 2.7B | 32 |
| Phi-1.5 (Li et al., 2023) | 1.3B | 24 |
| OPT (Zhang et al., 2022) | 350M | 24 |

## B.2 Recovery Attacks.

**Attack implementation details.** In performing FT-all and FT-closed model recovery attacks, we adhere to the training hyper-parameters outlined in the Llama2 report (Touvron et al., 2023), employing the AdamW optimizer with a cosine learning rate scheduler. The initial learning rate is set to $2 \times 10^{-5}$, with a weight decay of 0.1, a batch size of 128, and bfloat16 precision for input sequences of 512 tokens. The LLaMA2-70B model is trained for 3 epochs with a random seed of 42, while other models are trained for 5 epochs across three seeds: 42, 1234, and 20. Despite limiting training to 3 epochs for the 70B model, the training loss stabilized effectively. Our implementation builds upon the llama-recipes repository provided by META.

For SEM attacks, distinct configurations were employed for SCARA and SAP-DP. In the case of SCARA, hidden representations from the closed-source components were collected and paired with the input data to train a substitute model. In contrast, for SAP-DP, representations from the sixth decoder layer and the model's final logits were utilized to construct the training dataset. In accordance with (Tamber et al., 2024), we applied a learning rate of 1.5e-4, a weight decay of 0.01, and a linear learning rate scheduler with 500 warmup steps. Both training and validation batch sizes were set to 32, with MSE as the loss function. SCARA was trained for 30 epochs due to its smaller model size, whereas SAP-DP was trained for 5 epochs.

All recovery experiments were conducted on Nvidia 4090 24G, 6000 Ada 48G, and A100 80G GPUs, utilizing PyTorch 2.2.0 and CUDA 11.8 on Ubuntu 20.04.6 LTS.

**Base 51k Recovery Dataset.** We ensure dataset coverage and reliability by using a 1:1 ratio of the MMLU auxiliary training set [8] and Alpaca dataset [9], extracting 25.5k samples from each. From the MMLU auxiliary training data (Hendrycks et al., 2021), we sample 50%, and from Alpaca (Taori et al., 2023), we use a step size of 2 to enhance diversity. The datasets are then formatted for model training, applying Alpaca and MMLU prompts from Table 7.

---

[2] https://huggingface.co/meta-llama/Llama-2-70b-chat-hf
[3] https://huggingface.co/meta-llama/Llama-2-7b-chat-hf
[4] https://huggingface.co/mistralai/Mistral-7B-v0.1
[5] https://huggingface.co/microsoft/phi-2
[6] https://huggingface.co/microsoft/phi-1_5
[7] https://huggingface.co/facebook/opt-350m
[8] https://github.com/hendrycks/test
[9] https://github.com/tatsu-lab/stanford_alpaca/blob/main/alpaca_data.json

Table 7: Prompts for Alpaca and MMLU auxiliary training data

| Dataset | Prompt Type | Description |
|---|---|---|
| **Alpaca** | with input | Below is an instruction that describes a task, paired with an input that provides further context. Write a response that appropriately completes the request. |
| | w/o input | Below is an instruction that describes a task. Write a response that appropriately completes the request. |
| **MMLU** | Question Answering | Below is a question with no choices. Write the correct answer that appropriately solves the question. |
| | Multiple Choice | The following is a multiple choice question, paired with choices. Answer the question in the format: "Choice:content". |

**Extra Recovery Datasets.** To enhance dataset diversity, the 100K, 200K, 300K, and 500K datasets integrate additional specialized sources. As detailed in Table 8, these sources include Baize (Xu et al., 2023) (158K English multi-turn conversations via ChatGPT's self-chat), MathInstruct (Yue et al., 2023) (260K curated math instruction instances focusing on hybrid reasoning), and OpenOrca (Mukherjee et al., 2023) (augmented FLAN collection with 1M GPT-4 completions and 3.2M GPT-3.5 completions). These enrichments are intended to support complex computational and theoretical tasks, offering broader topic coverage.

Table 8: Composition of variously sized datasets

| Raw Data Set | 51k | 100k | 200k | 300k | 500k |
|---|---|---|---|---|---|
| Alpaca | 25.5 | 50 | 40 | 50 | 50 |
| MMLU auxiliary training set | 25.5 | 50 | 40 | 100 | 100 |
| Baize-MedQuAD | 0 | 0 | 40 | 50 | 50 |
| Baize-Quora | 0 | 0 | 40 | 50 | 50 |
| Baize-Stackoverflow | 0 | 0 | 40 | 50 | 50 |
| MathInstruct | 0 | 0 | 4 | 6 | 20 |
| OpenOrca | 0 | 0 | 0 | 0 | 180 |

**Validation Datasets.** Table 9 outlines the composition of the validation datasets. For *Validation Dataset 1*, we extracted 50% from each of the 57 MMLU validation sub-datasets, totaling 1.5K instances, paired with Alpaca data selected using a step size of 751. This dataset is used with the 51K and 100K training sets. For larger training sets (200K, 300K, and 500K), *Validation Dataset 2* was created by adding 400 instances from three Baize subsets, expanding the validation set to 4.0K.

Table 9: Composition of validation datasets of different sizes

| Raw Data Set | Validation Set | Evaluation Set |
|---|---|---|
| Alpaca | 765 | 765 |
| MMLU auxiliary training set | 751 | 751 |
| Baize-MedQuAD | 0 | 850 |
| Baize-Quora | 0 | 850 |
| Baize-Stackoverflow | 0 | 850 |
| **Total Length** | 1516 | 4066 |

### B.3 BASELINES.

In this section, we provide further details on the baselines used in our comparisons: SAP-DP and fully-closed. These schemes represent different strategies, each with distinct trade-offs in terms of customizability and resilience against model recovery attacks.

**SAP.** The Split-and-Privatize (SAP) framework (Shen et al., 2023) offers an approach to balance between protecting model privacy and data privacy while maintaining competitive performance. Specifically, the SAP framework keeps the bottom six encoder layers open, allowing user access and fine-tuning while closing the deeper layers on the vendor.

**SAP-DP.** To further strengthen protection while maintaining competitive performance, we extend SAP by incorporating differential privacy techniques by adding Laplace noise to perturb the logits during the fine-tuning process (Lee et al., 2018). The Laplace Distribution with mean $\mu$ and scale $b$ is the distribution with probability density function:

$$\text{Laplace}(x|\mu, b) = \frac{1}{2b} \exp\left(-\frac{|x - \mu|}{b}\right)$$

Specifically, in SAP-DP, the noise $n$ is sampled: $n \sim \text{Laplace}(0, 0.5)$ and added to the output logits of the model to balance privacy protection and model performance.

**Fully-closed.** Following (Eiras et al., 2024), we use the fully-closed approach as a baseline. This assumes the adversary has no access to internal model parameters, treating the model as a black-box, where only output data can be collected. We slightly broaden this setup by assuming the adversary knows the model's architecture but no other details. Thus, recovering the fully-closed model involves using the collected data to retrain a model with the same architecture to restore its general functionality.

### B.4 IMPLEMENTATION DETAILS OF SCARA.

**Evaluation Datasets.** We created a 1.5K Evaluation Set to assess model resilience under various closed-sourcing strategies. This set includes 50% of entries from each of the 57 MMLU validation sub-datasets (Hendrycks et al., 2021), distinct from Validation Set outlined in Table 9. Additionally, we selected an equal number of Alpaca dataset (Taori et al., 2023), using a step size of 751, ensuring no overlap with the Validation Set.

**Hyper-parameter Sensitivity.** As shown in Figure 9, we evaluate SCARA's sensitivity to tolerance magnitude $\varepsilon$, adjusting it from 0.05 to 1 in 0.05 increments while calculating the $\Delta$ARR for six recovered models. The results indicate that SCARA is minimally sensitive to changes in $\varepsilon$, with $\Delta$ARR values stabilizing as $\varepsilon$ increases. This stability arises from the need for a smaller closed-sourced layer at higher $\varepsilon$, allowing the condition $R(I) \leq (1 + \varepsilon)R([L])$ to be met with fewer layers. Additionally, the increase in $\Delta$ARR is smaller for larger models, suggesting that privatizing more parameters beyond a certain point offers diminishing returns in resilience.

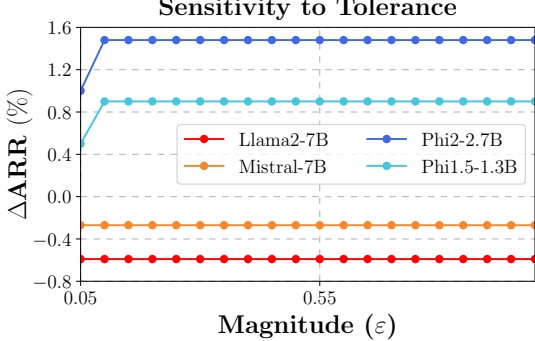

Figure 9: Sensitivity on $\varepsilon$.

### B.5 EVALUATION BENCHMARKS

Most of our evaluations are conducted using the lm-evaluation suite (Gao et al., 2023), the bigcode-evaluation-harness platform (Ben Allal et al., 2022), and MT-Bench (Zheng et al., 2023). For specific domains, such as finance and law, we utilize the official benchmark testing codes provided by their respective communities, as detailed below.

**Evaluation on Customizabilities.** We assess the customizability of models across six domains, as detailed in Table 10. Each domain includes specific benchmarks and metrics designed to evaluate different aspects of the model's performance in relation to customizability. In particular, for evaluating medical capabilities, we select two subcategories from the MMLU benchmark that are related to the medical domain: *mmlu_anatomy* and *mmlu_professional_medicine*. For assessing legal reasoning, we select 10 multiple-choice and judgment-based subcategories from Legalbench. The performance of the model in these legal tasks is measured using perplexity, following the prompt structure provided by Legalbench. Specifically, the selected subcategories include: *cuad_audit_rights*, *canada_tax_court_outcomes*, *definition_classification*, *cuad_affiliate_license-licensee*, *learned_hands_business*, *contract_nli_survival_of_obligations*, *contract_nli_explicit_identification*, *contract_nli_confidentiality_of_agreement*, *hearsay*, and *contract_qa*.

Table 10: Details of the Six Customizability Benchmarks

| Domain | Benchmark | Metric | n-shot | Reference |
|---|---|---|---|---|
| Code | HumanEval | Pass@1 | 0 | Chen et al. (2021) |
|  | MBPP | Pass@1 | 1 | Austin et al. (2021) |
| Math | GSM8K | Exact Match | 8 | Cobbe et al. (2021) |
| Medical | MMLU_Medical | Accuracy | 5 | Hendrycks et al. (2021) |
| Finance | FPB | F1 | 0 | Wang et al. (2023a) |
| Law | LegalBench | Accuracy | 0 | Guha et al. (2023) |
| Alignment | MT-Bench | Score | (GPT-4) | Zheng et al. (2023) |

**Evaluation on Resilience.** We follow the Llama-2 report Touvron et al. (2023) to evaluate the recovered model, including 16 benchmarks, which are categorized into 6 groups. Table 11 summarizes the functionality benchmarks used in our experiments, along with their test methods and performance metrics. Our model ranks choices in multiple-choice tasks and generates answers for open-ended generation tasks.

### B.6 MODEL CUSTOMIZATION

**Datasets.** To fine-tune the models for domain-specific tasks, we utilized several datasets tailored to different sectors, including Code (Zheng et al., 2024b), Math (Yue et al., 2023), Medical (Zhang et al., 2023), Finance (Wang et al., 2023b), Law (Guha et al., 2024), and Alignment (Meng et al., 2024). Table 12 lists the customization training datasets used in the experiments. For the code domain, we combine the datasets from CodeFeedback and CodeAlpaca. For law and finance, we merge all training datasets from Legalbench and FinGPT respectively. These datasets are then prepared for model training using the Alpaca prompts outlined in Table 7. Additionally, we randomly select 3,000 samples to serve as the validation dataset.

**Customization Training Hyperparameters.** In model customization, we use different hyperparameters depending on the model size. For LLaMA2-70B, we apply QLoRA with the settings outlined in Table 13, while for 7B models, we use LoRA. For smaller models like Phi2 and Phi-1.5, we fine-tune all model parameters. For LLaMA2-70B, we fine-tune it as a quantized 4-bit model over 1 epoch, starting with a learning rate of $1.5 \times 10^{-6}$. For the 7B models, we train for 3 epochs, with a seed value of 42. The training setup includes a weight decay of 0.1, a batch size of 128, a warmup ratio of 0.03, and input sequences of 512 tokens, following standard experimental practices (Hu et al., 2021).

Table 11: Details of the Sixteen Functionality Benchmarks

| Domain | Benchmark | Metric | n-shot | Reference |
|--------|-----------|--------|--------|-----------|
| **Commonsense Reasoning** | PIQA | Accuracy | 0 | Bisk et al. (2020) |
| | Hellaswag | Accuracy | 0 | Zellers et al. (2019) |
| | Winogrande | Accuracy | 0 | Sakaguchi et al. (2019) |
| | ARC_easy | Accuracy | 0 | Clark et al. (2018) |
| | ARC_challenge | Accuracy | 0 | Clark et al. (2018) |
| **Reading Comprehension** | OpenBookQ | Accuracy | 0 | Mihaylov et al. (2018) |
| | LAMBADA | Accuracy | 0 | Paperno et al. (2016) |
| | BoolQ | Accuracy | 0 | Clark et al. (2019) |
| | SQuADv2 | HasAns_EM | 2 | Rajpurkar et al. (2018) |
| | SQuADv2 | HasAns_F1 | 2 | Rajpurkar et al. (2018) |
| **World Knowledge** | NaturalQuestions | Exact Match | 5 | Kwiatkowski et al. (2019) |
| | TriviaQA | Exact Match | 5 | Joshi et al. (2017) |
| **Code** | HumanEval | Pass@1 | 0 | Chen et al. (2021) |
| | MBPP | Pass@1 | 1 | Austin et al. (2021) |
| **Math** | GSM8K | Exact Match | 8 | Cobbe et al. (2021) |
| **General Ability** | MMLU | Accuracy | 5 | Hendrycks et al. (2021) |
| | BBH | Accuracy | 3 | Suzgun et al. (2022) |

Table 12: Customization Training Datasets Composition

| Domain | Dataset Name | Size | Reference |
|--------|-------------|------|-----------|
| **Code** | CodeFeedback | 156k | Zheng et al. (2024a) |
| | CodeAlpaca | 20k | Chaudhary (2023) |
| **Math** | MathInstruction | 262K | Yue et al. (2023) |
| **Medical** | MedMCQA | 183k | Zhang et al. (2023) |
| **Law** | Legalbench | 90k | Guha et al. (2023) |
| **Finance** | FinGPT | 204k | Wang et al. (2023a) |
| **Alignment** | Ultrafeedback | 62k | Cui et al. (2024) |

For Phi2 and Phi-1.5, we use the training hyperparameters from the LLaMA2 report. We employ the AdamW optimizer with a cosine learning rate scheduler, starting with a learning rate of $2 \times 10^{-5}$, a weight decay of 0.1, a batch size of 128, and use bfloat16 precision for 512-token input sequences. Specifically, for alignment, we follow SimPO Meng et al. (2024) and set the preference parameters $\beta = 2$ and $\gamma = 1$. The learning rate is $1 \times 10^{-6}$ for LLaMA2-70B and $5 \times 10^{-7}$ for the 7B and smaller models. All experiments are conducted using the LLaMA-Factory on Nvidia 4090 24G, 6000 Ada 48G, and A100 80G GPUs, with PyTorch 2.2.0 and CUDA 11.8 on Ubuntu 20.04.6 LTS.

### B.7 AMPLIFICATION OF SMALL RECOVERY ERROR

To investigate the amplification of minor recovery errors in the pre-transition layers, we conducted experiments on the LLaMA2-7B model. Specifically, we added Gaussian noise, $x \sim N(0, 0.01^2)$, to the parameters of the first decoder layer of the model. Then, we compared the representation outputs generated by the noisy model with those from the original model. The difference between the two

Table 13: The Hyperparameters for Customization Training.

| Model | Method | Rank $r$ | Lora $\alpha$ | Dropout | Learning Rate | Epochs | Warmup R. |
|---|---|---|---|---|---|---|---|
| **Llama2-70B** | QLoRA | 96 | 16 | 0.05 | 1.50E-04 | 1 | 0.03 |
| **Llama2-7B** | LoRA | 32 | 64 | 0.05 | 2.00E-05 | 3 | 0.03 |
| **Mistral-7B** | LoRA | 32 | 64 | 0.05 | 1.00E-06 | 3 | 0.03 |

sets of outputs was measured using the Frobenius norm. We collected representation outputs from layers 0, 3, 6, 9, 12, 15, 18, 21, 24, 27, and 30, totaling 45,000 output samples. The results showed a significant amplification of these small errors.

### B.8 RESILIENCE AND CUSTOMIZATION TRANSITIONS

For the LLaMA2-7B model, the smallest closed-source layer set identified by SCARA consists of a single decoder layer, whereas for Phi-2, it includes two decoder layers. Consequently, for LLaMA2-7B, we opted to closed-source each even-indexed layer, while for Phi-2, we chose to closed-source non-overlapping pairs of layers (e.g., layers 0-1, 2-3). For each selected layer set, we first closed-source them, then subjected the semi-open model to FT-all attacks, and subsequently calculated the $\Delta$ARR of the layer set to assess its resilience.

When verifying the customization transition, due to computational constraints, we validated only every other layer set for both models (e.g., closed-source layers 0, 0-4, 0-8 ...). Specifically, we applied LoRA-based customization on LLaMA2-7B in the math domain, while for Phi-2, we utilized the full finetuning approach. The experimental hyperparameters remain consistent with those outlined in the Appendix B.6.

We further computed the $\Delta$ARR for each closed-source set within Mistral-7B-v0.1 and Phi-1.5. In these models, the smallest closed-source set identified by SCARA consists of one decoder layer and two decoder layers, respectively. Following the same experimental configuration as LLaMA2-7B and Phi-2, we closed-sourced each even-indexed layer for Mistral-7B, and non-overlapping pairs of layers for Phi-1.5. The complete results demonstrating the transition layers within the Mistral-7B and Phi-1.5 model that closed two non-overlapping consecutive layers are depicted in Figure 10. Once again, we observed a distinct presence of transition layers. Specifically, in Mistral-7B, the transition layer appears at the 24th layer, while in Phi-1.5, it is located within the first layer set. Further results for can be found in Appendix C.7.

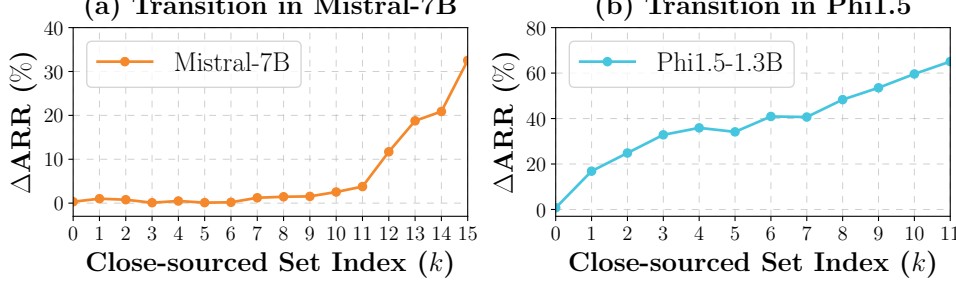

Figure 10: Resilience changes in Miatral-7B and Phi-1.5.

### B.9 RESILIENCE ACROSS CLOSED SIZES

To examine the influence of Closed layer size on model resilience, we conduct experiments on Closed-sourcing different amounts and proportions of parameters in the model's decoder layer. We give instructions on the detailed setting of closed-sourced models in Table 14. The module names are all derived from the overall implementation functions of each model in the Transformers open-source repositories in Table 6. We utilize abbreviated module names to denote specific settings.

We further computed $\Delta$ARR by close-sourcing varying quantities and proportions of parameters under FT-all attacks on three additional models. As shown in Figure 11 and Figure 6(b), we observed

Table 14: Closed-sourced Sizes Setting. "*" indicates an entire decoder layer.

| | | Llama-7B | Mistral-7B | Phi2-2.7B | Phi1.5-1.3B |
|---|---|---|---|---|---|
| **Proportion** | 0.25% | $W_k$ | $W_q, W_k$ | $W_k$ | $W_k$ |
| | 0.50% | $W_q, W_k$ | $W_o, MLP_{up}$ | $W_q, W_k$ | $W_q, W_k$ |
| | 1% | $W_q, W_k, W_v, W_o$ | $W_q, W_k, W_v, W_o$ | $W_q, W_k, W_v, W_d$ | $W_q, W_k, , W_v$ |
| | 3% | 0 | 0 | 0 | 0 |
| | 7% | 0-1 | 0-1 | 0-1 | 0-1 |
| | 15% | 0-4 | 0-4 | 0-3 | 0-3 |
| | 30% | 0-9 | 0-9 | 0-9 | 0-6 |
| | 50% | 0-15 | 0-15 | 0-15, $W_{em}$ | 0-11, $W_{em}$ |
| | 100% | Fully-closed | Fully-closed | Fully-closed | Fully-closed |
| **Quantity** | 20M | $W_k$ | $W_q, W_k$ | $W_q, W_k, W_v$ | $W_q, W_k, W_v, W_d$ |
| | 50M | $W_q, W_k, W_v$ | $W_q, W_k, W_v, W_o$ | $MLP$ | 0 |
| | 100M | $W_q, W_k, W_v, MLP$ | $W_q, W_k, W_v, W_o, MLP$ | $0, W_q, W_k, W_v$ | 0-1 |
| | 160M | $W_q, W_k, W_v, W_o, MLP$ | $W_q, W_k, W_v, W_o, MLP$ | 0-1 | 0-2 |
| | 200M | 0 | 0 | 0-1, $W_q, W_k, W_v, W_d, MLP_{fl}$ | 0-3 |
| | 300M | $0, W_q, W_v, W_o, MLP_{up}$ | $0, W_q, W_v, W_o, MLP_{up}$ | 0-3 | 0-5 |
| | 600M | 0-2 | 0-2 | 0-7 | 0-11 |

the same pattern as with Llama2-7B, where resilience emerges once a sufficient number of parameters are closed-sourced. For example, on Mistral-7B, resilience occurs after closed-sourcing 100 million parameters, which is less than a single decoder layer. Closed-sourcing fewer parameters leads to a notable drop in resilience, with $\Delta$ARR rising to around 40%. Beyond this threshold, resilience stabilizes near 0% $\Delta$ARR. This pattern holds across all models, highlighting a critical threshold for effective closed-source. Furthermore, different architectures require varying closed-sourcing quantities to achieve resilience, even with similar model sizes. For instance, Mistral-7B reaches resilience by closed-sourcing 100 million parameters, Llama2-7B requires 200 million, and Phi-1.5 needs a higher rate of 7%, compared to 3% for Llama2-7B.

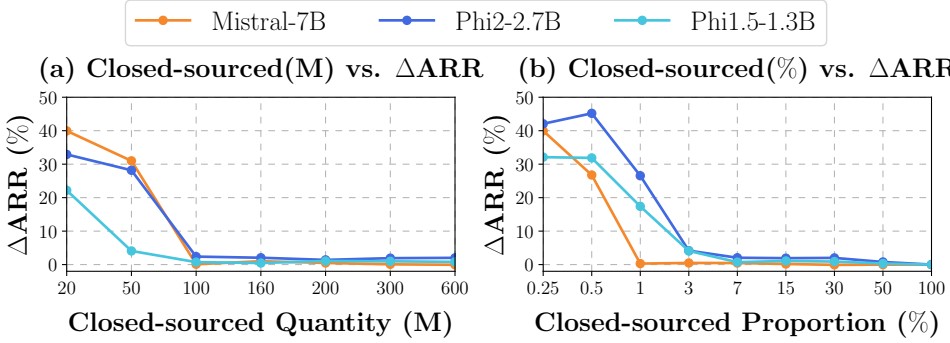

Figure 11: $\Delta$ ARR for different closed parameter quantities and proportions.

We explore how closed-sourced parameter ratio impacts the model resilience in Llama2-7B, as shown in Figure 12. For instance, technical skills such as Math show earlier transitions, with resilience emerging at 1% parameters closed-sourced, whereas domains such as Commonsense Reasoning require hiding 3%. In summary, closed-sourcing a small portion of parameters can provide sufficient resilience against model recovery, meanwhile, technical capabilities tend to be more challenging to recover than other domains.

## B.10 EFFECTIVENESS OF RECOVERY DIFFICULTY

The complete Pearson and Spearman results are presented in Table 15, revealing a negative correlation between RS and the average recovery ratio. For example, in Llama2-7B, both Pearson and Spearman coefficients fall below -0.77. Similar trends are seen in models with varying architectures and sizes, confirming that RD is a reliable predictor of recovered model performance and demonstrating the effectiveness of SCARA. Additionally, Figure 13 shows scatter plots depicting the relationship

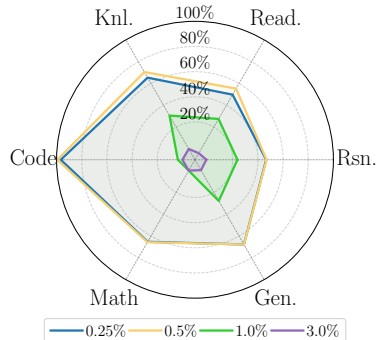

Figure 12: ΔRR in specific functions of Llama2-7B with varying closed-sourced parameter ratios.

between ΔARR and Recovery Difficulty(↑)s across four models, along with the corresponding Pearson and Spearman correlation coefficients. The Recovery Difficulty(↑)s were obtained from Section 5.3. As illustrated in Figure 13, we observe a clear trend: an increase in ΔARR corresponds to a decrease in model scores across all models analyzed. This inverse relationship is consistently supported by strong negative values for both Pearson and Spearman correlation coefficients, with the most significant negative correlation seen in Phi2-2.7B, indicating a substantial drop in model scores as ΔARR increases.

Table 15: Correlation coefficients (Spearman | Pearson) between recovery ratio and recovery difficult.

| Model | Rsn. | Read. | Knl. | Code & Math | Gen. | Avg. |
|---|---|---|---|---|---|---|
| Llama2-7B | -0.83 \| -0.97 | -0.77 \| -0.96 | -0.83 \| -0.95 | -0.85 \| -0.90 | -0.82 \| -0.93 | -0.80 \| -0.98 |
| Mistral-7B | -0.83 \| -0.89 | -0.72 \| -0.91 | -0.82 \| -0.94 | -0.78 \| -0.95 | -0.55 \| -0.87 | -0.67 \| -0.92 |
| Phi-2 | -0.93 \| -0.96 | -0.84 \| -0.96 | -0.74 \| -0.87 | -0.84 \| -0.80 | -0.84 \| -0.84 | -0.87 \| -0.95 |
| Phi-1.5 | -0.86 \| -0.97 | -0.78 \| -0.94 | -0.83 \| -0.94 | -0.90 \| -0.80 | -0.84 \| -0.89 | -0.80 \| -0.94 |

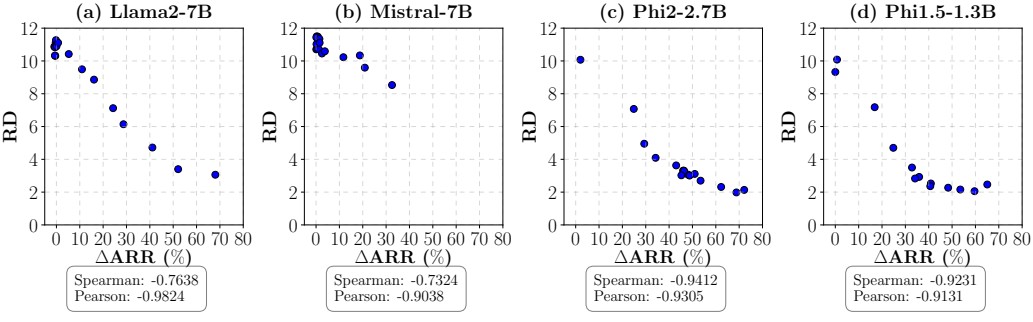

Figure 13: Correlation Analysis of ΔARR and Recovery Difficulty Across Different Models.

## B.11 ADVERSARIAL ATTACK

In this section, we provide a detailed comparison of SCARA and SAP-DP in their effectiveness against three types of black-box adversarial attacks on the Llama2-7B model. The attacks considered include Membership Inference Attacks (MIA), Attribute Inference Attacks (AIA), and Prompt Injection Attacks (PIA).

**Membership Inference Attack (MIA)**: This attack aims to determine whether a specific data point was included in the training dataset of the model. Attackers utilize model outputs to infer membership status, potentially exposing sensitive information about the training data (Fu et al., 2023; Chen &

Pattabiraman, 2024). We conducted our experiment following SPV_MIA [10], which provides a robust framework for assessing model vulnerabilities. We focus on the AUC scores for SPV-MIA against semi-open models across Ag News datasets (Zhang et al., 2016).

**Attribute Inference Attack (AIA)**: In this scenario, the adversary attempts to infer specific attributes of training data based on the model's outputs. This can lead to privacy breaches, particularly when sensitive attributes are involved (Staab et al., 2023; Li et al., 2024). We conducted our experiments following the methodology outlined in Staab et al. (2023) [11] and evaluated the top-3 accuracy on the PersonalReddit (PR) Dataset.

**Prompt Injection Attack (PIA)**: This attack manipulates input prompts to coerce the model into producing desired outputs that may compromise the integrity or security of the system (Zhao et al., 2024; Xu et al., 2024). In our experiment, we follow AutoDAN [12], which can automatically generate stealthy jailbreak prompts by the carefully designed hierarchical genetic algorithm. We evaluate the effectiveness of these prompts using the *keyword-based attack success rate* (ASR), which measures the presence of predefined keywords in responses generated LLMs. For gold standard, LED [13], significantly enhances the resilience of LLMs against prompt injection attacks (PIA), reducing the ASR to 0.

## C  DETAILED RESULTS

### C.1  COMPARISON IN TWO SEMI-OPEN LLAMA2-70B

In this experiment, we examine two semi-open Llama2-70B models, where either the first two decoder layers are closed-source (referred to as Semi-Open-1) or the last two decoder layers are closed-source (referred to as Semi-Open-2). The objective is to compare their performance in terms of customization and their resilience under the recovery attack. The results are summarized in Table 16 and Table 17.

Table 16: Customization Performance of Llama2-70B under Different Closed-Sourced Layers

|                      | Math  | Code  | Medical | Law   | Finance | Alignment |
|----------------------|-------|-------|---------|-------|---------|-----------|
| **Fully Closed-sourced** | 53.15 | 24.90 | 53.68   | 79.63 | 37.54   | 7.19      |
| **Semi-Open-1**      | 62.40 | 43.99 | 62.73   | 93.85 | 87.51   | 7.46      |
| **Semi-Open-2**      | 62.53 | 42.36 | 62.72   | 93.91 | 87.90   | 7.46      |

### C.2  EVALUATION RESULTS UNDER FT-ALL ATTACK

In this section, we provide a comprehensive analysis of the evaluation results, comparing SCARA with two baseline methods: SAP-DP and a fully-closed approach. This comparison is conducted across 16 benchmarks under the FT-all attack scenario. The detailed results for Llama2-70B are presented in Table 18, while the results for Llama2-7B and Mistral-7B are shown in Table 19. Additionally, the outcomes for Phi-2 and Phi-1.5 are provided in Tables 20.

### C.3  CUSTOMIZATION PERFORMANCE OF MODELS

In this section, we present detailed evaluation results of the model customization performance across six downstream tasks used in our experiments. The detailed results for Llama2-70B are presented in Table 21, while the results for Llama2-7B and Mistral-7B are shown in Table 22 and Table 23. Additionally, the outcomes for Phi-2 and Phi-1.5 are provided in Tables 24 and Table 25.

---

[10]https://github.com/wjfu99/MIA-LLMs
[11]https://github.com/eth-sri/llmprivacy
[12]https://github.com/SheltonLiu-N/AutoDAN
[13]https://github.com/ledllm/ledllm

Table 17: Recovery Performance of Llama2-70B under Different Closed-Sourced Layers

|  | Benchmarks | Fully Closed-sourced | Semi-Open-1 | Semi-Open-2 |
|---|---|---|---|---|
| **Rsn.** | PIQA | 50.82 | 50.49 | 79.05 |
|  | winogrande | 51.07 | 51.22 | 72.93 |
|  | arc_easy | 25.17 | 25.63 | 76.30 |
|  | arc_challenge | 23.55 | 20.48 | 50.17 |
|  | Hellaswag | 26.65 | 25.77 | 79.49 |
| **Read.** | lambada | 0.00 | 0.01 | 57.25 |
|  | BoolQ | 43.30 | 37.92 | 84.95 |
|  | SQuADv2_EM | 0.00 | 0.00 | 1.54 |
|  | SQuADv2_f1 | 0.23 | 1.01 | 35.59 |
|  | OBQA | 25.60 | 24.40 | 44.00 |
| **Knl.** | NQ | 0.00 | 0.00 | 15.18 |
|  | TriviaQA | 0.00 | 0.00 | 52.67 |
| **Code** | mbpp | 0.00 | 0.00 | 16.00 |
|  | HumanEval | 0.00 | 0.00 | 13.41 |
| **Math** | GSM8K | 0.03 | 0.01 | 27.75 |
| **Gen.** | MMLU | 23.01 | 23.22 | 63.61 |
|  | BBH | 0.00 | 0.00 | 49.45 |
| **Average Recovery Ratio($\downarrow$)** |  | 22.55 | 21.73 | 74.94 |

Table 18: Evaluation results of Llama2-70B under FT-all attack

|  | | Pre-train | SCARA | SAP-DP | Fully-closed |
|---|---|---|---|---|---|
| **Rsn.** | PIQA | 80.69 | 50.49 | 48.26 | 50.82 |
|  | Winogrande | 74.74 | 51.22 | 50.59 | 51.07 |
|  | ARC-easy | 80.35 | 25.63 | 26.35 | 25.17 |
|  | ARC-challenge | 53.24 | 20.48 | 20.31 | 23.55 |
|  | Hellaswag | 82.15 | 25.77 | 25.76 | 26.65 |
| **Read.** | LAMBADA | 75.07 | 0.01 | 0.00 | 0.00 |
|  | BoolQ | 86.70 | 37.92 | 37.83 | 43.30 |
|  | SQuADv2_EM | 51.23 | 0.00 | 0.00 | 0.00 |
|  | SQuADv2_f1 | 67.43 | 1.01 | 1.13 | 0.23 |
|  | OBQA | 44.80 | 24.40 | 24.40 | 25.60 |
| **Knl.** | NaturalQuestions | 32.38 | 0.00 | 0.00 | 0.00 |
|  | TriviaQA | 73.47 | 0.00 | 0.02 | 0.00 |
| **Code** | MBPP | 24.80 | 0.00 | 0.00 | 0.00 |
|  | HumanEval | 25.00 | 0.00 | 0.00 | 0.00 |
| **Math** | GSM8K | 53.15 | 0.01 | 0.00 | 0.03 |
| **Gen.** | MMLU | 63.09 | 23.22 | 24.19 | 23.01 |
|  | BBH | 61.40 | 0.00 | 0.00 | 0.00 |
| **Average Recovery Ratio($\downarrow$)** |  | - | 21.73 | 21.64 | 22.55 |

## C.4 COMPARISON IN CLOSING BASELINES ON LLAMA2-70B

We compare the recovery resilience of SCARA with SAP-DP and Fully-closed as baselines under FT-closed and SEM attack strategies. The evaluation results on sixteen benchmarks are shown in Table 26.

Table 19: Evaluation results of 7B models under FT-all attack

| | | Llama2-7B | | | Mistral-7B | | |
|---|---|---|---|---|---|---|---|
| | | SCARA | SAP-DP | Fully-closed | SCARA | SAP-DP | Fully-closed |
| **Rsn.** | PIQA | 49.56 | 49.56 | 49.47 | 51.63 | 50.22 | 49.35 |
| | Winogrande | 50.99 | 49.66 | 50.83 | 49.78 | 51.07 | 50.59 |
| | ARC-easy | 27.04 | 26.43 | 25.98 | 26.12 | 28.03 | 25.83 |
| | ARC-challenge | 21.07 | 20.56 | 22.47 | 19.94 | 21.42 | 22.35 |
| | Hellaswag | 25.56 | 25.69 | 26.39 | 26.10 | 25.97 | 25.39 |
| **Read.** | LAMBADA | 0.01 | 0.00 | 0.01 | 0.12 | 0.00 | 0.01 |
| | BoolQ | 44.30 | 41.70 | 48.34 | 39.05 | 37.83 | 45.80 |
| | SQuADv2_EM | 0.00 | 0.00 | 0.00 | 0.00 | 0.00 | 0.00 |
| | SQuADv2_f1 | 0.49 | 0.63 | 0.59 | 1.21 | 0.26 | 0.66 |
| | OBQA | 25.13 | 23.00 | 25.93 | 25.60 | 25.20 | 25.00 |
| **Knl.** | NaturalQuestions | 0.01 | 0.01 | 0.04 | 0.00 | 0.00 | 0.02 |
| | TriviaQA | 0.00 | 0.00 | 0.02 | 0.00 | 0.00 | 0.01 |
| **Code** | MBPP | 0.00 | 0.00 | 0.00 | 0.00 | 0.00 | 0.00 |
| | HumanEval | 0.00 | 0.00 | 0.00 | 0.00 | 0.00 | 0.00 |
| **Math** | GSM8K | 0.00 | 0.00 | 0.00 | 0.00 | 0.00 | 0.00 |
| **Gen.** | MMLU | 24.26 | 22.92 | 24.45 | 25.24 | 23.05 | 23.26 |
| | BBH | 0.00 | 0.00 | 0.00 | 0.00 | 0.00 | 0.00 |
| **Average Recovery Ratio(↓)** | | 25.03 | 24.16 | 25.62 | 22.41 | 22.28 | 22.68 |

Table 20: Evaluation results of small models under FT-all attack

| | | Phi-2 | | | Phi-1.5 | | |
|---|---|---|---|---|---|---|---|
| | | SCARA | SAP-DP | Fully-closed | SCARA | SAP-DP | Fully-closed |
| **Rsn.** | PIQA | 54.17 | 52.01 | 52.07 | 53.43 | 52.61 | 50.44 |
| | Winogrande | 51.56 | 48.93 | 48.91 | 51.09 | 49.25 | 49.12 |
| | ARC_easy | 34.57 | 28.20 | 27.03 | 30.81 | 28.79 | 27.50 |
| | ARC_challenge | 19.45 | 19.37 | 18.66 | 20.56 | 19.80 | 21.22 |
| | Hellaswag | 27.61 | 25.32 | 25.26 | 26.27 | 25.66 | 25.05 |
| | LAMBADA | 0.75 | 0.02 | 0.00 | 0.59 | 0.00 | 0.00 |
| **Read.** | BoolQ | 45.29 | 40.21 | 44.60 | 46.98 | 41.80 | 46.28 |
| | SQuADv2_EM | 0.02 | 0.00 | 0.00 | 0.00 | 0.00 | 0.00 |
| | SQuADv2_f1 | 2.61 | 0.28 | 0.64 | 0.78 | 0.65 | 1.60 |
| | OBQA | 24.80 | 26.60 | 25.80 | 26.60 | 28.60 | 26.87 |
| **Knl.** | NaturalQuestions | 0.00 | 0.00 | 0.02 | 0.04 | 0.00 | 0.00 |
| | TriviaQA | 0.01 | 0.00 | 0.01 | 0.01 | 0.00 | 0.00 |
| **Code** | MBPP | 0.00 | 0.00 | 0.00 | 0.00 | 0.00 | 0.00 |
| | HumanEval | 0.00 | 0.00 | 0.00 | 0.00 | 0.00 | 0.00 |
| **Math** | GSM8K | 0.00 | 0.00 | 0.00 | 0.00 | 0.00 | 0.00 |
| **Gen.** | MMLU | 24.16 | 22.87 | 22.95 | 24.07 | 22.95 | 22.95 |
| | BBH | 0.01 | 0.00 | 0.00 | 0.00 | 0.00 | 0.00 |

## C.5 COMPARISON IN RECOVERY ATTACK STRATEGIES

In this section, we present detailed evaluation results of the model recovery performance of SCARA under FT-closed and SEM attack strategies across six functionalities used in our experiments. The detailed results under the FT-closed recovery strategy are presented in Table 27. The results under SEM attack strategies are shown in Table 28.

Table 21: Detailed results of Llama2-70B closed by SCARA on six downstream tasks.

|  | Math | Code | Medical | Law | Finance | Alignment |
|---|---|---|---|---|---|---|
| **Fully-Closed** | 53.15 | 24.90 | 53.68 | 79.63 | 55.63 | 7.19 |
| **SAP-DP** | 61.10 | 36.87 | 54.55 | 83.40 | 65.78 | 7.41 |
| **SCARA** | 62.40 | 43.99 | 62.73 | 93.85 | 87.51 | 7.46 |
| **Fully-Open** | 64.06 | 44.58 | 63.40 | 94.17 | 88.22 | 7.42 |

Table 22: Detailed results of Llama2-7B closed by SCARA on six downstream tasks.

|  | Math | Code | Medical | Law | Finance | Alignment |
|---|---|---|---|---|---|---|
| **Fully-Closed** | 20.24 | 13.75 | 36.91 | 51.80 | 38.71 | 6.51 |
| **SAP-DP** | 20.24 | 13.75 | 36.91 | 51.80 | 38.71 | 6.52 |
| **SCARA** | 28.96 | 21.37 | 46.52 | 90.84 | 81.95 | 6.63 |
| **Fully-Open** | 29.34 | 21.265 | 47.60 | 90.49 | 84.09 | 6.63 |

Table 23: Detailed results of Mistral-7B closed by SCARA on six downstream tasks.

|  | Math | Code | Medical | Law | Finance | Alignment |
|---|---|---|---|---|---|---|
| **Fully-Closed** | 38.21 | 33.83 | 61.50 | 50.47 | 37.39 | 3.20 |
| **SAP-DP** | 41.47 | 34.44 | 63.08 | 50.37 | 38.10 | 2.47 |
| **SCARA** | 46.10 | 43.16 | 66.78 | 84.94 | 86.19 | 3.87 |
| **Fully-Open** | 45.26 | 46.08 | 66.47 | 88.13 | 84.91 | 3.78 |

Table 24: Detailed results of Phi-2 closed by SCARA on six downstream tasks.

|  | Math | Code | Medical | Law | Finance | Alignment |
|---|---|---|---|---|---|---|
| **Fully-Closed** | 57.77 | 47.59 | 43.13 | 56.46 | 54.07 | 5.22 |
| **SAP-DP** | 58.52 | 46.65 | 43.40 | 56.81 | 54.37 | 5.11 |
| **SCARA** | 59.59 | 47.79 | 45.85 | 57.11 | 56.26 | 5.26 |
| **Fully-Open** | 59.60 | 48.40 | 45.93 | 57.19 | 56.68 | 5.27 |

Table 25: Detailed results of Phi-1.5 closed by SCARA on six downstream tasks.

|  | Math | Code | Medical | Law | Finance | Alignment |
|---|---|---|---|---|---|---|
| **Fully-Closed** | 30.33 | 35.09 | 30.78 | 52.18 | 34.60 | 3.24 |
| **SAP-DP** | 30.25 | 35.45 | 32.66 | 51.99 | 34.27 | 3.68 |
| **SCARA** | 33.66 | 37.10 | 33.14 | 52.26 | 39.60 | 3.87 |
| **Fully-Open** | 34.49 | 37.45 | 33.23 | 52.34 | 39.90 | 3.68 |

## C.6 COMPARISON IN RECOVERY DATASETS SCALES

To investigate the impact of attack dataset scales on the efficiency of SCARA, we conduct model recovery attack on the Llama2-7B model using four different attack datasets of varying sizes: 100k, 200k, 300k, and 500k. The evaluation performance under different attack set scales are in Table 29

## C.7 TRANSITION LAYER RESULTS.

**Resilience Performance.** We close same-sized layer sets with different start points, and attack them using FT-all. Specifically, the sets consist of one layer for Llama2-7B (Table 30, Table 31), and two layers for Phi-2 (Table 34, Table 35). We further computed the ΔARR for each closed-source set within Mistral-7B-v0.1 and Phi-1.5 in Appendix B.8. The results for the Mistral-7B-v0.1 model are

Table 26: Evaluation results of Llama2-70B under FT-closed and SEM attack

|  |  | FT-closed | | SEM | |
|---|---|---|---|---|---|
|  |  | SCARA | SAP-DP | SCARA | SAP-DP |
| **Rsn.** | PIQA | 49.78 | 49.40 | 48.62 | 49.00 |
|  | Winogrande | 51.30 | 49.01 | 50.99 | 51.13 |
|  | ARC-easy | 26.43 | 25.59 | 25.33 | 24.55 |
|  | ARC-challenge | 21.41 | 21.42 | 22.01 | 20.93 |
|  | Hellaswag | 26.07 | 26.10 | 25.90 | 25.22 |
| **Read.** | LAMBADA | 0.00 | 0.00 | 0.00 | 0.00 |
|  | BoolQ | 45.09 | 37.83 | 44.95 | 39.80 |
|  | SQuADv2_EM | 0.00 | 0.00 | 0.00 | 0.00 |
|  | SQuADv2_f1 | 0.98 | 1.01 | 0.59 | 1.00 |
|  | OBQA | 24.40 | 23.80 | 25.03 | 22.96 |
| **Knl.** | NaturalQuestions | 0.00 | 0.00 | 0.00 | 0.00 |
|  | TriviaQA | 0.00 | 0.00 | 0.00 | 0.00 |
| **Code** | MBPP | 0.00 | 0.00 | 0.00 | 0.00 |
|  | HumanEval | 0.00 | 0.00 | 0.00 | 0.00 |
| **Math** | GSM8K | 0.00 | 0.00 | 0.00 | 0.00 |
| **Gen.** | MMLU | 23.18 | 23.66 | 22.98 | 22.83 |
|  | BBH | 0.00 | 0.00 | 0.00 | 0.00 |
| **Average Recovery Ratio($\downarrow$)** |  | 22.60 | 21.80 | 22.40 | 22.30 |

Table 27: Recovery Performance of SCARA under FT-Closed attacks.

|  |  | Llama2-7B | Mistral-7B | Phi-2 | Phi-1.5 |
|---|---|---|---|---|---|
| **Rsn.** | PIQA | 49.95 | 49.55 | 54.57 | 52.45 |
|  | Winogrande | 49.88 | 49.68 | 52.33 | 52.41 |
|  | ARC-easy | 27.65 | 25.88 | 33.33 | 31.06 |
|  | ARC-challenge | 20.81 | 22.69 | 19.03 | 18.77 |
|  | Hellaswag | 26.04 | 25.01 | 27.62 | 26.88 |
| **Read.** | LAMBADA | 0.00 | 0.00 | 0.77 | 0.71 |
|  | BoolQ | 38.13 | 46.01 | 44.34 | 57.49 |
|  | SQuADv2_EM | 0.00 | 0.00 | 0.00 | 0.00 |
|  | SQuADv2_f1 | 0.22 | 0.36 | 3.07 | 2.27 |
|  | OBQA | 25.70 | 25.12 | 24.40 | 25.20 |
| **Knl.** | NaturalQuestions | 0.00 | 0.00 | 0.00 | 0.00 |
|  | TriviaQA | 0.00 | 0.00 | 0.01 | 0.00 |
| **Code** | MBPP | 0.00 | 0.00 | 0.00 | 0.00 |
|  | HumanEval | 0.00 | 0.00 | 0.00 | 0.00 |
| **Math** | GSM8K | 0.00 | 0.00 | 0.00 | 0.00 |
| **Gen.** | MMLU | 24.23 | 23.56 | 23.03 | 24.10 |
|  | BBH | 0.00 | 0.00 | 0.00 | 0.00 |
| **Average Recovery Ratio($\downarrow$)** |  | 24.80 | 22.50 | 23.56 | 26.97 |

presented in Table 32 and Table 33. Additionally, the performance outcomes for the Phi-1.5 model can be found in Table 36.

In all the above tables, "Pretrain" represents the model's original performance without any layers closed-sourced. These columns indicate the index of layers in the model that have been closed-sourced. "*" indicates fully-closed. All evaluation scores are averages from three different seed

Table 28: Recovery Performance of SCARA under SEM attacks.

|  |  | Llama2-7B | Mistral-7B | Phi-2 | Phi-1.5 |
|---|---|---|---|---|---|
| **Rsn.** | PIQA | 51.52 | 48.53 | 49.46 | 50.82 |
|  | Winogrande | 50.28 | 51.02 | 48.70 | 50.59 |
|  | ARC-easy | 24.83 | 25.83 | 25.93 | 24.62 |
|  | ARC-challenge | 24.99 | 22.35 | 20.65 | 21.08 |
|  | Hellaswag | 25.58 | 25.39 | 25.84 | 25.39 |
| **Read.** | LAMBADA | 0.00 | 0.01 | 0.00 | 0.01 |
|  | BoolQ | 53.30 | 45.80 | 38.41 | 61.07 |
|  | SQuADv2_EM | 0.00 | 0.00 | 0.00 | 0.00 |
|  | SQuADv2_f1 | 0.77 | 0.66 | 0.00 | 1.35 |
|  | OBQA | 25.00 | 25.00 | 27.80 | 30.40 |
| **Knl.** | NaturalQuestions | 0.00 | 0.02 | 0.00 | 0.00 |
|  | TriviaQA | 0.00 | 0.01 | 0.01 | 0.00 |
| **Code** | MBPP | 0.00 | 0.00 | 0.00 | 0.00 |
|  | HumanEval | 0.00 | 0.00 | 0.00 | 0.00 |
| **Math** | GSM8K | 0.00 | 0.00 | 0.00 | 0.00 |
| **Gen.** | MMLU | 25.39 | 23.26 | 22.95 | 23.11 |
|  | BBH | 0.00 | 0.00 | 0.00 | 0.00 |
| **Average Recovery Ratio(↓)** |  | 25.00 | 22.00 | 22.10 | 24.70 |

Table 29: Evaluation Results of SCARA on Llama2-7B under Various Attack Set Scales.

|  |  | 51K | 100K | 200K | 300K | 500K |
|---|---|---|---|---|---|---|
| **Rsn.** | PIQA | 49.56 | 49.89 | 49.18 | 49.18 | 49.59 |
|  | Winogrande | 50.99 | 47.99 | 49.49 | 50.20 | 50.20 |
|  | ARC-easy | 27.04 | 27.06 | 27.06 | 27.02 | 27.01 |
|  | ARC-challenge | 21.07 | 21.33 | 20.90 | 21.16 | 21.48 |
|  | Hellaswag | 25.56 | 26.49 | 26.46 | 26.50 | 26.19 |
| **Read.** | LAMBADA | 0.01 | 0.01 | 0.00 | 0.00 | 0.01 |
|  | BoolQ | 44.30 | 44.41 | 44.10 | 44.07 | 44.96 |
|  | SQuADv2_EM | 0.00 | 0.00 | 0.00 | 0.02 | 0.00 |
|  | SQuADv2_f1 | 1.05 | 0.32 | 0.51 | 0.52 | 0.71 |
|  | OBQA | 25.13 | 25.00 | 23.80 | 25.20 | 25.60 |
| **Knl.** | NaturalQuestions | 0.01 | 0.08 | 0.08 | 0.06 | 0.06 |
|  | TriviaQA | 0.00 | 0.02 | 0.01 | 0.03 | 0.01 |
| **Code** | MBPP | 0.00 | 0.00 | 0.00 | 0.00 | 0.00 |
|  | HumanEval | 0.00 | 0.00 | 0.00 | 0.00 | 0.00 |
| **Math** | GSM8K | 0.00 | 0.00 | 0.00 | 0.00 | 0.00 |
| **Gen.** | MMLU | 24.26 | 25.34 | 25.43 | 26.14 | 26.41 |
|  | BBH | 0.00 | 0.00 | 0.00 | 0.00 | 0.00 |
| **Average Recovery Ratio(↓)** |  | 25.07 | 25.03 | 24.89 | 25.26 | 25.48 |

tests, corresponding to the values 20, 42, and 1234, following the details of the Sixteen Functionality Benchmarks in Appendix B.5.

**Customizability Performance.** We close varying numbers of layers from the start and fine-tune the open set, and then we observe the customizability transition in models. Table 37 shows the detailed evaluation results of Llama2-7B and Phi-2 on GSM8k benchmark.

Table 30: Evaluation Results of Llama2-7B under Different Closed Layers (Part1)

|  |  | Pretrain | 0 | 1 | 2 | 3 | 4 | 5 | 6 | 7 |
|---|---|---|---|---|---|---|---|---|---|---|
| **Rsn.** | PIQA | 76.66 | 49.56 | 51.43 | 49.53 | 50.45 | 49.84 | 50.27 | 50.96 | 51.09 |
| | Hellaswag | 75.45 | 25.56 | 25.75 | 25.88 | 26.16 | 25.91 | 27.20 | 29.39 | 28.89 |
| | Winogrande | 66.38 | 50.99 | 50.86 | 50.15 | 49.75 | 49.96 | 50.91 | 51.64 | 51.36 |
| | ARC_easy | 74.41 | 27.04 | 27.23 | 26.10 | 26.30 | 25.51 | 26.44 | 28.24 | 27.96 |
| | ARC_challenge | 44.11 | 21.07 | 20.31 | 20.19 | 21.30 | 22.04 | 21.56 | 20.62 | 22.92 |
| **Read.** | OpenBookQA | 68.49 | 0.01 | 0.11 | 0.02 | 0.02 | 0.01 | 0.00 | 0.05 | 0.04 |
| | LAMBADA | 80.67 | 44.30 | 41.22 | 38.36 | 41.43 | 38.08 | 38.14 | 38.40 | 41.55 |
| | BoolQ | 59.48 | 0.00 | 0.04 | 0.00 | 0.00 | 0.00 | 0.00 | 0.01 | 0.03 |
| | SQuADv2_em | 71.88 | 1.05 | 1.31 | 0.63 | 1.07 | 0.45 | 0.44 | 1.13 | 1.10 |
| | SQuADv2_f1 | 43.80 | 25.13 | 24.60 | 23.60 | 24.93 | 25.67 | 24.47 | 25.07 | 26.00 |
| **Knl.** | NaturalQuestions | 22.47 | 0.01 | 0.00 | 0.01 | 0.03 | 0.02 | 0.01 | 0.13 | 0.08 |
| | TriviaQA | 57.23 | 0.00 | 0.01 | 0.00 | 0.02 | 0.01 | 0.01 | 0.07 | 0.10 |
| **Code** | HumanEval | 10.90 | 0.00 | 0.00 | 0.00 | 0.00 | 0.00 | 0.00 | 0.00 | 0.00 |
| | MBPP | 16.60 | 0.00 | 0.00 | 0.00 | 0.00 | 0.00 | 0.00 | 0.00 | 0.00 |
| **Math** | GSM8K | 20.24 | 0.00 | 0.00 | 0.00 | 0.00 | 0.00 | 0.00 | 0.00 | 0.00 |
| **Gen.** | MMLU | 45.83 | 24.26 | 25.37 | 23.98 | 24.26 | 24.75 | 24.01 | 25.23 | 27.45 |
| | BBH | 39.86 | 0.00 | 0.00 | 0.00 | 0.00 | 0.00 | 0.00 | 0.50 | 0.38 |
| **Avg. Performance Score(↓)** | | 51.44 | 15.82 | 15.78 | 15.20 | 15.63 | 15.43 | 15.50 | 15.97 | 16.41 |
| **Average Recovery Ratio(↓)** | | - | 30.76 | 30.67 | 29.55 | 30.39 | 29.99 | 30.13 | 31.04 | 31.90 |
| **Recovery Difficulty(↑)** | | - | 11.11 | 11.27 | 10.87 | 10.31 | 10.83 | 10.33 | 10.90 | 11.11 |

## C.8 EVALUATION RESULTS UNDER DIFFERENT CLOSED SIZE

In this section, we present a comprehensive evaluation of the model's performance across sixteen benchmarks utilized in our experiments. The evaluation results for LLaMA2-7B, categorized by varying quantities and proportions of closed-source parameters, are displayed in Table 38 and Table 39, respectively. For the Mistral-7B model, the results are summarized in Table 40 and Table 41. Furthermore, the evaluation outcomes for the Phi-2 model can be found in Tables 42 and Table 43. The performance results for Phi-1.5 are also included in Tables 44 and Table 45 for comparison. For further details regarding the closed-source settings employed in our experiments, please refer to Appendix C.8.

## C.9 LIMITATION ON OPT-350M

To investigate the limitations of SCARA, we calculate the recovery ratio of each closed-source set within the smaller model, OPT-350M (Zhang et al., 2022) with only 350M parameters. We set the closed-source set size to 2 and subsequently calculate $\Delta$ARRs for each closed-source set. The detailed results are shown in Figure 46.

Table 31: Evaluation Results of Llama2-7B under Different Closed-sourced Layers (Part2). "*" indicates the fully closed-sourced model.

| | | 16 | 18 | 20 | 22 | 24 | 26 | 28 | 30 | * |
|---|---|---|---|---|---|---|---|---|---|---|
| **Rsn.** | PIQA | 51.47 | 52.99 | 58.22 | 65.83 | 69.60 | 73.45 | 75.46 | 75.99 | 49.47 |
| | Hellaswag | 31.38 | 36.55 | 45.61 | 56.60 | 62.70 | 67.88 | 71.37 | 72.94 | 26.39 |
| | Winogrande | 53.09 | 55.98 | 58.96 | 64.12 | 64.80 | 65.25 | 65.46 | 66.53 | 50.83 |
| | ARC_easy | 30.58 | 35.35 | 43.85 | 55.92 | 62.56 | 68.36 | 70.85 | 72.60 | 25.98 |
| | ARC_challenge | 24.26 | 26.85 | 30.97 | 35.38 | 38.17 | 41.41 | 43.00 | 44.17 | 22.47 |
| **Read.** | OpenBookQA | 0.28 | 1.58 | 6.79 | 30.88 | 44.58 | 56.23 | 62.33 | 63.11 | 0.01 |
| | LAMBADA | 57.55 | 70.53 | 71.36 | 78.85 | 79.69 | 80.29 | 79.39 | 80.40 | 48.34 |
| | BoolQ | 0.08 | 0.90 | 2.34 | 7.07 | 6.04 | 6.87 | 3.54 | 9.46 | 0.00 |
| | SQuADv2_em | 2.21 | 13.48 | 21.47 | 35.72 | 36.96 | 39.32 | 37.08 | 42.08 | 0.59 |
| | SQuADv2_f1 | 27.33 | 28.20 | 30.47 | 32.13 | 34.93 | 39.27 | 39.93 | 41.53 | 25.93 |
| **Knl.** | NaturalQuestions | 0.13 | 0.41 | 1.60 | 2.94 | 4.29 | 2.69 | 7.28 | 11.87 | 0.04 |
| | TriviaQA | 0.25 | 1.79 | 4.93 | 11.02 | 15.73 | 17.95 | 33.19 | 42.26 | 0.02 |
| **Code** | HumanEval | 0.00 | 0.00 | 0.00 | 0.00 | 0.00 | 3.25 | 8.34 | 10.98 | 0.00 |
| | MBPP | 0.00 | 0.00 | 0.00 | 0.07 | 0.47 | 2.27 | 8.80 | 13.27 | 0.00 |
| **Math** | GSM8K | 0.00 | 0.00 | 0.00 | 0.13 | 0.81 | 8.42 | 6.90 | 15.77 | 0.00 |
| **Gen.** | MMLU | 43.17 | 48.20 | 49.38 | 49.58 | 49.72 | 50.03 | 50.75 | 50.61 | 24.45 |
| | BBH | 0.76 | 11.44 | 19.79 | 28.87 | 31.16 | 35.98 | 38.24 | 40.54 | 0.00 |
| **Avg. Performance Score(↓)** | | 18.97 | 22.60 | 26.22 | 32.65 | 35.42 | 38.76 | 41.29 | 44.36 | 16.15 |
| **Average Recovery Ratio(↓)** | | 36.89 | 43.94 | 50.98 | 63.48 | 68.87 | 75.35 | 80.27 | 86.24 | 31.39 |
| **Recovery Difficulty(↑)** | | 10.42 | 9.49 | 8.86 | 7.12 | 6.14 | 4.72 | 3.40 | 3.06 | 11.19 |

Table 32: Evaluation Results of Mistral-7B under Different Closed-sourced Layers (Part1)

| | | Pretrain | 0 | 1 | 2 | 3 | 4 | 5 | 6 | 7 |
|---|---|---|---|---|---|---|---|---|---|---|
| **Rsn.** | PIQA | 81.99 | 51.63 | 53.20 | 53.63 | 53.47 | 51.56 | 52.61 | 50.71 | 55.15 |
| | Hellaswag | 81.04 | 26.10 | 26.36 | 26.36 | 26.66 | 27.10 | 25.51 | 26.18 | 28.16 |
| | Winogrande | 74.03 | 49.78 | 49.78 | 51.01 | 50.38 | 49.91 | 50.14 | 49.70 | 51.17 |
| | ARC_easy | 80.77 | 33.03 | 31.96 | 30.71 | 29.66 | 30.25 | 30.35 | 26.44 | 32.38 |
| | ARC_challenge | 50.26 | 19.94 | 21.27 | 20.45 | 19.60 | 20.05 | 21.36 | 21.25 | 20.73 |
| **Read.** | OpenBookQA | 44.40 | 25.60 | 25.20 | 25.20 | 25.47 | 25.87 | 26.33 | 25.07 | 27.20 |
| | LAMBADA | 73.29 | 0.12 | 0.44 | 1.91 | 2.08 | 0.80 | 0.30 | 0.17 | 1.95 |
| | BoolQ | 83.67 | 39.05 | 53.12 | 45.95 | 38.61 | 47.35 | 38.06 | 46.44 | 47.66 |
| | SQuADv2_em | 64.04 | 0.00 | 0.00 | 0.01 | 0.01 | 0.01 | 0.00 | 0.00 | 0.01 |
| | SQuADv2_f1 | 71.37 | 1.21 | 0.84 | 1.05 | 1.03 | 1.27 | 0.43 | 0.07 | 0.86 |
| **Knl.** | NaturalQuestions | 28.98 | 0.00 | 0.01 | 0.00 | 0.04 | 0.01 | 0.00 | 0.02 | 0.07 |
| | TriviaQA | 70.79 | 0.00 | 0.00 | 0.02 | 0.01 | 0.01 | 0.01 | 0.00 | 0.16 |
| **Code** | HumanEval | 29.88 | 0.00 | 0.00 | 0.00 | 0.00 | 0.00 | 0.00 | 0.00 | 0.00 |
| | MBPP | 38.40 | 0.00 | 0.00 | 0.00 | 0.00 | 0.00 | 0.00 | 0.00 | 0.00 |
| **Math** | GSM8K | 38.21 | 0.00 | 0.00 | 0.00 | 0.00 | 0.00 | 0.00 | 0.00 | 0.00 |
| **Gen.** | MMLU | 62.50 | 25.24 | 24.68 | 25.11 | 23.43 | 23.65 | 24.26 | 24.26 | 24.99 |
| | BBH | 56.40 | 0.00 | 0.00 | 0.00 | 0.00 | 0.00 | 0.00 | 0.00 | 0.01 |
| **Avg. Performance Score(↓)** | | 60.59 | 15.98 | 16.87 | 16.55 | 15.91 | 16.34 | 15.84 | 15.90 | 17.09 |
| **Average Recovery Ratio(↓)** | | - | 26.38 | 27.85 | 27.32 | 26.25 | 26.97 | 26.15 | 26.24 | 28.20 |
| **Average Recovery Ratio(↓)** | | - | 11.50 | 11.31 | 11.48 | 10.71 | 10.77 | 11.44 | 11.02 | 10.71 |

Table 33: Evaluation Results of Mistral-7B under Different Closed-sourced Layers (Part2)

| | | 16 | 18 | 20 | 22 | 24 | 26 | 28 | 30 | * |
|---|---|---|---|---|---|---|---|---|---|---|
| **Rsn.** | PIQA | 54.50 | 52.32 | 52.72 | 57.13 | 62.82 | 64.67 | 67.23 | 75.61 | 49.35 |
| | Hellaswag | 29.31 | 29.02 | 29.99 | 33.46 | 46.21 | 52.12 | 52.46 | 67.73 | 25.39 |
| | Winogrande | 51.20 | 54.17 | 51.07 | 55.75 | 58.59 | 62.41 | 63.09 | 66.33 | 50.59 |
| | ARC_easy | 32.84 | 29.35 | 30.80 | 38.04 | 47.24 | 51.99 | 54.74 | 69.95 | 25.83 |
| | ARC_challenge | 21.19 | 23.04 | 23.78 | 26.34 | 30.86 | 33.22 | 35.04 | 40.53 | 22.35 |
| **Read.** | OpenBookQA | 26.00 | 27.87 | 26.87 | 29.67 | 28.73 | 32.67 | 33.40 | 36.40 | 25.00 |
| | LAMBADA | 2.61 | 0.18 | 1.28 | 4.17 | 21.89 | 29.93 | 24.49 | 48.32 | 0.01 |
| | BoolQ | 53.98 | 53.60 | 58.79 | 55.76 | 64.10 | 74.72 | 68.48 | 81.30 | 45.80 |
| | SQuADv2_em | 0.01 | 0.00 | 0.47 | 0.13 | 2.39 | 3.59 | 1.87 | 1.82 | 0.00 |
| | SQuADv2_f1 | 0.96 | 0.18 | 1.27 | 2.60 | 14.88 | 22.61 | 21.12 | 34.16 | 0.66 |
| **Knl.** | NaturalQuestions | 0.01 | 0.10 | 0.19 | 0.58 | 1.84 | 3.15 | 3.53 | 8.87 | 0.02 |
| | TriviaQA | 0.03 | 0.01 | 0.61 | 0.62 | 5.14 | 7.51 | 10.32 | 25.44 | 0.01 |
| **Code** | HumanEval | 0.00 | 0.00 | 0.00 | 0.61 | 2.24 | 4.88 | 2.44 | 9.75 | 0.00 |
| | MBPP | 0.00 | 0.00 | 0.00 | 2.00 | 4.33 | 8.33 | 0.93 | 13.07 | 0.00 |
| **Math** | GSM8K | 0.00 | 0.00 | 0.00 | 0.00 | 0.00 | 0.00 | 0.00 | 0.25 | 0.00 |
| **Gen.** | MMLU | 24.30 | 25.84 | 29.54 | 24.55 | 34.77 | 40.77 | 40.84 | 50.44 | 23.26 |
| | BBH | 0.00 | 0.00 | 0.02 | 0.30 | 7.55 | 18.76 | 21.05 | 30.07 | 0.00 |
| **Avg. Performance Score(↓)** | | 17.47 | 17.39 | 18.08 | 19.51 | 25.51 | 30.08 | 29.47 | 38.83 | 15.78 |
| **Average Recovery Ratio(↓)** | | 28.83 | 28.71 | 29.84 | 32.20 | 42.09 | 49.64 | 48.64 | 64.08 | 26.05 |
| **Average Recovery Ratio(↓)** | | 11.34 | 11.11 | 10.45 | 10.59 | 10.23 | 10.34 | 9.59 | 8.53 | 11.20 |

Table 34: Evaluation Results of Phi-2 under Different Closed-sourced Layers (Part 1)

| | | Pretrain | 0 | 2 | 4 | 6 | 8 | 10 | 12 | 14 |
|---|---|---|---|---|---|---|---|---|---|---|
| **Rsn.** | PIQA | 79.27 | 54.17 | 72.85 | 73.76 | 75.03 | 76.75 | 78.00 | 78.91 | 77.84 |
| | Hellaswag | 73.73 | 27.61 | 56.49 | 57.73 | 60.47 | 62.84 | 66.39 | 66.91 | 66.95 |
| | Winogrande | 75.45 | 51.56 | 59.17 | 59.98 | 59.88 | 64.32 | 68.11 | 68.95 | 70.38 |
| | ARC_easy | 79.92 | 34.57 | 72.94 | 73.40 | 73.97 | 76.51 | 78.33 | 78.66 | 78.63 |
| | ARC_challenge | 52.90 | 19.45 | 41.75 | 39.82 | 44.11 | 45.65 | 47.92 | 49.74 | 48.78 |
| **Read.** | OpenBookQA | 51.20 | 25.80 | 35.73 | 37.47 | 40.13 | 42.00 | 44.00 | 45.67 | 44.80 |
| | LAMBADA | 56.28 | 3.25 | 28.55 | 30.42 | 34.64 | 40.05 | 45.41 | 45.52 | 46.66 |
| | BoolQ | 83.36 | 47.29 | 65.20 | 62.64 | 66.39 | 71.39 | 73.42 | 72.95 | 75.83 |
| | SQuADv2_em | 61.30 | 0.02 | 10.49 | 17.63 | 21.94 | 33.94 | 19.54 | 19.15 | 29.14 |
| | SQuADv2_f1 | 71.38 | 2.61 | 37.22 | 40.35 | 45.53 | 59.16 | 48.21 | 50.09 | 54.87 |
| **Knl.** | NaturalQuestions | 9.58 | 0.00 | 3.60 | 4.97 | 6.13 | 7.55 | 7.95 | 8.10 | 9.25 |
| | TriviaQA | 39.29 | 0.01 | 13.57 | 16.29 | 24.74 | 28.60 | 31.58 | 33.71 | 32.79 |
| **Code** | HumanEval | 48.78 | 0.00 | 1.42 | 6.50 | 10.98 | 16.66 | 22.76 | 19.51 | 23.17 |
| | MBPP | 46.80 | 0.00 | 5.07 | 6.87 | 9.47 | 19.60 | 25.67 | 23.47 | 25.73 |
| **Math** | GSM8K | 57.77 | 0.00 | 7.25 | 8.64 | 4.42 | 9.63 | 14.18 | 11.35 | 17.31 |
| **Gen.** | MMLU | 56.73 | 26.16 | 34.29 | 37.01 | 39.90 | 43.11 | 45.63 | 48.17 | 49.82 |
| | BBH | 59.53 | 0.01 | 15.27 | 18.37 | 16.38 | 14.58 | 4.93 | 4.35 | 11.37 |
| **Avg. Performance Score(↓)** | | 59.02 | 17.21 | 32.99 | 34.81 | 37.30 | 41.90 | 42.47 | 42.66 | 44.90 |
| **Average Recovery Ratio(↓)** | | - | 29.15 | 55.90 | 58.99 | 63.21 | 71.00 | 71.97 | 72.28 | 76.09 |
| **Recovery Difficulty(↑)** | | - | 10.07 | 7.07 | 4.95 | 4.09 | 3.63 | 3.31 | 3.31 | 3.11 |

Table 35: Evaluation Results of Phi-2 under Different Closed-sourced Layers (Part2). "*" indicates the fully closed-sourced model.

| | | 16 | 18 | 20 | 22 | 24 | 26 | 28 | 30 | * |
|---|---|---|---|---|---|---|---|---|---|---|
| **Rsn.** | PIQA | 77.44 | 77.80 | 77.69 | 76.77 | 76.89 | 77.55 | 78.16 | 78.58 | 52.07 |
| | Hellaswag | 67.20 | 66.90 | 67.13 | 68.00 | 68.86 | 70.01 | 71.44 | 71.18 | 25.26 |
| | Winogrande | 70.82 | 71.40 | 73.11 | 74.46 | 75.79 | 75.72 | 75.93 | 74.77 | 48.91 |
| | ARC_easy | 78.30 | 77.27 | 77.33 | 76.82 | 78.09 | 77.76 | 79.53 | 79.56 | 27.03 |
| | ARC_challenge | 49.71 | 48.29 | 48.52 | 48.04 | 49.80 | 50.68 | 53.16 | 52.67 | 18.66 |
| **Read.** | OpenBookQA | 46.53 | 46.47 | 45.87 | 45.27 | 46.33 | 45.53 | 46.53 | 48.27 | 20.80 |
| | LAMBADA | 45.67 | 46.88 | 47.95 | 50.17 | 50.54 | 52.77 | 53.01 | 53.23 | 0.00 |
| | BoolQ | 80.56 | 80.72 | 82.22 | 83.31 | 83.98 | 83.54 | 82.54 | 83.41 | 39.60 |
| | SQuADv2_em | 7.88 | 1.30 | 1.69 | 1.31 | 0.15 | 0.23 | 3.54 | 10.03 | 0.56 |
| | SQuADv2_f1 | 40.84 | 34.51 | 34.25 | 35.94 | 35.64 | 36.68 | 39.57 | 44.87 | 0.90 |
| **Knl.** | NaturalQuestions | 8.90 | 6.09 | 6.40 | 6.79 | 6.86 | 6.85 | 7.20 | 8.37 | 0.02 |
| | TriviaQA | 31.48 | 27.03 | 25.08 | 24.54 | 22.89 | 22.99 | 24.24 | 26.93 | 0.01 |
| **Code** | HumanEval | 22.56 | 21.34 | 25.41 | 32.52 | 38.01 | 46.14 | 46.54 | 43.90 | 0.00 |
| | MBPP | 26.73 | 25.33 | 24.80 | 31.73 | 36.67 | 41.80 | 43.13 | 43.20 | 0.00 |
| **Math** | GSM8K | 16.68 | 16.02 | 14.66 | 12.31 | 17.24 | 30.12 | 45.41 | 49.79 | 0.00 |
| **Gen.** | MMLU | 52.69 | 53.45 | 55.68 | 56.61 | 56.93 | 56.59 | 56.86 | 56.47 | 22.95 |
| | BBH | 3.42 | 17.36 | 8.33 | 18.24 | 30.09 | 48.12 | 52.28 | 56.36 | 0.00 |
| **Avg. Performance Score(↓)** | | 42.79 | 42.25 | 42.12 | 43.70 | 45.57 | 48.42 | 50.53 | 51.86 | 15.10 |
| **Average Recovery Ratio(↓)** | | 72.51 | 71.58 | 71.38 | 74.04 | 77.22 | 82.04 | 85.63 | 87.87 | 25.59 |
| **Recovery Difficulty(↑)** | | 3.07 | 3.29 | 3.03 | 3.01 | 2.70 | 2.32 | 1.98 | 2.13 | 11.32 |

Table 36: Evaluation Results of Phi-1.5 under Different Closed-sourced Layers

| | | Pretrain | 0-1 | 2-3 | 4-5 | 6-7 | 8-9 | 10-11 | 12-13 | 14-15 | 16-17 | 18-19 | 20-21 | 22-23 | * |
|---|---|---|---|---|---|---|---|---|---|---|---|---|---|---|---|
| **Rsn.** | PIQA | 75.68 | 53.43 | 69.52 | 71.53 | 73.50 | 74.76 | 75.08 | 74.94 | 74.64 | 73.90 | 74.63 | 74.54 | 74.81 | 50.44 |
| | Hellaswag | 62.56 | 26.27 | 46.66 | 50.71 | 52.98 | 54.51 | 55.11 | 56.01 | 56.78 | 57.90 | 58.76 | 59.35 | 58.58 | 25.05 |
| | Winogrande | 72.69 | 51.09 | 54.91 | 59.22 | 61.75 | 64.85 | 67.95 | 68.88 | 68.98 | 71.25 | 71.19 | 72.87 | 70.66 | 49.12 |
| | ARC_easy | 76.14 | 30.81 | 61.70 | 65.70 | 70.10 | 71.38 | 70.01 | 71.72 | 71.93 | 72.34 | 73.39 | 74.16 | 73.74 | 27.50 |
| | ARC_challenge | 44.62 | 20.56 | 32.85 | 34.10 | 38.08 | 40.05 | 40.30 | 39.48 | 40.87 | 41.52 | 42.84 | 42.58 | 45.42 | 21.22 |
| **Read.** | OpenBookQA | 48.00 | 26.60 | 33.93 | 35.73 | 40.40 | 41.13 | 40.67 | 41.73 | 41.67 | 40.27 | 41.33 | 43.27 | 45.47 | 26.87 |
| | LAMBADA | 44.10 | 0.59 | 17.96 | 26.45 | 29.37 | 33.83 | 33.85 | 36.46 | 37.06 | 37.96 | 39.98 | 41.10 | 40.49 | 0.00 |
| | BoolQ | 75.05 | 46.98 | 59.12 | 52.42 | 57.41 | 65.68 | 68.52 | 63.47 | 65.12 | 66.52 | 73.91 | 75.17 | 77.0 | 46.28 |
| | SQuADv2_em | 48.01 | 0.00 | 5.82 | 10.94 | 18.34 | 13.96 | 14.70 | 23.22 | 16.98 | 26.05 | 22.04 | 20.16 | 26.86 | 0.00 |
| | SQuADv2_f1 | 60.84 | 0.78 | 24.49 | 26.04 | 34.86 | 32.17 | 32.36 | 43.14 | 38.23 | 48.03 | 45.75 | 45.56 | 49.62 | 1.60 |
| **Knl.** | NaturalQuestions | 5.46 | 0.04 | 1.68 | 2.73 | 3.41 | 3.06 | 3.21 | 4.25 | 4.03 | 4.06 | 4.54 | 4.17 | 4.45 | 0.01 |
| | TriviaQA | 16.94 | 0.01 | 5.70 | 7.77 | 10.85 | 11.03 | 9.11 | 12.11 | 11.84 | 11.86 | 12.02 | 12.11 | 13.19 | 0.01 |
| **Code** | HumanEval | 35.98 | 0.00 | 3.05 | 10.57 | 12.20 | 16.26 | 13.82 | 17.48 | 18.70 | 23.17 | 29.68 | 31.91 | 31.71 | 0.00 |
| | MBPP | 35.40 | 0.00 | 2.80 | 7.80 | 10.93 | 17.40 | 16.53 | 16.13 | 16.67 | 22.27 | 27.33 | 28.27 | 28.53 | 0.00 |
| **Math** | GSM8K | 30.33 | 0.00 | 0.05 | 0.73 | 0.15 | 0.23 | 0.75 | 0.50 | 2.17 | 4.98 | 9.73 | 17.77 | 23.45 | 0.00 |
| **Gen.** | MMLU | 42.44 | 24.07 | 26.56 | 28.77 | 32.51 | 32.87 | 36.09 | 39.42 | 39.72 | 43.23 | 42.51 | 42.82 | 43.66 | 23.95 |
| | BBH | 28.80 | 0.00 | 2.07 | 3.97 | 8.38 | 7.37 | 2.81 | 7.79 | 4.12 | 10.63 | 6.94 | 10.34 | 11.45 | 0.00 |
| **Avg. Performance Score(↓)** | | 47.24 | 16.54 | 26.40 | 29.13 | 32.66 | 34.15 | 34.17 | 36.28 | 35.85 | 38.59 | 39.80 | 40.95 | 42.30 | 15.94 |
| **Average Recovery Ratio(↓)** | | - | 35.02 | 55.90 | 61.66 | 69.14 | 72.29 | 72.34 | 76.80 | 75.90 | 81.68 | 84.25 | 86.69 | 89.56 | 33.75 |
| **Average Recovery Ratio(↓)** | | - | 10.08 | 7.18 | 4.70 | 3.50 | 2.93 | 2.83 | 2.53 | 2.36 | 2.27 | 2.16 | 2.06 | 2.46 | 9.33 |

Table 37: Customization Performance under Different Closed Sets

| Llama2-7B | | Phi-2 | |
|---|---|---|---|
| **Closed Layers** | **GSM8K(↑)** | **Closed Layers** | **GSM8K(↑)** |
| **Fully-open** | 29.34 | **Fully-open** | 59.60 |
| **0** | 28.96 | **0-1** | 59.59 |
| **0-4** | 21.76 | **0-5** | 58.60 |
| **0-8** | 21.46 | **0-9** | 58.45 |
| **0-12** | 20.85 | **0-13** | 55.19 |
| **0-16** | 20.11 | **0-17** | 56.25 |
| **0-20** | 21.46 | **0-21** | 54.59 |
| **0-24** | 21.44 | **0-25** | 55.34 |
| **0-28** | 18.73 | **0-29** | 54.59 |
| **Fully-Closed** | 20.32 | **Fully-Closed** | 57.77 |

Table 38: Evaluation Results of Llama2-7B under Different Closed-source Proportion

| | | 0.25% | 0.5% | 1% | 3% | 7% | 15% | 30% | 50% | 100% |
|---|---|---|---|---|---|---|---|---|---|---|
| **Rsn.** | PIQA | 77.78 | 77.69 | 67.73 | 49.42 | 49.55 | 50.05 | 49.98 | 49.31 | 49.47 |
| | Hellaswag | 71.40 | 71.54 | 52.39 | 25.74 | 26.03 | 26.25 | 25.67 | 25.48 | 26.39 |
| | Winogrande | 64.64 | 65.64 | 54.12 | 50.38 | 50.43 | 49.65 | 49.59 | 49.62 | 50.83 |
| | ARC_easy | 74.69 | 75.04 | 53.82 | 26.03 | 26.76 | 26.46 | 26.64 | 26.66 | 25.98 |
| | ARC_challenge | 43.66 | 43.29 | 26.99 | 20.16 | 21.39 | 19.74 | 21.44 | 21.73 | 22.47 |
| **Read.** | OpenBookQA | 63.15 | 63.62 | 33.20 | 0.01 | 0.00 | 0.02 | 0.01 | 0.01 | 0.01 |
| | LAMBADA | 80.66 | 80.78 | 62.10 | 38.22 | 39.33 | 43.45 | 39.39 | 41.83 | 48.34 |
| | BoolQ | 11.39 | 12.14 | 5.47 | 0.00 | 0.00 | 0.00 | 0.00 | 0.00 | 0.00 |
| | SQuADv2_em | 40.24 | 40.74 | 32.65 | 0.78 | 0.20 | 0.24 | 2.09 | 2.13 | 0.59 |
| | SQuADv2_f1 | 40.73 | 40.67 | 30.47 | 22.93 | 23.40 | 25.53 | 24.07 | 23.07 | 25.93 |
| **Knl.** | NaturalQuestions | 7.83 | 7.89 | 5.61 | 0.00 | 0.01 | 0.02 | 0.01 | 0.00 | 0.04 |
| | TriviaQA | 44.29 | 45.95 | 18.78 | 0.00 | 0.01 | 0.00 | 0.00 | 0.00 | 0.02 |
| **Code** | HumanEval | 11.39 | 12.00 | 0.00 | 0.00 | 0.00 | 0.00 | 0.00 | 0.00 | 0.00 |
| | MBPP | 15.20 | 15.33 | 1.00 | 0.00 | 0.00 | 0.00 | 0.00 | 0.00 | 0.00 |
| **Math** | GSM8K | 13.22 | 13.29 | 0.00 | 0.00 | 0.00 | 0.00 | 0.00 | 0.00 | 0.00 |
| **Gen.** | MMLU | 45.04 | 45.03 | 30.90 | 24.06 | 24.04 | 25.01 | 23.19 | 23.11 | 24.45 |
| | BBH | 37.45 | 37.51 | 17.36 | 0.00 | 0.00 | 0.00 | 0.00 | 0.00 | 0.00 |
| **Avg. Performance Score(↓)** | | 43.69 | 44.01 | 28.98 | 15.16 | 15.36 | 15.67 | 15.42 | 15.47 | 16.15 |
| **Average Recovery Ratio(↓)** | | 84.94 | 85.56 | 56.33 | 29.48 | 29.86 | 30.47 | 29.97 | 30.07 | 31.39 |
| **Recovery Difficulty(↑)** | | 1.96 | 1.93 | 8.66 | 10.87 | 11.75 | 11.48 | 11.65 | 11.57 | 11.19 |

Table 39: Evaluation Results of Llama2-7B under Different Closed-source Quantity

| | | 20M | 50M | 100M | 160M | 200M | 300M | 600M |
|---|---|---|---|---|---|---|---|---|
| **Rsn.** | PIQA | 77.78 | 73.49 | 67.55 | 67.12 | 49.42 | 50.36 | 49.97 |
| | Hellaswag | 71.40 | 63.47 | 51.67 | 51.27 | 25.74 | 25.70 | 25.78 |
| | Winogrande | 64.64 | 57.54 | 53.07 | 52.04 | 50.38 | 49.28 | 50.49 |
| | ARC_easy | 74.69 | 66.50 | 51.97 | 52.11 | 26.03 | 26.43 | 26.29 |
| | ARC_challenge | 43.66 | 36.04 | 26.51 | 25.99 | 20.16 | 20.79 | 21.70 |
| **Read.** | OpenBookQA | 63.15 | 45.34 | 30.22 | 28.75 | 0.01 | 0.05 | 0.01 |
| | LAMBADA | 80.66 | 69.47 | 62.28 | 62.59 | 38.22 | 39.03 | 40.80 |
| | BoolQ | 11.39 | 2.21 | 4.18 | 7.24 | 0.00 | 0.00 | 0.01 |
| | SQuADv2_em | 40.24 | 33.98 | 28.98 | 31.05 | 0.78 | 0.74 | 0.37 |
| | SQuADv2_f1 | 40.73 | 33.93 | 29.13 | 30.00 | 22.93 | 23.80 | 23.53 |
| **Knl.** | NaturalQuestions | 7.83 | 2.98 | 5.33 | 5.73 | 0.00 | 0.00 | 0.02 |
| | TriviaQA | 44.29 | 15.28 | 13.71 | 17.25 | 0.00 | 0.00 | 0.01 |
| **Code** | HumanEval | 11.39 | 0.41 | 0.00 | 0.00 | 0.00 | 0.00 | 0.00 |
| | MBPP | 15.20 | 6.87 | 1.00 | 0.80 | 0.00 | 0.00 | 0.00 |
| **Math** | GSM8K | 9.00 | 0.10 | 0.00 | 0.00 | 0.00 | 0.00 | 0.00 |
| **Gen.** | MMLU | 45.04 | 36.15 | 28.95 | 29.04 | 24.06 | 23.70 | 23.45 |
| | BBH | 37.45 | 28.53 | 14.99 | 16.99 | 0.00 | 0.00 | 0.00 |
| **Avg. Performance Score(↓)** | | 43.44 | 33.66 | 27.62 | 28.12 | 15.16 | 15.29 | 15.44 |
| **Average Recovery Ratio(↓)** | | 84.46 | 65.44 | 53.69 | 54.66 | 29.48 | 29.72 | 30.01 |
| **Recovery Difficulty(↑)** | | 1.96 | 5.48 | 8.95 | 9.25 | 10.87 | 10.93 | 10.81 |

Table 40: Evaluation Results of Mistral-7B under Different Closed-sourced Proportion

| | | 0.25% | 1% | 0.5% | 3% | 7% | 15% | 30% | 50% | 100% |
|---|---|---|---|---|---|---|---|---|---|---|
| **Rsn.** | PIQA | 77.79 | 74.36 | 52.16 | 53.34 | 52.07 | 52.19 | 50.04 | 50.60 | 49.35 |
| | Hellaswag | 71.31 | 65.50 | 26.50 | 26.16 | 25.92 | 25.91 | 25.87 | 25.61 | 25.39 |
| | Winogrande | 67.09 | 60.32 | 49.22 | 51.65 | 50.01 | 51.36 | 51.36 | 49.65 | 50.59 |
| | ARC_easy | 74.52 | 69.51 | 29.95 | 30.82 | 29.73 | 30.44 | 28.20 | 27.45 | 25.83 |
| | ARC_challenge | 42.32 | 38.40 | 20.76 | 20.71 | 21.10 | 20.25 | 22.61 | 22.47 | 22.35 |
| **Read.** | OpenBookQA | 42.13 | 34.60 | 25.13 | 25.33 | 26.47 | 26.07 | 25.20 | 25.87 | 25.00 |
| | LAMBADA | 55.99 | 44.36 | 0.73 | 1.66 | 0.96 | 0.31 | 0.03 | 0.02 | 0.01 |
| | BoolQ | 78.35 | 74.06 | 43.18 | 42.01 | 42.09 | 40.02 | 38.53 | 39.91 | 45.80 |
| | SQuADv2_em | 13.91 | 6.97 | 0.00 | 0.01 | 0.00 | 0.00 | 0.00 | 0.00 | 0.00 |
| | SQuADv2_f1 | 41.13 | 33.88 | 1.60 | 0.93 | 1.27 | 0.71 | 0.99 | 0.86 | 0.66 |
| **Knl.** | NaturalQuestions | 8.46 | 5.82 | 0.03 | 0.00 | 0.02 | 0.03 | 0.00 | 0.00 | 0.02 |
| | TriviaQA | 34.04 | 17.03 | 0.01 | 0.01 | 0.02 | 0.01 | 0.00 | 0.00 | 0.01 |
| **Code** | HumanEval | 11.99 | 6.51 | 0.00 | 0.00 | 0.00 | 0.00 | 0.00 | 0.00 | 0.00 |
| | MBPP | 16.93 | 12.80 | 0.00 | 0.00 | 0.00 | 0.00 | 0.00 | 0.00 | 0.00 |
| **Math** | GSM8K | 6.32 | 0.45 | 0.00 | 0.00 | 0.00 | 0.00 | 0.00 | 0.00 | 0.00 |
| **Gen.** | MMLU | 44.17 | 37.98 | 23.98 | 24.34 | 25.10 | 23.91 | 23.68 | 24.12 | 23.26 |
| | BBH | 35.44 | 27.27 | 0.02 | 0.00 | 0.00 | 0.00 | 0.00 | 0.00 | 0.00 |
| **Avg. Performance Score(↓)** | | 42.46 | 35.87 | 16.08 | 16.29 | 16.16 | 15.95 | 15.68 | 15.68 | 15.78 |
| **Average Recovery Ratio(↓)** | | 70.08 | 59.20 | 26.53 | 26.89 | 26.67 | 26.33 | 25.87 | 25.88 | 26.05 |
| **Recovery Difficulty(↑)** | | 2.22 | 5.48 | 10.92 | 11.29 | 11.35 | 11.19 | 11.17 | 11.20 | 11.20 |

Table 41: Evaluation Results of Mistral-7B under Different Closed-sourced Quantity

| | | 20M | 50M | 100M | 160M | 200M | 300M | 600M |
|---|---|---|---|---|---|---|---|---|
| **Rsn.** | PIQA | 77.79 | 73.74 | 51.36 | 52.86 | 53.34 | 50.98 | 51.62 |
| | Hellaswag | 71.31 | 65.51 | 26.49 | 27.98 | 26.16 | 26.27 | 26.04 |
| | Winogrande | 67.09 | 64.51 | 50.06 | 49.51 | 51.65 | 50.17 | 50.85 |
| | ARC_easy | 74.52 | 68.29 | 27.84 | 30.95 | 30.82 | 27.36 | 28.30 |
| | ARC_challenge | 42.32 | 37.97 | 20.85 | 21.67 | 20.71 | 21.28 | 20.17 |
| **Read.** | OpenBookQA | 42.13 | 37.27 | 25.60 | 25.87 | 25.33 | 26.60 | 27.00 |
| | LAMBADA | 55.99 | 47.63 | 1.16 | 4.74 | 1.66 | 0.43 | 0.53 |
| | BoolQ | 78.35 | 75.00 | 40.17 | 47.05 | 42.01 | 42.05 | 39.03 |
| | SQuADv2_em | 13.91 | 8.65 | 0.01 | 0.04 | 0.01 | 0.01 | 0.00 |
| | SQuADv2_f1 | 41.13 | 35.50 | 1.01 | 0.49 | 0.93 | 0.28 | 0.39 |
| **Knl.** | NaturalQuestions | 8.46 | 7.82 | 0.02 | 0.05 | 0.00 | 0.01 | 0.02 |
| | TriviaQA | 34.04 | 22.89 | 0.02 | 0.19 | 0.01 | 0.01 | 0.01 |
| **Code** | HumanEval | 11.99 | 7.93 | 0.00 | 0.00 | 0.00 | 0.00 | 0.00 |
| | MBPP | 16.93 | 11.87 | 0.00 | 0.00 | 0.00 | 0.00 | 0.00 |
| **Math** | GSM8K | 6.32 | 2.48 | 0.00 | 0.00 | 0.00 | 0.00 | 0.00 |
| **Gen.** | MMLU | 44.17 | 41.28 | 24.22 | 24.44 | 24.34 | 23.78 | 23.33 |
| | BBH | 35.44 | 33.43 | 0.00 | 0.40 | 0.00 | 0.00 | 0.00 |
| **Avg. Performance Score(↓)** | | 42.46 | 37.75 | 15.81 | 16.84 | 16.29 | 15.84 | 15.72 |
| **Average Recovery Ratio(↓)** | | 70.08 | 62.31 | 26.10 | 27.79 | 26.89 | 26.14 | 25.95 |
| **Recovery Difficulty(↑)** | | 2.22 | 3.44 | 11.14 | 10.85 | 11.10 | 11.23 | 11.22 |

Table 42: Evaluation Results of Phi-2 under Different Closed-sourced Proportion

| | | 0.25% | 0.5% | 1% | 3% | 7% | 15% | 30% | 50% | 100% |
|---|---|---|---|---|---|---|---|---|---|---|
| **Rsn.** | PIQA | 70.40 | 70.71 | 74.64 | 54.43 | 54.17 | 54.75 | 54.37 | 52.39 | 52.07 |
| | Hellaswag | 53.13 | 52.99 | 62.84 | 27.88 | 27.61 | 27.77 | 28.01 | 26.30 | 25.26 |
| | Winogrande | 66.17 | 66.43 | 69.93 | 51.49 | 51.56 | 51.46 | 51.44 | 49.12 | 48.91 |
| | ARC_easy | 64.62 | 65.33 | 72.55 | 33.39 | 34.57 | 32.00 | 32.18 | 29.97 | 27.03 |
| | ARC_challenge | 43.26 | 43.86 | 40.67 | 20.82 | 19.45 | 20.00 | 20.56 | 19.88 | 18.66 |
| **Read.** | OpenBookQA | 41.80 | 42.67 | 38.87 | 26.87 | 25.80 | 26.33 | 26.53 | 26.07 | 20.80 |
| | LAMBADA | 32.51 | 32.25 | 40.24 | 10.58 | 3.25 | 3.87 | 6.06 | 0.66 | 0.00 |
| | BoolQ | 65.77 | 65.27 | 76.84 | 48.13 | 47.29 | 45.62 | 46.15 | 40.50 | 39.60 |
| | SQuADv2_em | 0.36 | 9.09 | 3.31 | 0.02 | 0.02 | 0.01 | 0.01 | 0.00 | 0.56 |
| | SQuADv2_f1 | 24.81 | 30.83 | 30.47 | 0.45 | 2.61 | 0.57 | 2.52 | 1.67 | 0.90 |
| **Knl.** | NaturalQuestions | 5.70 | 5.06 | 1.14 | 0.03 | 0.00 | 0.01 | 0.07 | 0.03 | 0.02 |
| | TriviaQA | 20.27 | 21.50 | 8.78 | 2.02 | 0.01 | 0.02 | 0.01 | 0.01 | 0.01 |
| **Code** | HumanEval | 22.16 | 26.83 | 17.68 | 0.00 | 0.00 | 0.00 | 0.00 | 0.00 | 0.00 |
| | MBPP | 25.07 | 26.40 | 9.73 | 0.00 | 0.00 | 0.00 | 0.00 | 0.00 | 0.00 |
| **Math** | GSM8K | 29.26 | 31.36 | 2.00 | 0.00 | 0.00 | 0.00 | 0.00 | 0.00 | 0.00 |
| **Gen.** | MMLU | 41.76 | 42.17 | 43.86 | 30.31 | 26.16 | 25.79 | 24.85 | 24.03 | 22.95 |
| | BBH | 18.98 | 21.55 | 9.59 | 3.06 | 0.01 | 0.79 | 0.24 | 0.00 | 0.00 |
| **Avg. Performance Score(↓)** | | 36.83 | 38.49 | 35.48 | 18.20 | 17.21 | 17.00 | 17.24 | 15.92 | 15.10 |
| **Average Recovery Ratio(↓)** | | 62.40 | 65.22 | 60.12 | 30.95 | 29.15 | 28.81 | 29.21 | 26.97 | 25.59 |
| **Recovery Difficulty(↑)** | | 6.70 | 6.65 | 2.00 | 9.14 | 10.07 | 10.13 | 10.14 | 9.82 | 11.32 |

Table 43: Evaluation Results of Phi-2 under Different Closed-sourced Quantity

| | | 20M | 50M | 100M | 160M | 200M | 300M | 600M |
|---|---|---|---|---|---|---|---|---|
| **Rsn.** | PIQA | 73.70 | 70.00 | 53.90 | 54.17 | 53.01 | 54.75 | 54.28 |
| | Hellaswag | 59.75 | 55.64 | 28.26 | 27.61 | 26.90 | 27.77 | 28.61 |
| | Winogrande | 66.61 | 67.17 | 51.96 | 51.56 | 52.28 | 51.46 | 50.88 |
| | ARC_easy | 70.96 | 67.02 | 35.17 | 34.57 | 31.84 | 32.00 | 31.62 |
| | ARC_challenge | 48.30 | 42.52 | 21.84 | 19.45 | 20.39 | 20.00 | 20.56 |
| **Read.** | OpenBookQA | 45.33 | 41.27 | 26.13 | 25.80 | 25.60 | 26.33 | 26.53 |
| | LAMBADA | 35.64 | 25.34 | 1.93 | 3.25 | 2.17 | 3.87 | 5.78 |
| | BoolQ | 75.37 | 66.25 | 51.66 | 47.29 | 40.81 | 45.62 | 47.69 |
| | SQuADv2_em | 10.62 | 0.10 | 0.14 | 0.02 | 0.02 | 0.01 | 0.00 |
| | SQuADv2_f1 | 38.28 | 22.83 | 1.33 | 2.61 | 1.36 | 0.57 | 1.13 |
| **Knl.** | NaturalQuestions | 5.44 | 4.51 | 0.06 | 0.00 | 0.02 | 0.01 | 0.05 |
| | TriviaQA | 12.34 | 12.77 | 0.05 | 0.01 | 0.01 | 0.02 | 0.01 |
| **Code** | HumanEval | 20.94 | 10.98 | 0.00 | 0.00 | 0.00 | 0.00 | 0.00 |
| | MBPP | 12.60 | 13.40 | 0.00 | 0.00 | 0.00 | 0.00 | 0.00 |
| **Math** | GSM8K | 7.52 | 7.78 | 0.00 | 0.00 | 0.00 | 0.00 | 0.00 |
| **Gen.** | MMLU | 43.07 | 39.45 | 26.26 | 26.16 | 25.85 | 25.79 | 25.38 |
| | BBH | 12.35 | 18.02 | 0.00 | 0.01 | 0.00 | 0.79 | 0.12 |
| **Avg. Performance Score(↓)** | | 37.57 | 33.24 | 17.57 | 17.21 | 16.49 | 17.00 | 17.22 |
| **Average Recovery Ratio(↓)** | | 63.67 | 56.32 | 29.77 | 29.15 | 27.93 | 28.81 | 29.17 |
| **Recovery Difficulty(↑)** | | 2.07 | 7.96 | 9.25 | 9.96 | 10.08 | 10.13 | 10.22 |

Table 44: Evaluation Results of Phi-1.5 under Different Closed-sourced Proportion

| | | 0.25% | 0.5% | 1% | 3% | 7% | 15% | 30% | 50% | 100% |
|---|---|---|---|---|---|---|---|---|---|---|
| **Rsn.** | PIQA | 68.21 | 68.37 | 69.68 | 65.85 | 53.43 | 52.94 | 52.36 | 51.25 | 50.44 |
| | Hellaswag | 49.05 | 49.18 | 49.30 | 30.72 | 26.27 | 26.74 | 27.02 | 26.10 | 25.05 |
| | Winogrande | 63.83 | 64.91 | 61.20 | 58.04 | 51.09 | 51.38 | 50.25 | 50.22 | 49.12 |
| | ARC_easy | 62.94 | 62.89 | 62.25 | 35.15 | 30.81 | 29.27 | 29.64 | 27.99 | 27.50 |
| | ARC_challenge | 36.98 | 37.49 | 32.91 | 25.97 | 20.56 | 20.36 | 20.08 | 20.88 | 21.22 |
| **Read.** | OpenBookQA | 39.07 | 40.20 | 35.00 | 33.87 | 26.60 | 27.67 | 27.73 | 26.47 | 26.87 |
| | LAMBADA | 24.71 | 24.99 | 25.36 | 0.11 | 0.59 | 0.78 | 1.15 | 0.06 | 0.00 |
| | BoolQ | 59.43 | 59.35 | 63.49 | 41.01 | 46.98 | 51.59 | 46.46 | 44.02 | 46.28 |
| | SQuADv2_em | 15.65 | 16.00 | 3.13 | 0.50 | 0.00 | 0.01 | 0.03 | 0.00 | 0.00 |
| | SQuADv2_f1 | 32.62 | 32.62 | 14.88 | 0.56 | 0.78 | 1.24 | 2.29 | 1.58 | 1.60 |
| **Knl.** | NaturalQuestions | 2.72 | 2.64 | 0.32 | 0.03 | 0.04 | 0.03 | 0.05 | 0.03 | 0.01 |
| | TriviaQA | 8.17 | 7.96 | 5.69 | 0.01 | 0.01 | 0.01 | 0.01 | 0.01 | 0.01 |
| **Code** | HumanEval | 14.43 | 13.41 | 2.03 | 0.00 | 0.00 | 0.00 | 0.00 | 0.00 | 0.00 |
| | MBPP | 17.20 | 18.67 | 6.47 | 0.00 | 0.00 | 0.00 | 0.00 | 0.00 | 0.00 |
| **Math** | GSM8K | 4.88 | 4.90 | 0.25 | 0.00 | 0.00 | 0.00 | 0.00 | 0.00 | 0.00 |
| **Gen.** | MMLU | 30.12 | 29.88 | 28.98 | 27.78 | 24.07 | 24.22 | 24.66 | 24.28 | 22.95 |
| | BBH | 4.34 | 3.19 | 0.98 | 0.50 | 0.00 | 0.00 | 0.00 | 0.00 | 0.00 |
| **Avg. Performance Score(↓)** | | 31.43 | 31.57 | 27.17 | 19.41 | 16.54 | 16.84 | 16.57 | 16.05 | 15.94 |
| **Average Recovery Ratio(↓)** | | 66.54 | 66.83 | 57.52 | 41.11 | 35.02 | 35.64 | 35.08 | 33.98 | 33.75 |
| **Recovery Difficulty(↑)** | | 6.18 | 6.15 | 2.76 | 9.28 | 10.08 | 11.19 | 10.54 | 10.23 | 11.26 |

Table 45: Evaluation Results of Phi-1.5 under Different Closed-sourced Quantity

|  |  | 20M | 50M | 100M | 160M | 200M | 300M | 600M |
|---|---|---|---|---|---|---|---|---|
| **Rsn.** | PIQA | 69.80 | 65.85 | 53.43 | 52.52 | 52.94 | 53.06 | 53.81 |
|  | Hellaswag | 49.51 | 25.72 | 30.27 | 26.31 | 26.74 | 27.05 | 26.51 |
|  | Winogrande | 62.56 | 58.04 | 51.09 | 50.83 | 51.38 | 50.57 | 49.99 |
|  | ARC_easy | 62.41 | 30.15 | 30.81 | 29.14 | 29.27 | 29.62 | 29.67 |
|  | ARC_challenge | 32.51 | 25.97 | 20.56 | 19.97 | 20.36 | 20.48 | 20.79 |
| **Read.** | OpenBookQA | 35.53 | 33.87 | 26.60 | 26.93 | 27.67 | 28.20 | 26.87 |
|  | LAMBADA | 28.14 | 0.11 | 0.59 | 0.45 | 0.78 | 1.30 | 0.61 |
|  | BoolQ | 64.77 | 41.01 | 46.98 | 47.33 | 51.59 | 46.09 | 45.59 |
|  | SQuADv2_em | 4.67 | 0.50 | 0.00 | 0.00 | 0.01 | 0.01 | 0.00 |
|  | SQuADv2_f1 | 22.47 | 0.56 | 0.78 | 1.02 | 1.24 | 2.31 | 2.01 |
| **Knl.** | NaturalQuestions | 1.64 | 0.03 | 0.04 | 0.05 | 0.03 | 0.06 | 0.03 |
|  | TriviaQA | 5.93 | 0.01 | 0.01 | 0.01 | 0.01 | 0.02 | 0.01 |
| **Code** | HumanEval | 7.73 | 0.00 | 0.00 | 0.00 | 0.00 | 0.00 | 0.00 |
|  | MBPP | 7.87 | 0.00 | 0.00 | 0.00 | 0.00 | 0.00 | 0.00 |
| **Math** | GSM8K | 0.28 | 0.00 | 0.00 | 0.00 | 0.00 | 0.00 | 0.00 |
| **Gen.** | MMLU | 31.11 | 27.78 | 24.07 | 23.41 | 24.22 | 24.54 | 24.68 |
|  | BBH | 3.38 | 0.50 | 0.00 | 0.00 | 0.00 | 0.00 | 0.00 |
| **Avg. Performance Score(↓)** |  | 28.84 | 19.89 | 16.54 | 16.35 | 16.84 | 16.67 | 16.50 |
| **Average Recovery Ratio(↓)** |  | 61.06 | 41.11 | 35.02 | 34.61 | 35.64 | 35.28 | 34.94 |
| **Recovery Difficulty(↑)** |  | 2.81 | 9.28 | 10.26 | 11.65 | 11.19 | 10.87 | 10.49 |

Table 46: Evaluation Results of OPT-350M under Different Closed-sourced Layers. "*" indicates the fully closed-sourced model.

|  |  | Pretrain | 0-2 | 3-5 | 6-8 | 9-11 | 12-14 | 15-17 | 18-20 | 21-23 | 24-26 | 27-29 | 30-32 | 33-35 | * |
|---|---|---|---|---|---|---|---|---|---|---|---|---|---|---|---|
| **Rsn.** | PIQA | 64.69 | 61.40 | 62.50 | 61.11 | 56.46 | 58.47 | 58.94 | 61.86 | 62.59 | 63.13 | 61.93 | 62.67 | 63.11 | 49.53 |
|  | Hellaswag | 36.68 | 34.03 | 34.27 | 33.69 | 31.79 | 32.24 | 32.78 | 33.27 | 33.68 | 33.25 | 33.94 | 33.63 | 33.07 | 25.77 |
|  | Winogrande | 52.09 | 51.62 | 52.96 | 51.57 | 50.83 | 52.83 | 51.06 | 51.52 | 51.85 | 52.06 | 52.04 | 51.99 | 50.94 | 49.85 |
|  | ARC_easy | 44.02 | 40.46 | 40.66 | 40.07 | 35.41 | 37.50 | 37.81 | 39.91 | 40.70 | 41.12 | 41.19 | 40.84 | 39.92 | 26.53 |
|  | ARC_challenge | 20.82 | 22.27 | 22.61 | 21.25 | 21.39 | 21.25 | 22.01 | 21.36 | 20.25 | 20.48 | 19.88 | 20.99 | 20.28 | 19.82 |
| **Read.** | OpenBookQA | 28.00 | 27.60 | 27.47 | 27.40 | 27.40 | 27.27 | 26.20 | 26.47 | 27.67 | 26.80 | 28.67 | 27.67 | 27.13 | 27.47 |
|  | LAMBADA | 40.47 | 30.62 | 32.97 | 28.62 | 21.65 | 23.87 | 28.23 | 29.07 | 29.83 | 29.81 | 31.72 | 31.43 | 18.08 | 0.00 |
|  | BoolQ | 57.74 | 50.87 | 48.51 | 50.58 | 51.60 | 52.83 | 53.42 | 54.37 | 53.30 | 51.42 | 59.79 | 53.14 | 60.42 | 37.83 |
|  | SQuADv2_em | 11.34 | 6.87 | 7.88 | 4.74 | 4.19 | 0.27 | 0.87 | 2.22 | 3.79 | 3.05 | 4.11 | 4.69 | 2.35 | 0.00 |
|  | SQuADv2_f1 | 19.35 | 16.27 | 17.00 | 12.00 | 11.72 | 9.04 | 6.92 | 10.11 | 10.90 | 10.08 | 8.88 | 11.47 | 7.30 | 0.01 |
| **Knl.** | NaturalQuestions | 1.08 | 1.05 | 0.83 | 0.83 | 0.78 | 0.55 | 0.69 | 0.41 | 1.00 | 0.85 | 0.71 | 0.52 | 0.75 | 0.04 |
|  | TriviaQA | 4.48 | 2.24 | 2.66 | 2.01 | 2.16 | 1.41 | 1.06 | 2.39 | 2.38 | 2.29 | 1.57 | 1.90 | 1.76 | 0.02 |
| **Code** | HumanEval | 0.00 | 0.00 | 0.00 | 0.00 | 0.00 | 0.00 | 0.00 | 0.00 | 0.00 | 0.00 | 0.00 | 0.00 | 0.00 | 0.00 |
|  | MBPP | 0.00 | 0.00 | 0.00 | 0.00 | 0.00 | 0.00 | 0.00 | 0.00 | 0.00 | 0.00 | 0.00 | 0.00 | 0.00 | 0.00 |
| **Math** | GSM8K | 1.59 | 0.15 | 0.25 | 0.00 | 0.08 | 0.00 | 0.05 | 0.00 | 0.03 | 0.18 | 0.00 | 0.00 | 0.00 | 0.00 |
| **Gen.** | MMLU | 26.05 | 25.52 | 26.02 | 25.20 | 25.60 | 25.05 | 25.73 | 23.97 | 25.57 | 26.04 | 25.13 | 25.17 | 25.73 | 22.95 |
|  | BBH | 16.97 | 6.87 | 12.58 | 5.51 | 5.11 | 2.55 | 5.98 | 2.74 | 11.15 | 13.98 | 14.25 | 13.02 | 12.57 | 0.00 |
| **Avg. Performance Score(↓)** |  | 25.02 | 22.23 | 22.89 | 21.44 | 20.36 | 20.30 | 20.69 | 21.16 | 22.04 | 22.03 | 22.58 | 22.30 | 21.38 | 15.28 |
| **Average Recovery Ratio(↓)** |  | - | 88.83 | 91.49 | 85.71 | 81.38 | 81.13 | 82.69 | 84.55 | 88.08 | 88.05 | 90.23 | 89.13 | 85.43 | 61.08 |
| **Recovery Difficulty(↑)** |  | - | 5.92 | 9.32 | 9.04 | 8.60 | 8.83 | 8.73 | 7.17 | 5.82 | 5.18 | 4.65 | 4.92 | 4.15 | 10.89 |

