# OpenReview forum: "Archilles' Heel in Semi-open LLMs: Hiding Bottom against Recovery Attacks"
_ICLR.cc/2025/Conference — ICLR 2025 Conference Withdrawn Submission_

### Official Review · Reviewer_ag9Z · 2024-10-24

**Soundness:** 3
**Presentation:** 3
**Contribution:** 4
**Rating:** 8
**Confidence:** 4

**Summary:**

This paper presents SCARA, a method to identify decoder layers to hide in decoder-only LLMs. The authors provide theoretical proof of transition layers, where early errors are amplified in subsequent layers. They introduce RD to assess post-recovery performance when specific layers are hidden. Experiments show that SCARA, by hiding only a few layers, achieves a recovery ratio close to baselines while maintaining customization performance similar to fully open approach. The experiments also confirm the existence of transition layers in the models.

**Strengths:**

1. The paper is well-written, featuring thorough experiments and clear explanations from both theoretical and empirical perspectives.
2. The overall layout is visually pleasing, and the figures are diverse and effectively illustrate the content, aiding readers' understanding.
3. It proposes a straightforward and effective method for constructing a semi-open model by hiding only a few layers, while achieving baseline-level resilience and customization performance comparable to the fully open setting.
4. The insight regarding the existence of a transition layer contributing to resilience is particularly compelling, with a detailed theoretical explanation that enhances understanding.
5. The authors provide comprehensive empirical validation across multiple architectures and benchmarks, covering models of various sizes (1.3B-70B), and testing customizability and recovery performance on several benchmarks. They also conducted experiments on recovery datasets of different sizes, demonstrating sufficient experimental rigor.
6. The authors proposed additional enhancements to the original baseline, strengthening the protection of baseline SAP and highlighting SCARA’s effectiveness in preserving resilience.
7. The authors empirically validated their theory of the transition layer’s existence and pointed out that smaller models exhibit transition layers earlier than larger models.
8. The authors clearly identified the limitations of the SCARA method, noting its ineffectiveness on small models (OPT-350M) and its inability to defend against other adversary attacks.
9. The proposed SCARA algorithm has clear practical applications, offering a viable solution for enhancing the customizability of semi-open models while preserving comparable resilience.

**Weaknesses:**

1. One mathematical notation in Section 4.2 is unclear. The loss function $\ell$ for RD(I) is not specified, making it confusing.
2. Figures 1 and 2 have minimal captions and small text, reducing readability and limiting their ability to convey insights.

**Questions:**

1. Is the "attack datasets" mentioned in Figure 2 the same as the "recovery datasets" discussed later in the paper?
2. Could you clarify the formula and the loss function used for RD(I)?
3. Could you clarify how fully-closed and semi-open models differ in practice?
4. Could you explain more about the distinctions between FT-all, FT-closed, and SEM in Section 5.1?
5. Can the row and column headers in the tables be made clearer by avoiding abbreviations?
6. Could you explain more about potential future work that could be included in the paper?
7. Could the authors clarify what the value 0.00 represents in Table 1 and Table 2?
8. The authors discussed the impact of datasets of different lengths on the effectiveness of SCARA in the experimental section, but these datasets did not appear in the setup. Could the authors provide a detailed introduction to the composition of these datasets?

---

> ### Author Response · Authors · 2024-11-18
> **Response to Reviewer ag9Z**
>
> Thanks for your time and valuable comments. We are pleased that you acknowledged the novelty and significance of our focused problem, promising results and straightforward presentation. In the response below, we provide answers to your questions in order to address the concerns and increase your confidence.
>
> **Q1: Is the "attack datasets" mentioned in Figure 2 the same as the "recovery datasets" discussed later in the paper?**
>
> **R1:** Thank you for your insightful comments and for highlighting this inconsistency. The terms "attack datasets" and "recovery datasets" refer to the same dataset, which is used to facilitate model recovery attacks aimed at replicating victim model’s behavior. We sincerely appreciate your feedback and have revised Figure 2 in the manuscript to ensure consistent terminology is used throughout.
>
> **Q2: Could you clarify the formula and the loss function used for RD(I)?**
>
> **R2:** Thank you for your comment on clarifying RD. We have revised the definition and implementation details in Sections 4.2 and 5.1. Below is the detail formulation and explanation of RD:
>
> The Recovery Difficulty (RD) quantifies the challenge of recovering the closed-source module and is defined as:   $\text{RD}(I) = \mathbb{E}_{\mathbf{X},Y,{\theta_0}(I)}\left[\ell\left(f(\mathbf{X};{\theta}_0(I)), Y\right)\right]$
>
> Where:
>
> \- $I$: The set of indices for the closed-source layers, indicating which hidden layers are kept private.
>
> \- **${\theta}_0(I)$**: The initial parameters of the replacement model. Parameters for hidden layers are randomly initialized, while parameters for public layers remain unchanged.
>
> \- $\mathbf{X}$: Input features sampled to target general capabilities from the underlying distribution.
>
> \- $Y$: Labels corresponding to \(\mathbf{X}\).
>
> \- $f$: The final output of the semi-open model.
>
> \- $\ell$: The loss function, where we use cross-entropy loss in this paper.
>
> \- $\mathbb{E}$: The expectation over the joint distribution of random inputs, labels, and randomly initialized
>
> We estimate RD using 1,500 samples drawn from two diverse datasets: the MMLU benchmark and Alpaca 52k. These datasets cover tasks such as text comprehension, summarization, generation, code writing, and mathematical reasoning. Additional details are provided in Section 5.1 and Appendix B.4.  Moreover, hidden layers for different closed-source sets are randomly initialized using Xavier initialization with PyTorch's default settings. In this study, the RD is averaged over three random seeds (20, 42, 1234) to ensure robustness.
>
> **Q3: Could you clarify how fully-closed and semi-open models differ in practice?**
>
> **R3:** Thank you for your thoughtful feedback. Fully closed models provide strong vendor control but significantly limit user customization options. For example, users must rely on fine-tuning APIs provided by vendors, which handle the fine-tuning process using vendor-controlled computational resources. In this setup, users do not have access to the model's internal parameters, restricting their ability to perform detailed customization.
>
> In contrast, semi-open models strike a balance between customizability and robustness against recovery attacks. This approach allows vendors to retain control over proprietary components, secure revenue streams, and reduce the computational burden of fine-tuning. Meanwhile, users benefit from the flexibility to customize open-source modules offline, optimizing them for specific tasks. We hope this explanation clarifies our motivations, as outlined in our response to R1 in the general response section.
>
> **Q4: Could you explain more about the distinctions between FT-all, FT-closed, and SEM in Section 5.1?**
>
> **R4:** Thank you for your thoughtful comments. In the R2 of our general response, we have provided a detailed explanation of the distinctions among the three strategies. These distinctions enable a more comprehensive evaluation of SCARA's effectiveness under our proposed threat model. Additionally, we have revised the manuscript to further clarify these three strategies in our threat model. Thank again for your constructive suggestions.
>
> **Q5: Can the row and column headers in the tables be made clearer by avoiding abbreviations?**
>
> **R5:** Thank you for your suggestion. While we aim to maintain the table’s length and visual clarity, we understand the importance of readability. Following your suggestion, we carefully revise the manuscript to minimize abbreviations where possible and improve clarity.

---

> > ### Author Response · Authors · 2024-11-18
> > **Response to Reviewer ag9Z**
> >
> > **Q6: Could you explain more about potential future work that could be included in the paper?**
> >
> > **R6:** Thank you for your comment. Our current work focuses on constructing Semi-open models to prevent recovery of general abilities but does not explore protecting specific domain abilities. Additionally, our method does not yet accurately identify transition layers or determine the optimal number of parameters to protect. Future work will address these limitations by focusing on specific domain protection, identifying transition layers, and optimizing parameter protection strategies.
> >
> >
> >
> > **Q7: Could the authors clarify what the value 0.00 represents in Table 1 and Table 2?**
> >
> > **R7:** Thank you for your insightful comment. In Tables 1 and 2, the value "0.00" signifies that the model's performance on the corresponding benchmark or domain is entirely lost, achieving a score of 0 on a scale of 0 to 100. Following your valuable feedback, we have further clarified the meaning of these values in the captions to ensure better understanding.
> >
> >
> >
> > **Q8: The authors discussed the impact of datasets of different lengths on the effectiveness of SCARA in the experimental section, but these datasets did not appear in the setup. Could the authors provide a detailed introduction to the composition of these datasets?**
> >
> > **R8:** Thank you for your insightful comment. The extensive datasets used to evaluate SCARA’s effectiveness are extensions of the 51K attack dataset. Specifically, we construct the 100K, 200K, 300K, and 500K datasets incorporating additional sources, including Baize (158K multi-turn conversations from ChatGPT self-chat), MathInstruct (260K curated mathematical instruction instances), and OpenOrca (1M GPT-4 completions and 3.2M GPT-3.5 completions). These supplementary datasets enhance the attack by supporting complex tasks and providing broader topic coverage. For further details, please refer to Section 5.1 and Appendix B.2 of the paper. Following your suggestion, we have revised the manuscript to clarify the datasets used in our analysis.
> >
> >
> >
> >
> >
> > **Thanks for your kind and helpful comments and we are looking forward to discussing with you to further improve our paper!**

---

> ### Comment · Reviewer_ag9Z · 2024-11-20
>
> Thank you for the detailed and comprehensive responses, which have clarified my concerns and provided a better understanding of SCARA.
>
> After reviewing other feedback and the authors’ responses, I like the thoughtful revisions, which have notably improved the paper.
> (1) The paper effectively highlights the rise of semi-open models, where closed-source embedding models integrate with open-source modules.  In  your response to Reviewer KKBu, the distinction between 'semi-open pipeline' and 'semi-open model' is not crucial. What matters is the understanding and application of the concept.
>
> (2) SCARA addresses performance challenges in semi-open models by partitioning pretrained LLMs into closed-source and open-source components. This design balances customizability with enhanced resilience to recovery attacks, benefiting both vendors and users.
>
> (3)According to the general response, the threat model discussed is well-recognized within LLM communities. As highlighted in your paper, there is substantial research on model recovery and model extraction attacks, indicating a significant interest and concern in these areas.
>
> (4) I agree with Reviewers d5Gh and 1rG5, the analysis of a transition layer is particularly compelling and may have broad implicatio
> ns for the research community. This insight helps us estimate the minimal number of hidden layers required without fine-tuning.
>
> This paper provides a well-structured, insightful contribution of broad interest. The SCARA framework balances customizability with resilience to recovery attacks, enhancing semi-open models. I recommend to accept.

---

> > ### Author Response · Authors · 2024-11-22
> > **Official Response to Reviewer ag9Z**
> >
> > We deeply appreciate your recognition of the motivation behind our semi-open model framework and the threat model we proposed. We would like to further clarify our research intentions and their broader significance.
> >
> > As highlighted by OpenAI in its response to the NTIA’s (National Telecommunications and Information Administration) request regarding LLMs with open weights: “When we did not observe significant misuse effects, this gave us the confidence to openly release the full model weights” [1]. This statement underscores the persistent risks of misuse, which have driven many LLM vendors to prefer releasing models through black-box APIs to mitigate these challenges. Unfortunately, as noted by OpenAI and others [2], such an approach often restricts downstream customizability. Addressing the tension between maintaining control and promoting openness has been a key motivation for our exploration of a semi-open model design.
> >
> > In our work, we propose a selective closed-sourcing framework that seeks to balance controlled risk management with enhanced opportunities for downstream adaptation. By keeping only the early layers of the model closed-source while making the remaining layers accessible, this approach aims to address vendor concerns about misuse while empowering users with greater customizability. To the best of our knowledge, few prior studies have explicitly tackled this challenge. We see our framework as a preliminary step toward inspiring more open practices among LLM vendors, mitigating potential risks, and enabling broader access to powerful, adaptable models. We hope this approach supports academic research and diverse applications, fostering meaningful collaboration and innovation across domains.
> >
> > Once again, we sincerely thank you for your valuable feedback, which has been instrumental in refining our work. Thank you for your thoughtful review and consideration.
> >
> > [1] OpenAI’s comment to the NTIA on open model weights
> >  https://openai.com/global-affairs/openai-s-comment-to-the-ntia-on-open-model-weights/
> >
> >  [2] Open-Source LLM vs Closed Source LLM for Enterprise Use-Cases
> >  https://www.fluid.ai/blog/open-source-llm-vs-closed-source-llm-for-enterprise-use-cases

---

> > ### Author Response · Authors · 2024-11-27
> > **Official Comment by Authors**
> >
> > Dear Reviewer ag9Z:
> >
> > We sincerely thank you for your recognition of the motivation behind our semi-open model framework. However, as pointed out by several other reviewers, our motivation does indeed face concerns regarding its practicality. Therefore, in our general response 2, we have provided further clarification of our motivation and made the necessary revisions to the manuscript. We would greatly appreciate it if you could review our response at your earliest convenience. We respectfully request that you reconsider our manuscript and kindly revise your score.
> >
> > Regards,
> >
> > Authors

---

> > ### Author Response · Authors · 2024-12-01
> > **Humbly Seeking Further Discussion**
> >
> > Dear Reviewer ag9Z,
> >
> > We kindly note that the author-reviewer discussion period is currently ongoing and approaching its conclusion. We would be grateful if you could review our latest general response (General Response 2) at your earliest convenience, where we have provided additional clarification regarding our **motivation**, particularly with respect to the **“semi-open model.”** We sincerely hope that this will futher clarify the contribution of our manuscript and address your concerns. We humbly request that you reconsider our manuscript and, if appropriate, kindly revise your score. Thank you very much for your time and thoughtful review.
> >
> > Best regards,
> >
> > The Authors

---

> ### Comment · Reviewer_ag9Z · 2024-12-03
> **Official Comment by Reviewer ag9Z**
>
> I would like to  thank the authors for their effort in rebuttal and their thoughtful response, which further clarifies their practical motivation. In the initial review, I found the overall structure of the paper to be coherent and the proposed theoretical contributions to be highly innovative, which led me to assign a score of 8. However, as noted by other reviewers, the inclusion of embedding models introduced some confusion. The revised scenario of on-premises deployment is more aligned with my expectations, and I appreciate that the authors have now focused on this setting. Given that, to the best of my knowledge, there has been no prior work that addresses this issue both theoretically and empirically, I continue to strongly recommend the acceptance of this paper.

---

> ### Author Response · Authors · 2024-12-04
>
> Dear Reviewer ag9Z:
>
> We sincerely appreciate your kind and positive response. We are truly grateful for your recognition and support of our work and contributions. Furthermore, we deeply value the effort you have dedicated during the rebuttal process. Thank you once again.
>
> Best Regards,
>
> Authors

---

### Official Review · Reviewer_1rG5 · 2024-11-03

**Soundness:** 3
**Presentation:** 3
**Contribution:** 2
**Rating:** 6
**Confidence:** 4

**Summary:**

This paper studies the problem of how to design semi-open models (i.e., models whose weights are only partially open-sourced) that can simultaneously also be resilient to recovery attacks. The paper finds that a transition layer exists, such that even small recovery errors in layers before this layer can lead to recovery failure. Building on these insights, the paper proposes an approach called SCARA that keeps only a few bottom layers as closed-source. With this new approach, the paper shows that it is possible to improve downstream customization performance while maintaining similar resilience.

**Strengths:**

1. The paper is well-written, and the presentation is clear and clean.

2. The approach is well motivated --- it first starts from an empirical observation that closed-sourcing the first two layers offers significantly greater resilience than the last two, while the model shares similar customizability under the two cases. This implies that close sourcing the later layers may be the optimal solution for keeping the resistance to recovery attacks. The paper subsequently further formally establishes this finding with rigorous theoretical analysis, showing the existence of a transition layer such that even small recovery errors in layers before this layer can lead to recovery failure. This also intuitively makes sense --- when the attacker is asked to recover the earlier layers as opposed to the later layers, the errors in early layers will be amplified by later layers. This asymmetry is natural.

3. Based on this insight, the paper also proposes an effective approach for the selectively closed-sourcing model to defend against recovery attacks. The experiment results support the effectiveness of the approach.

Overall, the paper is nicely done.

**Weaknesses:**

One outstanding weakness of this paper is that the threat model considered may not be practically relevant. It seems the authors coined the semi-open model's scenario, that seems not really exist in the real world.

Currently, the most common setups are either open-source or closed-source. For close-source context, when developers do want their users to customize their models, the standard practice is to deploy fine-tuning APIs (e.g., https://platform.openai.com/docs/guides/fine-tuning) rather than partially open-source a part of the model. It seems to make no sense to only open-source the first few layers of a model to enable customization. Because the customization anyway still needs the involvement of the closed-source developers --- so they can fine-tune and connect the first few layers and the later layers to really deploy the model. Then, why not just close-source all weights and directly ask the users to upload custom data, and then the closed-source developers fine-tune and deploy the model for the users, like what is being done in fine-tuning APIs?

I worry that if not developers will do the partial open-sourcing like the authors of this paper consider, then the problem itself may not hold.

**Questions:**

Can the authors explain and clarify why the semi-open models are practically relevant?

---

> ### Author Response · Authors · 2024-11-18
> **Response to Reviewer 1rG5**
>
> Thanks for your time and valuable comments. We are pleased for your positive comments. In the response below, we provide answers to your questions in order to address the concerns and increase your confidence.
>
> **Q1:**  **Why the semi-open models are practically relevant?**
>
> **R1:** Thank you for your positive, constructive and valuable comments. We appreciate the reviewer’s feedback on the concern of semi-open models' practically relevant. We outlined the advantage of semi-open models in the general response section. We hope our response provides a clearer understanding of the motivations behind our approach.
>
>
>
> **Q2:  Threat models are unclear.**
>
> **R2：** Thank you for your insightful feedback. We have detailed the threat model in general response **R2** and further refined Section 3.1. Your suggestions have greatly improved the clarity of our paper.
>
>
>
> **Thanks for your kind and  helpful comments and we are looking forward to discussing with you to further improve our paper!**

---

> ### Author Response · Authors · 2024-11-20
> **Official Comment by Authors**
>
> Dear Reviewer 1rG5:
>
> Kindly note that the author-reviewer discussion period is currently ongoing. We would greatly appreciate it if you could review our response when convenient. We earnestly request that you reconsider our manuscript and consider upgrading your score.
>
> Regards,
>
> Authors

---

> > ### Comment · Reviewer_1rG5 · 2024-11-25
> > **Response to authors**
> >
> > I would like to thank the authors for their response. My concern regarding the practical relevance of the problem considered here still holds. I also read the reviews by Reviewer d5Gh & kKBu, and it seems that they also share the same concern, and I agree with them.
> >
> > Regarding the authors' rebuttal, the authors cited the embedding model from companies as an example to support the practicality of the problem here. This is a smart idea, but it can be easily disputed. By "embedding models," they inherently already choose to close-source the first multiple layers of these models --- that the paper suggests doing. And for these embedding models, there are no such things as open-sourcing the later layers of the model. Moreover, it anyway makes no sense to only open-source the later layers of a model in any existing settings. This makes the main conclusion from this paper seem to be trivial.
> >
> > Overall, I think this paper is quite standard in terms of the conduct of the study & the writing. So, I initially gave a 6.
> > Having said that, the problem itself studied in this paper is not a very valid and convincing problem. I will keep my current rating, but I won't strongly recommend acceptance.

---

> > > ### Author Response · Authors · 2024-12-01
> > > **Humbly Seeking Further Discussion**
> > >
> > > Dear Reviewer 1rG5,
> > >
> > > Kindly note that the author-reviewer discussion period is currently ongoing and is approaching its conclusion. We would greatly appreciate it if you could review our latest general response (General Response 2) at your earliest convenience, where we have provided additional clarification on our **motivation**, particularly regarding the **“semi-open model”**. We sincerely hope this can address your concerns. We humbly request that you reconsider our manuscript and, if appropriate, consider upgrading your score. Thank you for your time and thoughtful review.
> > >
> > > Best regards,
> > >
> > > The Authors

---

> > > > ### Comment · Reviewer_1rG5 · 2024-12-02
> > > >
> > > > Dear Authors,
> > > >
> > > > I did read the general response. I agree with Reviewer kKBu's opinion here --- also, I have provided a similar point earlier:
> > > > > By "embedding models," they inherently already choose to close-source the first multiple layers of these models --- that the paper suggests doing. And for these embedding models, there are no such things as open-sourcing the later layers of the model. Moreover, it anyway makes no sense to only open-source the later layers of a model in any existing settings.
> > > >
> > > > Embedding models are also by no means for defending model-stealing attacks — they usually just output the embeddings of the last layer of the model.
> > > >
> > > > Overall, as many other reviewers suggested --- the threat model in this paper is not convincing. This impression still holds, even given the authors' arguments.

---

> > > > > ### Author Response · Authors · 2024-12-03
> > > > > **Official Comment by Authors**
> > > > >
> > > > > Dear reviewer 1rG5:
> > > > >
> > > > > We sincerely thank you for your time and valuable comments. We would like to take this opportunity to clarify the deployment scenarios of semi-open models and address your concerns regarding our threat model.
> > > > >
> > > > > We acknowledge your point that framing the closed-source module as an embedding model may trivialize the problem, as embedding models typically start from the first layer and do not involve partial public exposure. To reduce the misunderstandings of the novelty and significance of our contribution, we have removed all references to embedding models in the revised manuscript.
> > > > >
> > > > > Furthermore, we believe that discussing semi-open model design in the context of on-premises deployment is more appropriate. In local deployment scenarios, vendors face the challenge of balancing flexibility with the risk of model theft [1-7]. Due to the limited memory and processing speed of hardware security methods, the semi-open model approach has been widely explored [1-3,6-7]. For instance, a 2020 MobiSys paper (with 195 citations) [2] states: *“Due to the limited memory of the edge device’s TEE, we partition model layers into more sensitive layers (executed inside the device’s TEE) and layers executed in the untrusted part of the operating system.”* More recent work by Song Han [6] further explores the balance between model customization flexibility and security, as stated in the abstract: *“In this paper, we propose Offsite-Tuning, a privacy-preserving and efficient transfer learning framework that adapts billion-parameter foundation models to downstream data without access to the full model.”* However, despite these efforts, no principled theory exists to determine which layers should be concealed for optimal security and customization. To the best of our knowledge, our work is the first to theoretically address this gap. We have also revised our manuscript to clarify our motivation.
> > > > >
> > > > > Regarding our threat model, model distillation and recovery attacks have been widely studied in local deployment, where attackers query the hidden module and attempt to replicate its functionality by training a substitute model. This attack strategy has been discussed in recent top-tier security conferences, such as  Hai "Helen" Li’s paper [8] (USENIX '24), which states: “*Query-based model extraction attacks aim at learning a substitute model with the predictions returned by a black-box target model*”.
> > > > >
> > > > > In this paper, we first theorectically prove the existance of transition layer in LLMs, demonstrating that hiding layers before this transition enhances resistance to recovery attacks. Based on this, we propose SCARA, an effective and efficient method that identifying only a few bottom layers to conceal with a fine-tuning-free metric, enabling effective fine-tuning while safeguarding against recovery risks.
> > > > >
> > > > > Thank you again for your constructive and helpful comments. We hope our response can addresses your concerns.
> > > > >
> > > > > Best Regards,
> > > > >
> > > > > Authors
> > > > >
> > > > >
> > > > >
> > > > > [1] TEESlice: Protecting Sensitive Neural Network Models in Trusted Execution Environments When Attackers have Pre-Trained Models. https://arxiv.org/pdf/2411.09945
> > > > >
> > > > > [2] Darknetz: towards model privacy at the edge using trusted execution environments http://arxiv.org/abs/2004.05703
> > > > >
> > > > > [3] Securing AI Model Weights https://www.rand.org/pubs/research_reports/RRA2849-1.html
> > > > >
> > > > > [4] Deepsniffer: A dnn model extraction framework based on learning architectural hints https://dl.acm.org/doi/pdf/10.1145/3373376.3378460
> > > > >
> > > > > [5] DNN model architecture fingerprinting attack on CPU-GPU edge devices. https://ieeexplore.ieee.org/abstract/document/9797366
> > > > >
> > > > > [6] Offsite-Tuning: Transfer Learning without Full Model https://arxiv.org/pdf/2302.04870
> > > > >
> > > > > [7] SoK: All You Need to Know About On-Device ML Model Extraction - The Gap Between Research and Practice https://www.usenix.org/system/files/usenixsecurity24-nayan.pdf
> > > > >
> > > > > [8] MODELGUARD: Information-Theoretic Defense Against Model Extraction Attacks https://www.usenix.org/system/files/sec24summer-prepub-409-tang.pdf

---

> ### Author Response · Authors · 2024-11-27
> **Official Comment by Authors**
>
> Dear Reviewer 1rG5:
>
> We sincerely thank you for raising your concerns regarding the practicality of the motivation behind the problem we addressed. In our general response 2, we have provided further clarification of our motivation and revised the manuscript accordingly. We would greatly appreciate it if you could review our response at your earliest convenience. We respectfully request that you reconsider our manuscript and kindly revise your score.
>
> Regards,
>
> Authors

---

### Official Review · Reviewer_kKBu · 2024-11-04

**Soundness:** 2
**Presentation:** 2
**Contribution:** 1
**Rating:** 3
**Confidence:** 5

**Summary:**

The paper proposes a method for identifying which layers in the model that, if recovered by an adversary in an iterative layer recovery process, will make subsequent layers easier to recover.

**Strengths:**

The evaluation seems comprehensive. It was easy to follow the problem setup and the method.

**Weaknesses:**

I question the threat model that the authors are introducing. I don't think there's any chance of stealing an LLM through any method that exists no matter how semi-closed/semi-open it is. The only methods that have been proposed that can do something like this specifically target the embedding layer.

It seems like the main insight of the paper, that hiding the earlier layers in the model is more impactful than hiding later layers because if an attacker wants to recover the model they'll pay an accuracy error scaling in the depth of the model past the layer they haven't yet recovered, is trivial. If you asked someone who had never heard anything about this literature of hiding layers, whether they should hide the first block or the last block, I'm certain everyone would choose to hide the first block. There's plenty of work already showing that later layers are more or less redundant and don't learn anything new. This is because attention heads in block N have the ability to learn Nth order interactions, but for N > 2, these interactions typically don't get learned and the attention heads just degenerate [1].

The actual implementation of the method is not sophisticated. It just takes this straightforward insight and turns it into a metric. But that metric is itself just "what happened if I closed the first N layers of the model" and then returns the first one that passes some threshold of difficulty.

It doesn't seem like the evaluation is really fair. The authors evaluate against SEM. But SEM just wants to recover the embedding and the authors are trying to show what happens if they hide the early parts of the network. This seems like an indication that this isn't a particularly realistic threat model.

[1] https://arxiv.org/abs/2404.08634

**Questions:**

n/a

---

> ### Author Response · Authors · 2024-11-18
> **Response to Reviewer kKBu**
>
> Thanks for your time and valuable comments. In the response below, we provide answers to your questions in order to address the concerns and increase your confidence.
>
> **Q1: Questions on the threat model.**
>
> **R1:** Thank you for your thoughtful comments and concerns about the threat model. We appreciate your mention of work related to stealing parts of the embedding layer in LLMs [1]. In this paper, we focus on a different type of threat model, known as model recovery or model extraction attacks. Since 2016, researchers have studied how to extract parameters and structures from black-box encoders through these attacks [2]. These typically involve querying the black-box model, collecting input-output pairs, and training a replacement model to replicate the original's behavior.
>
> In our design, the closed-source component acts as a black-box encoder, generating hidden representations for input data. As such, we believe the closed-source component in our design is vulnerable to model recovery attacks. To address this, we investigate how hiding certain layers can help defend against such threats. Further clarification is provided in the general response R2, and we hope this addresses and alleviates your concerns.
>
> In response to the question regarding the potential for stealing a semi-open LLM, we address this concern in Section 5.3 of our paper. Our findings suggest that poorly designed semi-open models are indeed vulnerable to recovery attacks. For instance, in our experiments, designating a later decoder layer (e.g., the 29th layer in Llama2-7B) as the closed-source component while open-sourcing the remaining layers substantially increased the model's susceptibility to recovery. Specifically, when only the 29th layer was hidden, we observed an average recovery ratio approaching 100% under recovery attacks. This indicates that the recovered model could closely replicate the victim model’s behavior across various functionalities. These results highlight that without careful design, the risk of stealing a semi-open LLM remains significant.
>
> **Q2: The main insight of the paper is trivial.**
>
> **R2** Thank you for your detailed feedback. We appreciate the opportunity to clarify and expand on the contributions of our work.
>
> First, we would like to highlight that the question of "hiding the first layer versus the last layer" is not as trivial as it may appear. Prior work, such as SAP [3], has studied a similar problem, but they open-source the bottom six layers, and keep the remaining layers closed-source. They empirically investigated the customizability of this design for downstream tasks. However, in this paper, we demonstrate that such heuristic is not optimal for balancing customizability and resilience against recovery attacks.
>
> Second, the primary message of our theorem is not simply that "hiding earlier layers is better than hiding later layers." Instead, our theorem establishes the existence of a **transition layer**—a critical point in the model such that hiding layers before this transition offers strong resilience against recovery attacks, while hiding layers after it does not. The example of "hiding the first layer versus the last layer" was included for simplicity and to make the idea more accessible, but it represents just one instance of our broader result. To the best of our knowledge, this is the first work to both theoretically and empirically identify such a transition layer and rigorously prove its existence in the context of defending against recovery attacks. Our proof demonstrates the crucial role of the bottom layers in providing resilience, making them a key focus for effective defenses. Moreover, we introduced several novel techniques in our proof process that, to the best of our knowledge, have not been explored before. These techniques may have broader implications and could benefit the research community beyond this specific context.
>
> Finally, we appreciate the reviewer pointing out related empirical observations about the redundancy of later layers and their limited role in learning higher-order interactions. While this is not directly related to our main theorem, we believe our theoretical findings provide valuable insights into understanding LLMs in scenarios beyond defending against recovery attacks. We thank the reviewer for highlighting the potential broader implications of our results.
>
> [1] Stealing Part of a Production Language Model http://arxiv.org/abs/2403.06634
>
> [2] Stealing Machine Learning Models via Prediction APIshttps://arxiv.org/pdf/1609.02943
>
> [3] A Split-and-Privatize Framework for Large Language Model Fine-Tuninghttp://arxiv.org/abs/2312.15603

---

> ### Author Response · Authors · 2024-11-18
> **Response to Reviewer kKBu**
>
> **Q3: The actual implementation of the method is not sophisticated. It just takes this straightforward insight and turns it into a metric.**
>
> **R3:** Thank you for your detailed feedback. Our goal is to predict the recovery ratio, which evaluates: "If the first $N$ layers of the model are hidden, to what extent can adversaries recover the model's general capabilities?" As discussed in Section 3.2, directly calculating this metric is highly computationally intensive. This process requires constructing a large query set and performing extraction attacks, which involve fine-tuning the entire semi-open model to replicate the original model for each possible hidden module configuration.
>
> To address this challenge, we aimed to develop a metric that avoids fine-tuning while being highly correlated with the recovery ratio. This task is non-trivial because the score function used to measure recovery (e.g., accuracy in a reading comprehension task) can be non-differentiable.
>
> To overcome this, we used the cross-entropy loss as a differentiable surrogate metric, which is negatively correlated with the score function. Using Taylor expansion and theoretical insights from prior work, we leveraged the observation that for large neural networks relative to the dataset size, the difference between the fine-tuned parameters $ \|{\theta}_{\text{FT}}(I,\mathcal{D}) - {\theta}_0(I)\|_2 $ is minor. Previous research has shown that for models like single-layer ReLU networks, this difference is of the order $\mathcal{O}\left(\frac{|\mathcal{D}|}{\sqrt{N}}\right)$, where $N$ is the number of model parameters, which is significantly larger than the dataset size in large language models.
>
> By incorporating these insights, we derived our final metric, providing a computationally efficient and theoretically grounded approach to approximate the recovery ratio.
>
>
>
> **Q4: The evaluation isn't really fair. The authors evaluate against SEM. But SEM just wants to recover the embedding and the authors are trying to show what happens if they hide the early parts of the network. This seems like an indication that this isn't a particularly realistic threat model.**
>
> **R4:** Thank you for your comments. As discussed in the general response section, the closed part of our model functions similarly to an encoder in our design. SEM is an attack strategy aimed at replicating the black-box encoder, which we believe aligns with our threat model and is applicable to our setting. If the adversary successfully replicates the closed-source module using SEM, it would constitute a successful model stealing, effectively compromising the entire model. We appreciate the reviewer pointing out the potential confusion regarding SEM, and we have revised the paper to provide a clearer explanation of the SEM process.
>
>
> **Thanks for your helpful comments and we are looking forward to discussing with you to further improve our paper!**

---

> > ### Comment · Reviewer_kKBu · 2024-11-18
> >
> > I have read the other reviews and the author's responses and still strongly recommend rejecting the paper. To summarize my review: there are no semi-open LLMs, and there are no attacks that can steal these models if they did exist, therefore because the authors are operating in a fictitious setting there is no strong prior work in this domain. If anyone cared about this setting, they would immediately arrive at the trivial insights that constitute the entirety of this paper.
> >
> > > Semi-Open LLMs exist
> >
> > Your provided references for this are End-to-end systems that have a closed-source embedding model as one component of the pipeline. Nothing like what you are trying to attack/defend, an LLM that actually has some number of the layers open and some number of the layers closed, exists.
> >
> > > Threat model
> >
> > Stating that the closed-source component acts as a black-box embedding layer does not actually make it an embedding layer. The SVD attack of Carlini et al. only works when there is exactly 1 layer working to take the inputs from representation space to vocabulary space. So there is no analogue here. There is still no evidence that the attack you are studying is a realistic threat.
> >
> > > The paper's insight is trivial
> >
> > The existence of a transition layer follows immediately from the straightforward observation of compounding error at each layer. None of the techniques in the paper are novel.
> >
> > > The method just turns the insight into a metric
> >
> > Stating that it is non-trivial to optimize a non-differentiable metric, and then proceeding to say that you just take a Taylor expansion, does not convince me that you did anything non-trivial here. The final metric is neither computationally efficient nor theoretically principled. It is just "what happened if I closed off the first N layers of the model".
> >
> > > The evaluation is not fair
> >
> > The closed part of your model does not function similarly to an embedding model, so SEM likely is not a fair baseline here.

---

> ### Author Response · Authors · 2024-11-19
> **Response to Reviewer kKBu (1/2)**
>
> Dear Reviewer kKBu:
>
>
>
> Thank you for your time and comments. Below, we have provided responses to your questions to address your concerns.
>
>
>
> **Q1: Semi-Open LLMs exist**: Your provided references for this are End-to-end systems that have a closed-source embedding model as one component of the pipeline. Nothing like what you are trying to attack/defend, an LLM that actually has some number of the layers open and some number of the layers closed, exists.
>
>
>
> **R1:** Thank you for acknowledging the existence of end-to-end systems with semi-open pipelines that combine closed-source embedding models and open-source components. This aligns with the discussion in our paper, where the closed-source module serves as a component providing hidden representations, while the open-source module remains customizable. We use the term "model" broadly to describe these pipelines. However, if the reviewer prefers the term "semi-open pipeline" over "semi-open model," we are open to adopting this terminology, although we believe "model" effectively conveys the overarching concept. Furthermore, "semi-open models" are also referred to as "grey-box models" in references [1-4].
>
>
>
> **Q2: Threat model**: Stating that the closed-source component acts as a black-box embedding layer does not actually make it an embedding layer. The SVD attack of Carlini et al. only works when there is exactly 1 layer working to take the inputs from representation space to vocabulary space. So there is no analogue here. There is still no evidence that the attack you are studying is a realistic threat.
>
> **R2:** We use the term "embedding model" to broadly describe an encoder model that transforms input sentences into high-dimensional real vectors, which aligns with the role of the closed-source component in our framework. However, we acknowledge that in some contexts, "embedding model" may specifically refer to the output of the last decoder layer in large language models. To eliminate potential confusion, we have updated the paper and responses to replace "embedding model" with "encoder." There is a rich literature on stealing/extracting closed-source encoders such as [4-9].
>
>
>
> **Q3: The paper's insight is trivial**: The existence of a transition layer follows immediately from the straightforward observation of compounding error at each layer. None of the techniques in the paper are novel.
>
> **R3:** The observation of compounding error at each layer suggests a gradual change in the impact of individual layers in defending against recovery attacks. However, this gradual change does not directly lead to the sharp transition demonstrated in our theoretical results. Our contribution lies in rigorously establishing a theoretical analysis that proves the existence of this sharp transition, as acknowledged by all other reviewers.
>
> [1] Risks and Opportunities of Open-Source Generative AI https://arxiv.org/pdf/2405.08597
>
> [2] Grey-box Extraction of Natural Language Models https://proceedings.mlr.press/v139/zanella-beguelin21a.html
>
> [3] A Comparative Analysis of White Box and Gray Box Adversarial Attacks to Natural Language Processing Systems https://www.atlantis-press.com/proceedings/iciaai-24/126004152
>
> [4] I Know What You Trained Last Summer: A Survey on Stealing Machine Learning Models and Defences https://dl.acm.org/doi/pdf/10.1145/3595292
>
> [5] StolenEncoder: Stealing Pre-trained Encoders in Self-supervised Learning https://arxiv.org/pdf/2201.05889
>
> [6] Transferable Embedding Inversion Attack: Uncovering Privacy Risks in Text Embeddings without Model Queries [2406.10280 (arxiv.org)](https://arxiv.org/pdf/2406.10280)
>
> [7] Refine, Discriminate and Align: Stealing Encoders via Sample-Wise Prototypes and Multi-relational Extraction https://arxiv.org/abs/2312.00855
>
> [8] Pre-trained Encoder Inference: Revealing Upstream Encoders In Downstream Machine Learning Services https://arxiv.org/pdf/2408.02814
>
> [9] Sentence Embedding Encoders are Easy to Steal but Hard to Defend https://publications.cispa.de/articles/conference_contribution/Sentence_Embedding_Encoders_are_Easy_to_Steal_but_Hard_to_Defend/25287991?file=44691499

---

> > ### Author Response · Authors · 2024-11-19
> > **Response to Reviewer kKBu (2/2)**
> >
> > **Q4:The method just turns the insight into a metric:** Stating that it is non-trivial to optimize a non-differentiable metric, and then proceeding to say that you just take a Taylor expansion, does not convince me that you did anything non-trivial here. The final metric is neither computationally efficient nor theoretically principled. It is just "what happened if I closed off the first N layers of the model".
> >
> > **R4:** Estimating the testing performance of an LLM after a recovery attack, without relying on fine-tuning, is a non-trivial challenge. While implementing the estimator is relatively straightforward, the theoretical insights behind this metric go beyond a simple application of Taylor expansion. Specifically, we use Taylor expansion and neglect higher-order terms based on the theoretical insight that gradient descent does not deviate significantly from the initial point, as demonstrated in convergence analyses for neural networks [10-12]. This approach is both non-trivial and grounded in theoretical principles.
> >
> >
> >
> > Regarding computational efficiency, this metric is advantageous as it requires only the evaluation of a partially initialized LLM on a small test set, without necessitating fine-tuning. In contrast, directly estimating recovered performance after an extraction attack would require fine-tuning, which is computationally expensive and time-consuming.
> >
> >
> >
> > **Q5:The evaluation is not fair:** The closed part of your model does not function similarly to an embedding model, so SEM likely is not a fair baseline here.
> >
> > **R5:** As we discussion in R2, to eliminate potential confusion, we have updated the paper and responses to replace "embedding model" with "encoder." There is a rich literature in stealing/extracting closed-source encoder [4-9], where SEM is one of the attack in this type. We include this attack method since it is one of the main attack method used for stealing or extracting the encoder. Additionally, the SEM attack is used in the grey-box extraction to steal the classification layer in a grey-box (i.e., semi-open) model.
> >
> >
> >
> >
> >
> > [10] Neural tangent kernel: Convergence and generalization in neural networks. https://arxiv.org/pdf/1806.07572
> >
> > [11] Convergence analysis of recurrent neural networks[M]. Springer Science & Business Media, 2013.
> >
> > [12] A convergence analysis of gradient descent for deep linear neural networks. https://arxiv.org/pdf/1810.02281

---

> ### Author Response · Authors · 2024-11-27
> **Official Comment by Authors**
>
> Dear Reviewer kKBu:
>
> We sincerely thank you for raising your concerns regarding the practicality of the motivation behind the problem we addressed. In our general response 2, we have provided further clarification of our motivation and revised the manuscript accordingly. We would greatly appreciate it if you could review our response at your earliest convenience. We respectfully request that you reconsider our manuscript and kindly revise your score.
>
> Regards,
>
> Authors

---

### Official Review · Reviewer_d5Gh · 2024-11-05

**Soundness:** 3
**Presentation:** 2
**Contribution:** 2
**Rating:** 5
**Confidence:** 3

**Summary:**

The paper introduces SCARA, a selective closed-sourcing approach for designing semi-open large language models (LLMs) that enhance customizability while maintaining resilience against recovery attacks. The authors develop an algorithm that strategically keeps only a few bottom layers closed-source, ensuring model flexibility without compromising security. They theoretically demonstrate a "transition layer" within deep transformer models, showing that recovery errors in layers before this point lead to recovery failure, while errors in later layers have a limited impact. SCARA estimates the optimal number of layers to hide using a novel metric based on initial recovery loss, bypassing the need for fine-tuning. The method is applied to five models ranging from 1.3B to 70B parameters, tested across six downstream tasks and sixteen recovery benchmarks. Results show that SCARA improves downstream performance while requiring over ten times fewer closed-source parameters than baselines, achieving improvements, especially in domain-specific tasks like Financial, with 30% higher performance on Llama2-70B. SCARA maintains comparable resilience against recovery attacks.

**Strengths:**

1. The paper introduces SCARA, a method that selectively closes only the bottom layers of semi-open large language models (LLMs) to enhance customizability while maintaining resilience against recovery attacks.
2. It provides a theoretical analysis of the existence of a transition layer in transformer-based models.

**Weaknesses:**

1. **Unclear Motivation for Semi-Open Models:** The market is dominated by closed-source models and fully open-source models. If customization needs are already addressed by existing fine-tuning services provided for closed-source models (e.g., API-based fine-tuning on closed models like GPT-4), it would be insightful to understand the specific motivations and advantages driving the development of a semi-open architecture.
2. **The threat model is not clear.** The threat model concerning recovery attacks on semi-open LLMs is insufficiently defined. The paper does not clearly specify the adversary's capabilities, such as the extent of access to the model's architecture, parameters, or outputs. This lack of clarity makes it challenging to assess the effectiveness of the proposed SCARA method in mitigating such threats.
3. **Insufficient Details on SCARA's Implementation:** The description of SCARA's methodology is vague, particularly regarding the fine-tuning-free metric used to determine which layers to keep closed-source. The paper does not provide a clear explanation of how this metric is calculated, the data required, or the computational resources involved etc.
4. **Evaluation minors:** While the authors present experimental results across multiple models and tasks, the evaluation lacks depth. The paper does not offer a comprehensive analysis of SCARA's performance compared to existing methods, nor does it explore potential trade-offs between customizability and security.

**Questions:**

Please refer to the weaknesses.

---

> ### Author Response · Authors · 2024-11-18
> **Response to Reviewer d5Gh**
>
> Thanks for your kind and helpful comments and we are looking forward to discussing with you to further improve our paper! We are encouraged that you appreciated our contributions, including the theoretical proof and comprehensive experiments. In the response below, we provide answers to your questions in order to address the concerns.
>
> **Q1:Unclear motivation for semi-open models.**
>
> **R1:** Thank you for your constructive and valuable comments. We greatly appreciate your feedback and the opportunity to clarify our motivations, as outlined in R1 within the general response section. We hope our response provides a clearer understanding of the motivations behind our approach.
>
>
>
> **Q2: The threat model is not clear.**
>
> **R2:** Thank you for your constructive and valuable comments. In our general response R2, we have further clarified our threat model and provided a more detailed description of the adversary's capabilities.
>
>
>
> **Q3: Insufficient** **details on SCARA's implementation**
>
> **R3:** Thank you for your comments on SCARA's implementation. We have revised the implementation details part in Section 5.1.The key component of SCARA is the **Recovery Difficulty (RD)**, which quantifies the difficulty of recovering the closed-source module. It is defined as:
>
> $$\text{RD}(I) = \mathbb{E}_{\mathbf{X},Y,{\theta_0}(I)}\left[\ell\left(f(\mathbf{X};{\theta}_0(I)), Y\right)\right]$$
>
> Where:
>
> \- $I$: The set of indices for the closed-source layers, indicating which hidden layers are kept private.
>
> \- **${\theta}_0(I)$**: The initial parameters of the replacement model. Parameters for hidden layers are randomly initialized, while parameters for public layers remain unchanged.
>
> \- $\mathbf{X}$: Input features sampled to target general capabilities from the underlying distribution.
>
> \- $Y$: Labels corresponding to \(\mathbf{X}\).
>
> \- $f$: The final output of the semi-open model.
>
> \- $\ell$: The loss function, where we use cross-entropy loss in this paper.
>
> \- $\mathbb{E}$: The expectation over the joint distribution of random inputs, labels, and randomly initialized parameters of the closed-source module. This expectation is approximated in practice.
>
> **--Approximation of RD--**
>
> \- **Evaluation Dataset**:  To estimate RD, we construct an evaluation set that represents the general capabilities of the victim model. The dataset includes 1,500 samples evenly drawn from two diverse datasets: the MMLU benchmark and Alpaca 52k. These datasets cover tasks such as text comprehension, summarization, generation, code writing, mathematical reasoning, and knowledge reasoning. For more details, please refer to Section 5.1 and Appendix B.4.
>
> \- **Random Initialization of Closed-Source Parameters**:  To evaluate RD for different closed-source layer sets, the hidden layers are randomly initialized using Xavier initialization with PyTorch's default settings. The RD is averaged over three random seeds (20, 42, and 1234) during SCARA implementation.
>
> **--Computational Resources for RD--**
>
> \- For models with up to 7 billion parameters, RD calculation and SCARA execution are performed on a system with **4×RTX 4090 GPUs**, completing in approximately **8 minutes**.
>
> \- For larger models like Llama2-70B, the process is carried out on **4×A100 GPUs**, taking around **30 minutes**.
>
>
> **Q4: No comprehensive comparison to existing methods:**
>
> **R4:** Thank you for your constructive and valuable comments on the comparison to existing methods. While the design of semi-open models has been widely studied in areas such as clustering, to the best of our knowledge, in the domain of complex tasks such as deep reasoning, knowledge-intensive operations, and problem-solving in code or math, SAP [1] is the only approach that serves as a basis for a semi-open LLM construction framework with which we can directly compare. We would sincerely welcome any suggestions for related approaches, and we would be happy to incorporate additional experiments based on your recommendations.
>
>
>
> **Q5: No analysis on potential trade-offs between customizability and security.**
>
> **R5:** We appreciate your feedback regarding the trade-offs between customizability and security. We have revised our manuscript to further explore this trade-off, Specifically, we examine these aspects using Llama2-7B and Phi-2. As shown in Section 5.3, we barely observe significant trade-offs in closed-source set placement but a clear trade-off in the number of hidden layers for smaller models like Phi-2.
>
>
> **Thanks for your kind and helpful comments and we are looking forward to discussing with you to further improve our paper!**
>
> [1] A Split-and-Privatize Framework for Large Language Model Fine-Tuning https://arxiv.org/abs/2312.15603

---

> ### Author Response · Authors · 2024-11-20
> **Official Comment by Authors**
>
> Dear Reviewer d5Gh:
>
> Kindly note that the author-reviewer discussion period is currently ongoing. We would greatly appreciate it if you could review our response when convenient. We earnestly request that you reconsider our manuscript and consider upgrading your score.
>
> Regards,
>
> Authors

---

> ### Comment · Reviewer_d5Gh · 2024-11-21
>
> Thank you for your efforts in rebuttal. Parts of my concerns are addressed, however, the "semi-open model" or "grey-box" motivation under the LLM context is still not clear. I decide to maintain my score.

---

> > ### Author Response · Authors · 2024-11-22
> > **Official Response to Reviewer d5Gh**
> >
> > Thank you for your thoughtful comments and for taking the time to review the rebuttal. Regarding your concern about the motivation for the "semi-open" or "grey-box" model, I would like to further clarify our research intentions and their significance.
> >
> > As OpenAI noted in its response to NTIA’s (National Telecommunications and Information Administration) request regarding LLMs with open weights: “When we did not observe significant misuse effects, this gave us the confidence to openly release the full model weights” [1]. This highlights that the risk of misuse remains a significant concern, prompting many LLM vendors to prefer releasing their models via black-box APIs to better manage and mitigate these risks. At the same time, many companies, like OpenAI, recognize that this approach unfortunately limits LLMs' downstream customizability [2]. Therefore, it is crucial to maintain control over the model and decide whether it could be useful for any malicious purposes. This perspective has inspired us to consider the model as a pipeline—keeping part of it closed-source and proprietary while releasing other parts to enable downstream customization.
> >
> > To the best of our knowledge, only a few prior studies have addressed these challenges. Our work introduces a pioneering paradigm that employs a selective closed-sourcing approach for LLMs, wherein only the key early layers are closed-source, while most layers remain publicly accessible. This forward-thinking solution demonstrates the ability to simultaneously ensure control over the model and enhance customization for downstream tasks, encouraging LLM vendors to adopt greater openness. We take pride in being the first to propose such a framework, which not only safeguards vendor interests but also empowers downstream users with greater customization capabilities. This approach allows researchers broader access to powerful models, enabling advancements in areas such as academic research and creative modifications for downstream performance across various domains. We believe this will foster meaningful discussions and encourage LLM vendors to open more parts of their advanced models, thereby promoting innovation and development in the AI community.
> >
> > Thank you again for your valuable feedback, which is instrumental in refining this work. We are hopeful that the additional clarifications provided might encourage a reconsideration of the score, reflecting the innovative potential of our work to positively influence the AI community.
> >
> > [1] OpenAI’s comment to the NTIA on open model weights
> > https://openai.com/global-affairs/openai-s-comment-to-the-ntia-on-open-model-weights/
> >
> > [2] Open-Source LLM vs Closed Source LLM for Enterprise Use-Cases
> > https://www.fluid.ai/blog/open-source-llm-vs-closed-source-llm-for-enterprise-use-cases

---

> > ### Author Response · Authors · 2024-12-01
> > **Humbly Seeking Further Discussion**
> >
> > Dear Reviewer d5Gh,
> >
> > Kindly note that the author-reviewer discussion period is currently ongoing and is approaching its conclusion. We would greatly appreciate it if you could review our latest general response (General Response 2) at your earliest convenience, where we have provided additional clarification on our **motivation**, particularly regarding the **“semi-open model”**. We sincerely hope this can address your concerns. We humbly request that you reconsider our manuscript and, if appropriate, consider upgrading your score. Thank you for your time and thoughtful review.
> >
> > Best regards,
> >
> > The Authors

---

> ### Author Response · Authors · 2024-11-27
> **Official Comment by Authors**
>
> Dear Reviewer d5Gh:
>
> We sincerely thank you for raising your concerns regarding the practicality of the motivation behind the problem we addressed. In our general response 2, we have provided further clarification of our motivation and revised the manuscript accordingly. We would greatly appreciate it if you could review our response at your earliest convenience. We respectfully request that you reconsider our manuscript and kindly revise your score.
>
> Regards,
>
> Authors

---

### Author Response · Authors · 2024-11-18
**General Response by Authors (1/2)**

Dear area chair and reviewers,

We sincerely thank the reviewers for their time and valuable comments. Overall, the reviewers appreciated our innovative theoretical insights (d5Gh, 1rG5, ag9Z), particularly the identification of transition layers (ag9Z). All reviewers also commended the comprehensive evaluation across various model sizes and benchmarks, as well as the clear and effective presentation of our findings (kKBu, ag9Z, 1rG5). Additionally, some reviewers highlighted the effectiveness of SCARA (1rG5) and the potential impact of our work on future research directions (ag9Z).

We acknowledge the reviewers' concerns regarding the real-world applicability of semi-open models and the details of our threat model. In this general response, we aim to clarify our motivation and provide a more detailed explanation of the threat model and the adversary’s capabilities. In the individual responses, we address each specific comment and question raised. We hope these clarifications and responses adequately address the reviewers’ concerns.

**Q1:** **Are semi-open models widely used? what is the advantage of these semi-open models? What is the motivation of designing semi-open models?**

**R1:** Companies like OpenAI offer fine-tuning APIs [1] for closed-source models (e.g., GPT-4), while META open-sources models like the Llama series [2] for customization. Semi-open models, which combine closed-source embeddings (e.g., OpenAI's text-embedding-ada-002 [3] , Cohere's embeddings [4], or Google Vertex AI [5]) with open-source modules (e.g., LlamaIndex [6], Haystack [7]), have gained popularity for tasks like search [8], recommendation systems [9], anomaly detection [10], and classification [11]. This hybrid approach benefits both vendors and users. Vendors retain control over proprietary components, generate revenue, and reduce the computational demands associated with fine-tuning for downstream tasks [12-14]. Users, in turn, can customize open-source modules offline to better optimize performance for specific tasks [15-16].

Despite their success in tasks like search and recommendation systems, semi-open models face challenges with more complex tasks, such as deep reasoning, knowledge-intensive operations, and problem-solving in code or math. To address these challenges, we propose a novel semi-open framework that enhances performance on complex tasks while ensuring protection against recovery attack. Our approach partitions a pretrained LLM into closed-source and open-source components, fully leveraging the model's capabilities for handling complex tasks. We theoretically prove that errors in the early decoder layers significantly impact performance, while later layers are less critical. Based on this insight, we introduce SCARA, a design that closed-source key early layers while allowing users to customize open modules. To optimize this design, we propose a fine-tuning-free metric, ``recovery difficulty'', to determine the optimal partition point. This framework provides a balance between high customizability and strong resilience against recovery attacks, advancing the capabilities of semi-open models.

We have revised the first paragraph in the introduction to provide more application of semi-open models.

---

> ### Author Response · Authors · 2024-11-18
> **General Response by Authors (2/2)**
>
> **Q2: The threat model is unclear and please provide more details.**
>
> **R2:** In our design, the closed-source component functions as a black-box encoder, generating hidden representations for input data. The open-source component then uses these representations and is fine-tuned for specific downstream tasks. However, since 2016, researchers have studied how to extract parameters and structures from black-box models in attacks known as **model recovery** or **model extraction attacks** [17]. These attacks typically involve querying the black-box encoder, collecting input-output pairs, and training a replacement model that replicates the behavior of the original.
>
> We adopt a common threat model [18-19], where we assume the adversary can query the semi-open model, access its final outputs, and retrieve the representations generated by the closed-source module.  Additionally, as described in [20-21], the adversary is assumed to know the architecture of the closed-source module but does not have direct access to its parameters. By leveraging full access to the open-source components, the adversary can fine-tune a replacement model based on the known architecture of the closed-source module, using the retrieved representations or final outputs as training labels. We examine three recovery strategies under this threat model:
>
> \- **FT-all**: The adversary fine-tunes both the replacement model for the closed-source module and the open-source module together, using the input and final output of the semi-open model.
>
> \- **FT-closed**: The adversary fine-tunes only the replacement model of the closed-source module, keeping the parameters of the open-source module unchanged, using the input and final output.
>
> \- **SEM**: The adversary fine-tunes the replacement model of the closed-source module using the input and the representations generated by the original closed-source model, without involving the open-source module.
>
> We have revised the paragraph begining with "semi-open model recovery attack" in Section 3.1 to provide more details on threat model.
>
>
> **Summary of the Paper Revisions**
>
> The main updates are summarized as follows:
>
> 1. Page 1, Sec. 1: Add related works on semi-open models and their applications.
> 2. Page 3, Figure 2: Correct the description of the attack datasets
> 3. Page 3, Sec. 3.1: Provide more detailed threat model of the Semi-open Model Recovery Attack.
> 4. Page 5, Sec. 4.2: Supplement the definition of Recovery Difficulty (RD) with additional details.
> 5. Page 6, Sec. 5.1: Add more details on SCARA's implementation， and a brief description of the attack dataset sizes.
> 6. Page 9, Sec. 5.3: Rewrite the analysis of the trade-off between customizability and resilience in SCARA.
> 7. Page 9, Figure 6(a)(b): Revise the figure and caption to better illustrate the trade-offs between customizability and resilience to recovery attack.
> 8. Page 28, Appendix B.9: Add a discussion on how resilience transitions vary across specific capabilities.
>
> **References**
>
> [1] https://openai.com/index/introducing-vision-to-the-fine-tuning-api/
>
> [2] https://ai.meta.com/resources/models-and-libraries/
>
> [3] https://platform.openai.com/docs/guides/embeddings/embedding-models
>
> [4] https://cohere.com/embeddingsh
>
> [5] https://cloud.google.com/vertex-aih
>
> [6] https://www.llamaindex.ai/h
>
> [7] https://haystack.deepset.ai/
>
> [8] https://unstructured.io/blog/understanding-embedding-models-make-an-informed-choice-for-your-rag
>
> [9] Vector Search with OpenAI Embeddings: Lucene Is All You Need https://arxiv.org/pdf/2308.14963
>
> [10] Unsupervised Anomaly Detection in Multi-Topic Short-Text Corporahttps://cnrs.hal.science/hal-04471726/file/EACL_2023_ait-saada.pd
>
> [11] Detection of Hate Speech using BERT and Hate Speech Word Embedding with Deep Model https://www.tandfonline.com/doi/full/10.1080/08839514.2023.2166719
>
> [12] Performance Optimization in the LLM World 2024 https://dl.acm.org/doi/10.1145/3629527.3651436
>
> [13] How Open Source Machine Learning Software Shapes AI https://dl.acm.org/doi/10.1145/3514094.3534167
>
> [14] [2024 AI Predictions | NVIDIA Blog](https://blogs.nvidia.com/blog/2024-ai-predictions/)
>
> [15] Peeking Inside the Black-Box: A Survey on Explainable Artificial Intelligence (XAI)
>
> https://ieeexplore.ieee.org/document/8466590/?arnumber=8466590
>
> [16] https://www.preprints.org/manuscript/202307.2142/v2
>
> [17] Stealing Machine Learning Models via Prediction APIs https://arxiv.org/pdf/1609.02943
>
> [18] I Know What You Trained Last Summer: A Survey on Stealing Machine Learning Models and Defences https://dl.acm.org/doi/full/10.1145/3595292
>
> [19] Can't Hide Behind the API: Stealing Black-Box Commercial Embedding Models http://arxiv.org/abs/2406.09355
>
> [20] Stealing Part of a Production Language Model http://arxiv.org/abs/2403.06634
>
> [21] Grey-box Extraction of Natural Language Models https://proceedings.mlr.press/v139/zanella-beguelin21a/zanella-beguelin21a.pdf

---

### Author Response · Authors · 2024-11-27
**General Response 2 by Authors: More Elaboration on Motivation**

We sincerely thank all reviewers for their constructive comments on the semi-open paradigm. We would also like to take this opportunity to further elaborate on the scenarios where semi-open models are widely deployed.

Many vendors now offer large language models (LLMs) that come with high training costs but deliver exceptional performance across various industries [1-2]. Industrial users, such as healthcare organizations, financial institutions, and government agencies, often request these models to be customized with private data, fine-tuned, and deployed locally in on-premises environments [3-5]. However, deploying models without encryption exposes their architecture and parameters, making them susceptible to model stealing attacks that retrieve parameters from CPU, RAM, and vRAM, posing significant risks to intellectual property [5-9]. To mitigate these risks, vendors use hardware-based security techniques like Trusted Execution Environments (TEEs) and encrypted inference, which conceal the model weights and restrict unauthorized access during inference [10-11]

Despite their benefits, TEEs have significant limitations, such as restricted secure memory (e.g., 128MB–256MB in Intel SGX)[7,14], which is insufficient for the gigabytes of memory required by large language models [12,14,15]. As a result, only a portion of the model's layers can be secured within the environment, leaving the remainder exposed. This creates a **semi-open** model setting that balances the need for security and customization.

However, this semi-open approach introduces a tradeoff. Securing or "**closed-sourcing**" more parameters limits the scope of fine-tuning on private data, reducing the model's ability to adapt to specific needs. Conversely, exposing or "**open-sourcing**" more parameters increases vulnerability to model extraction or distillation attacks, where attackers can query the closed-source module, construct input-output pairs, and train a mimic model that replicates its functionality. In this paper, we address this challenge by exploring methods to determine which parts of the model should be closed-sourced in a secure environment, enabling effective fine-tuning while safeguarding against extraction risks.

We have revised the abstract and introduction and hope this updated explanation effectively clarifies our approach and addresses the concerns about the motivation for designing semi-open models.

[1] https://openai.com/index/hello-gpt-4o/

[2] GPT-4 Technical Report https://arxiv.org/pdf/2303.08774

[3] How to run LLMs locally: Hardware, tools and best practices https://www.techtarget.com/searchEnterpriseAI/tip/How-to-run-LLMs-locally-Hardware-tools-and-best-practices

[4] Locally Run Large Language Models May Help Preserve Patient Privacy https://www.techtarget.com/healthtechanalytics/news/366590151/Locally-Run-Large-Language-Models-May-Help-Preserve-Patient-Privacy

[5] Securing AI Model Weights https://www.rand.org/pubs/research_reports/RRA2849-1.html

[6] Deepsniffer: A dnn model extraction framework based on learning architectural hints https://dl.acm.org/doi/pdf/10.1145/3373376.3378460

[7] SoK: All You Need to Know About On-Device ML Model Extraction - The Gap Between Research and Practice https://www.usenix.org/system/files/usenixsecurity24-nayan.pdf

[8] DNN model architecture fingerprinting attack on CPU-GPU edge devices.  https://ieeexplore.ieee.org/abstract/document/9797366

[9] Deepsteal: Advanced model extractions leveraging efficient weight stealing in memories https://arxiv.org/pdf/2111.04625

[10] Shadownet: A secure and efficient on-device model inference system for convolutional neural networks. https://arxiv.org/pdf/2011.05905

[11] No privacy left outside: On the (in-) security of tee-shielded dnn partition for on-device ml.  https://arxiv.org/pdf/2310.07152

[12] CoreGuard: Safeguarding Foundational Capabilities of LLMs Against Model Stealing in Edge Deployment. https://arxiv.org/pdf/2410.13903

[13] Open-Source Solutions for Running LLMs Offline: Benefits, Pros and Cons, and Should You Do It? Is it the Time to Have Your Own Skynet? https://medevel.com/running-llms-offline-benefits-should-you-do-it-1300/

[14] A Fast, Performant, Secure Distributed Training Framework For LLM. https://ieeexplore.ieee.org/abstract/document/10446717

[15] TEESlice: Protecting Sensitive Neural Network Models in Trusted Execution Environments When Attackers have Pre-Trained Models. https://arxiv.org/pdf/2411.09945

---

> ### Comment · Reviewer_kKBu · 2024-11-27
>
> Multiple reviewers made it clear that the authors are studying a threat model that is not well motivated, and the conclusions of the method are basically trivial. In response, the authors have pivoted their paper to focus on a setting where the original model developers hosts some layers are on the TEE that are closed, and some layers outside the TEE that can be finetuned. However, TEE memory is numbered in the MBs, not GBs, and is not even large enough to fit a single layer of any model worth stealing. The setting the authors are considering simply does not exist in the real world, because it doesn't make any sense to only open some layers of the model. Because the problem of "what LLM layers should be open and which should be closed to prevent model stealing during finetuning" has no practical relevance, it has not been studied by any prior work. The authors cite Carlini 2024 to support their case that the problem is worth studying, but Carlini 2024 does not operate in this threat model. Without any prior work to measure their method against, we have to just evaluate the method on its merits. And the conclusions of the method are trivial: just evaluate the model to see how many of the first few layers you can reasonable close, and close those layers. If anyone cared about this problem, this is the natural thing to do, and as another reviewer has pointed out, this is what embedding models already do.
>
> In summary: the paper proposes a trivial solution to an unrealistic threat model. I recommend a reject.

---

> > ### Author Response · Authors · 2024-11-29
> > **Official Response to Reviewer kKBu by Authors (2/3)**
> >
> > **Q3. The authors cite Carlini 2024 to support their case that the problem is worth studying, but Carlini 2024 does not operate in this threat model.**
> >
> > Thank you for your constructive feedback regarding our citations. We reference Carlini’s paper to demonstrate that model extraction attacks can occur at different levels. Specifically, Carlini’s [10] method focuses on **partially extracting** information from fully closed-source production LLMs via API access. Meanwhile, other approaches such as model recovery attacks focus on **replicating the functionality of the entire model**, which has also been extensively studied [11-14]. In the on-premises deployment [10] scenario we consider, attackers aim at replicating the closed-source modules hidden within a TEE so that they can replicate the functionality of the entire model. We would like to note here that this is a very common attack in private deployment, which has been widely studied [8, 11,15-17]. For example, [11] propose a functionally-equivalent extraction attack in model stealing, where attackers train a local model to mimic the target's functionality. Similarly, [15] introduce Knockoff Nets, which steal the functionality of victim models by querying and training a knockoff model based on the obtained predictions. These high-fidelity attacks have also been discussed in Carlini's paper, which notes "In this paper, we focus on high-fidelity attacks. Milli et al. (2019)[18] showed that if an attacker can compute gradients of a target two-layer ReLU model, then they can steal a nearly bitfor-bit equivalent model.".
> >
> >
> >
> > **Q4. The conclusions of the method are trivial: just evaluate the model to see how many of the first few layers you can reasonable close, and close those layers. If anyone cared about this problem, this is the natural thing to do.**
> >
> > Thank you for your valuable feedback on our proposed method. While prior works [4-8] have studied which layers of small models should be protected in a TEE, few studies have explored which parts of LLMs should be secured in a TEE to defend against recovery attacks. SAP [19] proposes opening the bottom six layers while keeping the remaining layers closed-source. However, it remains unclear why open-sourcing the first few layers provides a good balance between customization and security. In our paper, we address this issue theoretically, showing that protecting the bottom layers offers better resilience against recovery attacks than protecting the upper layers, for the same closed-layer size.
> >
> > We believe our method is non-trivial, as no prior work has explicitly addressed the balance between theft risk and customizability in private deployment scenarios for LLMs, nor has anyone studied the effectiveness of protecting different layers to achieve this balance. To our best knowledge,  we are unaware of any work suggesting that bottom-up protection provides sufficient resilience with a smaller protection size. While we acknowledge the reviewer has pointed out related empirical observations about the redundancy of later layers and their limited role in learning higher-order interactions [20], this is not directly applicable to our scenario. To the best of our knowledge, only a few studies address safeguarding vendor interests while allowing greater customization for industrial users. We welcome any related work showing that hiding the first few layers is the natural approach and would be happy to incorporate further discussion based on your recommendations.
> >
> > [10] Stealing part of a production language model.http://arxiv.org/abs/2403.06634
> >
> > [11] High accuracy and high fidelity extraction of neural networks. https://www.usenix.org/system/files/sec20-jagielski.pdf
> >
> > [12] Grey-box extraction of natural language models.  http://proceedings.mlr.press/v139/zanella-beguelin21a/zanella-beguelin21a.pdf
> >
> > [13] Can't Hide Behind the API: Stealing Black-Box Commercial Embedding Models.  https://arxiv.org/pdf/2406.09355
> >
> > [14] Sentence embedding encoders are easy to steal but hard to defend. https://openreview.net/pdf?id=XN5qOxI8gkz
> >
> > [15] Knockoff Nets: Stealing Functionality of Black-Box Models https://arxiv.org/abs/1812.02766
> >
> > [16] Practical black-box attacks against machine learning. https://dl.acm.org/doi/pdf/10.1145/3052973.3053009
> >
> > [17] CoreGuard: Safeguarding Foundational Capabilities of LLMs Against Model Stealing in Edge Deployment. https://arxiv.org/pdf/2410.13903
> >
> > [18] Model reconstruction from model explanations. https://arxiv.org/pdf/1807.05185
> >
> > [19] A Split-and-Privatize Framework for Large Language Model Fine-Tuning http://arxiv.org/abs/2312.15603
> >
> > [20] Inheritune: Training Smaller Yet More Attentive Language Models https://arxiv.org/pdf/2404.08634

---

> > ### Author Response · Authors · 2024-11-29
> > **Official Response to Reviewer kKBu by Authors (3/3)**
> >
> > **Q5.  Embedding models already address the problem we studied.**
> >
> > Thank you for your valuable feedback on our proposed method. We greatly appreciate the opportunity to further clarify and elaborate on the contributions of our work.
> >
> > We acknowledge the reviewer’s point (kKBu 1rG5) that embedding models have already implemented aspects of what we propose, demonstrating that our design has been widely deployed in online scenarios, which we agree is a more trivial case. However, in the context of on-premises deployment of LLMs, it is not necessary for vendors to protect the model from the very first layer. Instead, vendors have the flexibility to secure any layer of the model and store it in a TEE or TDX. Thus, our contribution is twofold. First, we provide theoretical evidence showing that the design widely adopted in online scenarios can effectively balance model theft risks and customization performance. To the best of our knowledge, this has not been explored before. Second, we determine how many layers should be protected in the TEE to offer sufficient defense against recovery attacks. We would like to note that directly determining the optimal number of layers through recovery attacks would require time-consuming fine-tuning, while our method is significantly more efficient, and this approach has not been studied previously.
> >
> >
> >
> > **Thanks for your kind and helpful comments. We hope our responses could address your concerns and raise your confidence and we are looking forward to discussing with you to further improve our work!**

---

> ### Author Response · Authors · 2024-11-29
> **Official Response to Reviewer kKBu by Authors (1/3)**
>
> We sincerely thank you for your time and valuable comments. In the response below, we provide answers in order to address the concerns.
>
> **Q1. TEE memory is numbered in the MBs, not GBs, and is not even large enough to fit a single layer of any model worth stealing**
>
> Thank you for your constructive comments. In order to address your concerns, we provide further clarification regarding TEEs. While individual TEE hardware typically has limited memory, certain recent implementations allow for the extension of secure memory. For instance, Microsoft Azure recently introduced the DCesv5 series VMs [1] (released on October 25, 2024), which leverage Intel’s Trust Domain Extensions (TDX) [2]. TDX is a hardware-based TEE that facilitates the deployment of trust domains (TDs) and can be scaled to support more than 8GB of secure memory [3]. This capacity is sufficient to protect several layers in models with 70B parameters or larger (e.g., a single layer of Llama2-70B requires approximately 1.75G of memory under float16 precision), but is insufficient to prtect the entire model (e.g., Llama2-70B requires more than 140G under float16 precision) .
>
> Besides from the capacity limits, [4-8] also state that using TEE to protect all parameters of an entire model is not a practical solution. As noted in [8], "Attempting to shield the complete DNN model within a TEE (shielding-whole-model) could result in a 50x reduction in the model’s speed". Since TEEs are CPU-based, they do not provide the same level of efficiency for fine-tuning and customization on private data as GPUs. This limitation results in a trade-off, where security is prioritized at the cost of meeting users’ customization needs.
>
> We have revised the introduction of our manuscript to further clarify the capacity of TEE. We hope this revision addresses your concern.
>
>
>
> **Q2. The setting the authors are considering simply does not exist in the real world, because it doesn't make any sense to only open some layers of the model.**
>
> Thank you for your valuable feedback regarding the practicality of the setting we consider. We appreciate the opportunity to provide further examples that support the setting we have explored.
>
> First, we would like to emphasize that model asset protection in private deployments on users' local servers is a real-world challenge, which has been studied since 2018 [4-8]. For instance, a paper published in MobiSys in 2020 [4] states: *"Due to the limited memory of the edge device’s TEE, we partition model layers into more sensitive layers (to be executed inside the device’s TEE) and a set of layers to be executed in the untrusted part of the operating system."*  This scenario aligns with our own, wherein certain layers are partitioned to be executed within the TEE, while the remaining layers are executed outside the trusted environment. The key difference is that the study [4] primarily focuses on defending against membership inference attacks (MIA), whereas our work focuses on mitigating risks related to model theft.
>
> Second, we consider fine-tuning open-sourced layers to allow users to customize them using their local, private data. In our design, we employ the SCARA to identify and isolate a few layers within the TEE to secure the model, while allowing the remaining layers to be accessible for customization. As discussed in Section 5.3 of our manuscript, hiding more layers reduces the model’s customizability for downstream tasks but does not significantly affect the resilience provided by securing only the first few layers identified by SCARA. Meanwhile, protecting only those layers identified by SCARA offers customizability comparable to that of a fully open-source model. Therefore, the semi-open model constructed using SCARA strikes a balance between model security and customization, effectively addressing the trade-off between security and customization in on-premises deployments [9].
>
> [1] https://learn.microsoft.com/en-us/azure/virtual-machines/sizes/general-purpose/dcesv5-series?tabs=sizebasic
>
> [2] https://www.intel.com/content/www/us/en/developer/tools/trust-domain-extensions/overview.html
>
> [3] https://www.intel.com/content/www/us/en/developer/articles/technical/tdx-performance-isolated-partitioned-vms.html
>
> [4] Darknetz: towards model privacy at the edge using trusted execution environments http://arxiv.org/abs/2004.05703
>
> [5] No privacy left outside: On the (in-) security of tee-shielded dnn partition for on-device ml.  https://arxiv.org/pdf/2310.07152
>
> [6] Confidential Inference via Ternary Model Partitioning. https://arxiv.org/abs/1807.00969
>
> [7] Slalom: Fast, verifiable and private execution of neural networks in trusted hardware. https://arxiv.org/abs/1806.03287
>
> [8] TEESlice: Protecting Sensitive Neural Network Models in Trusted Execution Environments When Attackers have Pre-Trained Models. https://arxiv.org/pdf/2411.09945
>
> [9] Securing AI Model Weights https://www.rand.org/pubs/research_reports/RRA2849-1.html

---

### Note · Authors · 2024-12-16

I have read and agree with the venue's withdrawal policy on behalf of myself and my co-authors.